# A bin-microphysics parcel model investigation of secondary ice formation in an idealised shallow convective cloud

Rachel L. James[1,a], Jonathan Crosier[1,2], and Paul J. Connolly[1]

[1]Department of Earth and Environmental Sciences, The University of Manchester, Manchester, UK
[2]National Centre for Atmospheric Science (NCAS), University of Manchester, Manchester, UK
[a]now at: School of Chemistry, University of Leeds, Leeds, UK

**Correspondence:** Rachel L. James (r.james@leeds.ac.uk) and Paul J. Connolly (paul.connolly@manchester.ac.uk)

**Abstract.** We provide the first systematic study of ice formation in idealised shallow clouds from collisions of supercooled water drops with ice particles ('mode 2'). Using the University of Manchester bin-microphysics parcel model, we investigated the sensitivity of ice formation due to mode 2 for a wide range of parameters: aerosol particle size distribution, updraft speed, cloud base temperature, cloud depth, ice-nucleating particle concentration and freezing fraction of mode 2. We provide context
to our results with other secondary ice production mechanisms as single mechanisms and combinations (rime-splintering, spherical freezing fragmentation of drops ['mode 1'] and ice-ice collisions). There was a significant sensitivity to aerosol particle size distribution when updraft speeds were low (0.5 m s-1); secondary ice formation did not occur when the aerosol particle size distribution mimicked polluted environments. Where secondary ice formation did occur in simulated clouds, significant ice formation in the shallower clouds (1.3 km deep) was due to mode 2 or a combination which included mode 2.
The deeper clouds (2.4 km deep) also had significant contributions from rime-splintering or ice-ice collisions SIP mechanisms. While simulations with cloud base temperatures of 7 °C were relatively insensitive to ice-nucleating particle concentrations, there was a sensitivity in simulations cloud base temperatures of 0 °C. Increasing the ice-nucleating particle concentration delayed ice formation. Our results suggest that collisions of supercooled water drops with ice particles may be a significant ice formation mechanism within shallow convective clouds where rime-splintering is not active.

## 1 Introduction

Ice crystals in clouds can significantly effect weather and climate (Elsom, 2001; Changnon, 2003; Field and Heymsfield, 2015; Púčik et al., 2019). Yet, the formation of ice crystals, especially in mixed-phase clouds, is still not well understood. Where temperatures are sub-zero, yet warmer than the homogeneous freezing temperature of water, -35°C, freezing of supercooled water drops can occur via the action of ice-nucleating particles (INPs). However, INPs are relatively rare in the atmosphere, typically falling between ice crystal number concentrations (ICNCs) of $1 \times 10^{-4}$ L$^{-1}$ to 1 L$^{-1}$ at -10°C (Kanji et al., 2017). In contrast, many observations show enhancement in ICNCs, orders of magnitude higher than those predicted by the action of INPs, for a variety of clouds across the globe (e.g. Crawford et al., 2012; Lloyd et al., 2015; Lasher-Trapp et al., 2016; O'Shea et al., 2017). Secondary ice production (SIP), involving the formation of ice from pre-existing ice crystals, is often proposed

to explain these higher observed ICNCs. However, there are still many uncertainties within these SIP mechanisms (e.g. see reviews by Field et al., 2017; Korolev et al., 2020).

While several SIP mechanisms exist, e.g. rime-splintering, fragmentation of freezing drops, ice-ice collisional breakup, only rime-splintering is widely implemented in numerical weather prediction (NWP) models. Rime-splintering, also called the Hallett-Mossop process, is thought to occur when supercooled water droplets accrete on ice particles and splinter during freezing. Rime-splintering is only active in a narrow temperature range, between -3 and -8 °C, and when supercooled water drop diameters of both $\lesssim$ 13 μm and $\gtrsim$ 24 μm are present (Hallett and Mossop, 1974; Mossop and Hallett, 1974; Mossop, 1978). Although rime-splintering may account for the differences between ICNCs from observations and those predicted from the action of INPs for some mixed phase clouds (e.g. Harris-Hobbs and Cooper, 1987; Blyth and Latham, 1993, 1997; Phillips et al., 2001, 2005; Crosier et al., 2011; Crawford et al., 2012; Taylor et al., 2016; Huang et al., 2017a), there are also cases where it does not. For example, rime-splintering may be too slow to explain observed ICNCs (Rangno and Hobbs, 1991; Sotiropoulou et al., 2020, 2021), or clouds may have temperatures or drop size distributions outside the range in which rime-splintering is active (Rangno and Hobbs, 1991, 2001; Fridlind et al., 2007; Lawson et al., 2015). A recent study by Luke et al. (2021) showed that radar observations of Arctic mixed-phase clouds indicates that the freezing fragmentation of drops is more efficient at enhancing ICNCs than rime-splintering.

Despite the preferential inclusion of the rime-splintering SIP mechanism in NWP models, it does not account for ice enhancement in all mixed-phase clouds. Other SIP mechanisms may account for ice enhancement in mixed-phase clouds, given the wider range of conditions in which they are active. For example, in contrast to rime-splintering, fragmentation of freezing drops can occur over a wider temperature range, between 0 and -30 °C, with a drop size range between 4–1000 μm (e.g Table 1 of Korolev et al., 2020). Recently, Phillips et al. (2018) described the fragmentation of freezing drops as occurring via two modes. The first mode, 'mode 1', occurred when drops froze due to the action of INPs or ice particles smaller than the freezing drop and maintained their symmetrical symmetry. Based on the available laboratory literature of drops in free fall only, Phillips et al. (2018) found that the maximum number of fragments occurred at a thermal peak of approximately -15°C, with larger drops (e.g. mm-sized) forming more fragments than smaller drops (e.g. sub-mm sized). From here on, we shall refer to the freezing fragmentation of drops as mode 1, using the classification described in Phillips et al. (2018). The second mode described by Phillips et al. (2018), 'mode 2', occurred when drops collided with larger ice particles causing fragmentation upon collision disrupting the symmetrical symmetry. Phillips et al. (2018) used theory to describe the fragmentation due to a lack of laboratory experiments, and we have recently shown proof-of-concept laboratory results for the potential of mode 2 as a SIP mechanism (James et al., 2021). Unlike rime-splintering and mode 1, the theoretical description of mode 2 does not have a thermal peak and increases linearly with dimensionless energy and supercooling. However, large experimental uncertainties exist around the treatment of the number of secondary drops emitted per impact with a more massive ice surface. Another SIP mechanism, ice-ice collisional breakup, may also occur in mixed-phase clouds. Experiments by Vardiman (1978) and Taka-hashi et al. (1995) showed that the number of fragments increased as a function of change in momentum or collisional energy, respectively.

Various models include these additional SIP mechanisms, sometimes alongside the rime-splintering mechanism (Phillips et al., 2018; Sotiropoulou et al., 2020; Zhao et al., 2021; Zhao and Liu, 2021; Georgakaki et al., 2022; Zhao and Liu, 2022; Huang et al., 2022). For example, Sotiropoulou et al. (2021) included the ice-ice collision breakup mechanism in their Weather and Research Forecasting Model simulations of summer clouds over the Antarctic coast. Their results suggested that ice-ice collisional breakup alone could be responsible for observed ICNCs in Antarctic clouds, despite including the rime-splintering mechanism in their simulations. Georgakaki et al. (2022) found that a combination of ice-ice collisional breakup and external ice seeding could account for the observed ICNCs in wintertime alpine mixed-phase clouds observed during the CLACE campaign.

There is also increasing evidence suggesting that combinations of multiple SIP mechanisms are more likely responsible for ice enhancement in mixed-phase clouds. For example, Sotiropoulou et al. (2020) modelled Arctic stratocumulus clouds and modelled ICNCs only matched observations when rime-splintering and ice-ice collisional breakup were combined. Phillips et al. (2018) showed that rime-splintering, mode 1 and mode 2 were all required in parcel simulations of tropical maritime deep convection to match the observed ICNCs. Combinations of SIP mechanisms are also important on a global scale, as demonstrated in a study by Zhao and Liu (2021) which showed that combined effect of mode 1, mode 2 and ice-ice collisional breakup SIP mechanisms triggered changes in the liquid-ice partitioning and cloud radiative forcing.

While SIP can happen in a variety of clouds, in this paper we will focus on idealised shallow convective clouds. Shallow convective clouds are widespread across the globe, but ice formation in cloud tops warmer than -10 °C is still largely uncertain (Rangno and Hobbs, 2005; Blyth and Latham, 1997; Hobbs and Rangno, 1990; Sun et al., 2010). For example, Hobbs and Rangno (1990) showed that rapid ice enhancement occurred within 15 min of formation in polar cumulus maritime clouds. While rime-splintering was ruled out as a SIP mechanism for being too slow, there was no consideration of other SIP mechanisms. In shallow convective clouds across the Southern Ocean, SIP mechanisms may account for ice enhancement (Huang et al., 2017b). While rime-splintering was suggested as the cause of ice enhancement in shallow aged wintertime cumulus cloud across the UK, no other secondary ice mechanisms were considered (Crawford et al., 2012).

In this paper, we present a bin-microphysics investigation of idealised shallow convective clouds. We focus on ice enhancement by the recently discovered mode 2 SIP mechanism in shallow convective clouds and investigate its sensitivity to initial aerosol size distributions, updraft speeds, INP concentrations, cloud depths and cloud base temperatures. To put the ice enhancement of mode 2 into context with the other SIP mechanisms, we run simulations with rime-splintering, mode 1 and ice-ice collisional breakup separately as well as for all possible combinations including mode 2. The model used to explore this large parameter space is the University of Manchester bin-microphysics model, BMM (e.g. Fowler et al., 2020), which is an update on the Aerosol Cloud and Precipitation Interactions Model, ACPIM (e.g. Connolly et al., 2009, 2012, 2014; Simpson et al., 2014).

## 2 Model

### 2.1 Model description

All simulations in this paper used the bin microphysics model (BMM, https://github.com/UoM-maul1609/bin-microphysics-model), an adapted version of the control model described in Fowler et al. (2020), developed at the University of Manchester. The bin microphysics model includes activation of cloud droplets and condensation / deposition from water vapour, collision and coalescence of water drops, inertial impaction of aerosol particles, ice-ice aggregation, riming and secondary ice processes.

Aerosol particles are represented as multiple log-normal modes of different chemical compositions (externally-mixed modes). Each externally-mixed mode is described by an internal mixture that has the same chemical composition across all sizes. The BMM is initialised by summing multiple log-normal size distributions:

$$\frac{dN}{d\ln(D_a)} = \frac{N_T}{\sqrt{2\pi}\ln\sigma_g} \exp\left[-\frac{\ln^2\left(\frac{D_a}{D_a,m}\right)}{2\ln^2\sigma_g}\right] \tag{1}$$

where $N$ is the number density of aerosol particles, $D_a$ is the aerosol particle diameter for the mode, $N_T$ is the total number of aerosol particles in the mode, $D_a, m$ is the median aerosol particle diameter for the mode and $\sigma_g$ is the geometric standard deviation of the logarithmic distribution.

The activation of cloud condensation nuclei is calculated from condensation of liquid water onto the aerosol particles with the equilibrium vapour pressure described by $\kappa$-Kohler theory, where the size and hygroscopicity of an aerosol particle is related by a single parameter, $\kappa$ (Petters and Kreidenweis, 2007). The rate of drop growth via diffusion takes into account mass accommodation through modified diffusivity and conductivity terms (Jacobson, 2005; H.R. Pruppacher, 2010). While initial growth of cloud drops occurs via the diffusional growth equation, later growth to raindrops occurs via the collision-coalescence process. Collision-coalescence growth is described by the stochastic collection equation which is solved using the method of Bott (1998), and collisional efficiencies are calculated according to the Long (1974) kernel. The model treats binned distribution for liquid particles and a separate binned distribution for ice particles.

Homogeneous freezing from a supercooled water drop follows the method described in Koop et al. (2000). Heterogeneous freezing occurring via ice nucleating particles is calculated using the DeMott et al. (2010) ice nucleation parameterisation, which requires knowledge of the aerosol particle number density with diameter $\geq 0.5$ μm. The same parameterisation is used to describe the freezing of rain drops. We first determine the number of aerosol particles with diameter $\geq 0.5$ μm contained within a rain drop and multiply this by the number concentration of particles within the same category. The number of ice nucleating particles that are active is then calculated using the DeMott et al. (2010) parameterisation. This is scaled by how many aerosol particles are contained within the drop giving the number of frozen drops in that category.

Ice particle growth from the vapour is described using a growth rate which takes into account mass accommodation through modified diffusivity and conductivity terms (Jacobson, 2005). Once formed, ice particles grow according to the model described in Chen and Lamb (1994). The ice particle bins carry properties that are averaged within a mass bin. These properties are the aspect ratio of the ice crystals; the volume of the crystals; the rime mass; and the number of ice crystal 'monomers' per ice

particle. Ice-ice aggregation and riming are also calculated using the method of Bott (1998), which is modified to transport the extra properties discussed above. The terminal velocity of ice particles is determined from Heymsfield and Westbrook (2010) based on the mass and shape of the ice particles.

## 2.2 Secondary ice parameterisations

Four SIP mechanisms are included in the model. These are the rime-splintering mechanism (RS); Collisional breakup of ice particles during collision (CB); droplet shattering during symmetrical freezing (M1) and droplet shattering during asymmetrical freezing (M2). Their parameterisations are given below.

### 2.2.1 Rime-splintering (RS)

We use a modified version of the RS parameterisation given by Reisner et al. (1998). This parameterisation is based on lab-
130 oratory experiments by Hallett and Mossop (1974) who found a maximum splinter production rate of 350 splinters mg$^{-1}$ of rimed ice around -5°C.

$$\frac{dN_{RS}}{dt} = 350 \times 10^6 \times \left(\frac{dm_r}{dt}\right) \times f_{RS}(T) \tag{2}$$

where $\frac{dN_{RS}}{dt}$ is the splinter production rate due to RS, $\frac{dm_r}{dt}$ is the riming rate, and $f_{RS}(T)$ is the temperature-dependent function of RS that has a maximum of unity at $T = -5$ °C and zero at $T < -2.5$ °C and $T > -7.5$ °C.

135 ### 2.2.2 Ice-ice collisional breakup (CB)

We use the parameterisation derived by Phillips et al. (2017) which is based on the energy conservation principle, and relates collisional kinetic energy with ice particle habits.

$$N_{CB} = \alpha A \left( 1 - \exp \left\{ -\left[ \frac{CK_{0(CB)}}{\alpha A} \right]^{\gamma} \right\} \right) \tag{3}$$

where $\alpha$ is the equivalent spherical surface area of the smaller colliding particle ($\alpha = \pi D^2$, $D$ the diameter of the smaller
colliding particle), $A$ represents the number density of the breakable asperities in the region of contact, $C$ is the asperity-fragility coefficient, $\gamma$ is a parameter related to riming intensity, $K_{0(CB)}$ is the collisional kinetic energy calculated by:

$$K_{0(CB)} = \frac{1}{2} \left( \frac{m_1 m_2}{m_1 + m_2} \right) (v_1 - v_2)^2 \tag{4}$$

where $m_1$ and $m_2$ are the mass of the colliding ice particles, and $v_1$ and $v_2$ are their fall speeds.

Three types of collisions were identified: graupel or hail with graupel/hail, hail with hail and snow or ice crystals (dendritic
or spatial planar) with any ice particle. The terms $A$, $C$, $\gamma$ are dependent on the type of colliding pairs, details of which can be

found in Table 1 of Phillips et al. (2017). In the BMM we implement the parameterisation by distinguishing between the type of ice particle using the modelled density, rime-mass and aspect ratio, details of which are given in Table 1.

**Table 1.** Parameters used to identify the surface of the ice category, which go into the Eq. 3 and are used in conjunction with Table 1 of Phillips et al. (2017). $D_S$ is the diameter of the smaller colliding particle, $R_{FS}$ is the rime fraction of the smaller colliding particle, $R_{FL}$ is the rime fraction of the larger colliding particle, $\rho_L$ is the density of the larger colliding particle and $\Phi_S$ is the aspect ratio of the smaller colliding particle.

| Collision type | $D_s$ (m) | $R_{FS}$ | $R_{FL}$ | $\rho_L$ (kg m$^{-3}$) | $\Phi_S$ |
|---|---|---|---|---|---|
| Graupel with graupel or hail | $5\times10^{-4}$ to $5\times10^{-3}$ | $0.5 \leq R_{FS} < 0.9$ | $\geq 0.5$ | n/a | n/a |
| Hail with hail | n/a | $\geq 0.9$ | $> 0.9$ | n/a | n/a |
| Dendrites with any ice particle | $5\times10^{-4}$ to $5\times10^{-3}$ | $< 0.5$ | n/a | $< 400$ | $< 1$ |
| Spatial planar with any ice particle | $5\times10^{-4}$ to $5\times10^{-3}$ | $< 0.5$ | n/a | $\geq 400$ | $< 1$ |

### 2.2.3 Freezing Fragmentation of drops: Mode 1 (M1)

We use the parameterisation derived by Phillips et al. (2018) who compiled available laboratory data on the freezing fragmentation of drops. If fragmentation occurred, two size regimes were identified. Small fragments, which had smaller diameters but larger number of fragments and large fragments, which had larger diameters but smaller number of fragments. Phillips et al. (2018) found that M1 was most efficient around -15 °C. The parameterisations for the total number of fragments ($N_{M1T}$) and number of large fragments ($N_{M1L}$) are given below:

$$N_{M1T} = F(D)\Omega(T)\left[\frac{\zeta\eta^2}{(T-T_0)^2+\eta^2} + \beta T\right] \tag{5}$$

$$N_{M1L} = min\left\{F(D)\Omega(T)\left[\frac{\zeta_B\eta_B^2}{(T-T_{B0})^2+\eta_B^2}\right], N_T\right\} \tag{6}$$

where $\zeta, \eta, \beta, \zeta_B, \eta_B, T_0, T_{B0}$ are parameters taken from Phillips et al. (2018) which were derived from fitting Lorentzians to the laboratory data. $F(D)$ and $\Omega(T)$ are interpolation functions for the onset of fragmentation, and T is the freezing temperature of the drop.

### 2.2.4 Freezing Fragmentation of drops: Mode 2 (M2)

We use the parameterisation derived by Phillips et al. (2018) from theory and based on the assumption that fragmentation is controlled by the ratio of collision kinetic energy ($K_0$) and initial surface energy, referred to as dimensionless energy (*DE*). For M2 fragmentation to occur, the drop diameter must be greater than 0.15 mm and the mass of the drop must be less massive than the ice particle. First, $K_0$ and *DE* are calculated:

$$K_0 = \frac{1}{2}\left(\frac{m_d m_i}{m_d + m_i}\right)(v_d - v_i)^2 \tag{7}$$

$$DE = \frac{K_0}{\gamma_l \pi D_d^2} \tag{8}$$

where $m_d$ and $m_i$ are the mass of the drop and ice respectively, $v_d$ and $v_i$ are the velocity of the drop and ice particle respectively, $D_d$ is the diameter of the drop and $\gamma_l$ is the surface tension of liquid water which is set as a constant of 0.073 J m$^{-2}$.

Then the number of fragments per drop accreted due to M2 ($N_{M2}$) is calculated:

$$N_{M2} = 3\Phi(T) \times [1 - f(T)] \times \max(DE - DE_{crit}, 0) \tag{9}$$

where $\Phi$ is the probability of any drop in the splash containing ice, *f(T)* is the temperature dependent mass fraction of a drop frozen by the end of stage 1 freezing ($f(T) = -c_w T / L_f$) with $c_w$ the specific heat capacity of liquid water and $L_f$ the latent heat of freezing, $DE_{crit}$ is the critical value of DE for onset of splashing ($\sim$0.2). We used the experimentally determined $\Phi$ value of 0.3 (James et al., 2021).

## 3   Method

### 3.1   Initial conditions

The model is run using an initial relative humidity of 0.95 and an initial altitude of 900 m. The parcel of air ascends at a constant updraft speed and the depth of the cloud is controlled by setting the updraft speed to zero after a predefined time ($t_w$). All simulations run for a total of 8000 s (133.3 min), and for each sensitivity the SIP mechanisms, RS, CB, M1 and M2 were investigated individually and as combinations. We ran all possible combinations of SIP mechanisms which gave a total of 15 simulations for each sensitivity investigated in addition to a control simulation with no SIP mechanisms. The combinations are given in Table A1 of the Appendix.

### 3.2   Investigated sensitivities

A summary of the sensitivities investigated in this paper are given in Table 2, and further details are given below.

### 3.2.1   Cloud depth, updraft speed and $t_w$

Two cloud depths were investigated, a shallower cloud with a depth of 1.3 km and a deeper cloud with a depth of 2.4 km. To maintain the desired cloud depth the updraft speed was set to zero after $t_w$ was reached. For the shallower clouds, this was at 2600, 650 and 130 s for updraft speeds of 0.5, 2 and 10 m s$^{-1}$ respectively. For the deeper clouds, this was at 5000, 1250 and 250 s for updraft speeds of 0.5, 2 and 10 m s$^{-1}$ respectively.

**Table 2.** Sensitivities investigated in idealised shallower (1.3 km) and deeper (2.4 km) clouds. See Section 3.2.2 for the details of the aerosol size distribution.

| Sensitivity | Shallower cloud (1.3 km) | Deeper cloud (2.4 km) |
|---|---|---|
| Aerosol size distribution | 'Natural', 'Near city' | 'Natural', 'Near city' |
| Cloud base temperature (°C) | 0, 7 | 0, 7 |
| Updraft speed (m s$^{-1}$) | 0.5, 2, 10 | 0.5, 2, 10 |
| INP concentration | ×0.1, ×1, ×10 | ×1 |
| M2 $\Phi$ | 0.001, 0.01, 0.1, 0.3, 0.5 | 0.3 |
| M2 $DE_{crit}$ | 0.2, 3, 6 | 0.2 |

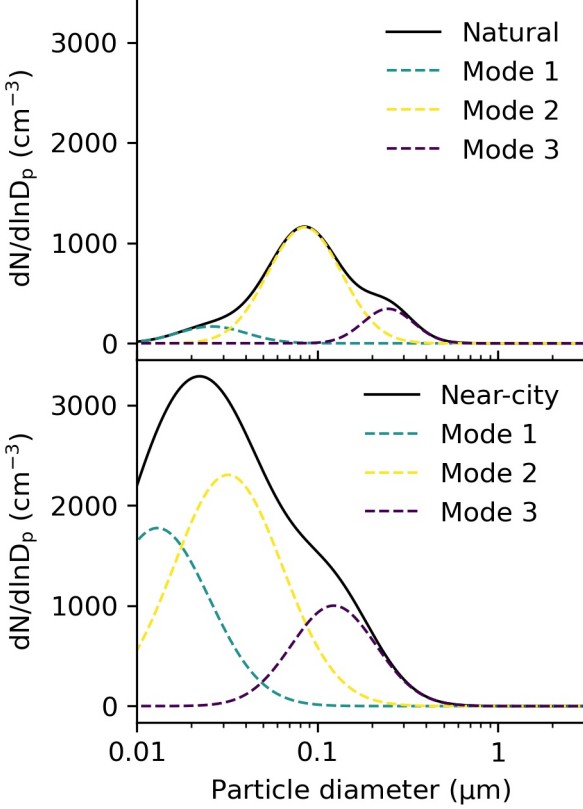

**Figure 1.** Aerosol particle size distribution for a natural aerosol size distribution (top panel) and a near-city aerosol size distribution (bottom panel) based on three lognormal size distribution fits provided by Crooks et al. (2018) to describe each aerosol size distribution based on summertime afternoon measurements from Van Dingenen et al. (2004).

**Table 3.** Lognormal size distribution fits for the natural and near city aerosol size distributions. Fits were obtained from Crooks et al. (2018) based on summertime afternoon measurements presented in Van Dingenen et al. (2004). $N_i$ the total number density (cm$^{-3}$), $d_{i,m}$ the median diameter (nm) and $\ln \sigma_i$ the standard geometric mean deviation

| Aerosol size distribution | Mode A | | | Mode B | | | Mode C | | |
|---|---|---|---|---|---|---|---|---|---|
| | $N_A$ | $D_{mA}$ | $\ln \sigma_A$ | $N_B$ | $D_{mB}$ | $\ln \sigma_B$ | $N_C$ | $D_{mC}$ | $\ln\sigma_C$ |
| Natural | 185 | 26 | 0.44 | 1364 | 85 | 0.47 | 276 | 246 | 0.32 |
| Near city | 2938 | 13 | 0.66 | 3989 | 32 | 0.69 | 1356 | 123 | 0.54 |

### 3.2.2 Aerosol size distribution

Two different aerosol size distributions were investigated, a 'natural' aerosol size distribution and a 'near-city' aerosol size distribution. Natural and near-city refer to a study by Van Dingenen et al. (2004), which compiled data on aerosol measurements in Europe between 1996–2001. A natural aerosol size distribution refers to measurements taken from sites > 50 km from a large pollution source such as a city or motorway. A near city aerosol size distribution refers to measurements taken from sites between 3–10 km from a major pollution source. We use three lognormal size distribution fits provided by Crooks et al. (2018) to describe each aerosol size distribution based off summertime afternoon measurements from Van Dingenen et al. (2004). Details of these fits are provided in Table 3 and plotted as aerosol particle size distributions in Fig. 1. For all simulations we used an aerosol size bin range from 10 nm to 3 μm. For the aerosol we used 60 particle size bins and a further 80 bins for the cloud and precipitation, as a balance between resolution and computational expense considering that a total of 1162 simulations were performed. Ammonium sulphate was the only chemical composition investigated. It has a molecular weight of 132.14 g mol$^{-1}$, a density of 1.77 g cm$^{-3}$ and a $\kappa$ value of 0.61.

### 3.2.3 Cloud base temperature

Two initial cloud base temperatures were investigated, a warmer cloud base of 7 °C and a colder cloud base of 0 °C. For the shallower cloud simulations, clouds with a warmer cloud base had cloud top temperatures of -1 °C and clouds with a colder cloud base had cloud top temperatures of -9 °C. For the deeper cloud simulations, clouds with a warmer cloud base had cloud top temperatures of -9 °C and clouds with a colder cloud base had cloud top temperatures of -19 °C. The initial starting pressure for the warmer cloud base (7 °C) was set to 90781 Pa and 90271 Pa for the colder cloud base (0 °C). Figures showing the temperature profiles as functions of simulation time for the shallower and deeper clouds are given in Figs. S1 & S2 of the Supplement.

### 3.2.4 M2 $\Phi$ and DE$_{crit}$

The M2 parameterisation depends on both the probability of the splash containing ice ($\Phi$) and the onset of splashing ($DE_{crit}$) as shown in Eq. 9. Both of these parameters are determined on experimental studies and present a source of uncertainty.

Therefore, we individually investigated the M2 $\Phi$ and $DE_{crit}$ for shallower clouds with an updraft speed of 2 m s$^{-1}$ for both natural and near-city aerosol size distributions and warmer and colder cloud bases.

## 4 Results

The results section is organised as follows. The shallower (1.3 km) convective cloud results are split into two main sections based on the aerosol size distribution. Section 4.1 gives the results for simulations using the natural aerosol size distribution for cloud base temperature, updraft speed, INP concentrations and M2 $\Phi$ parameter sensitivities. Section 4.2 gives the results for the same sensitivities in Sect. 4.1, but for simulations using a near-city aerosol size distribution. The deeper ($\sim$2.4 km) convective cloud is also split into two main sections based on aerosol size distribution. Section 4.3 gives the results for cloud base temperature and updraft speed sensitivities for a natural aerosol size distribution and Sect. 4.4 gives the results for the same sensitivities for simulations using a near-city aerosol size distribution.

### 4.1 Shallower cloud: Natural aerosol size distribution

#### 4.1.1 Sensitivity test: Cloud base temperature and updraft speed

We performed simulations using a natural aerosol size distribution for two cloud base temperatures, 7 and 0 °C, and three updraft speeds, 0.5, 2 and 10 m s$^{-1}$. The SIP mechanisms, RS, CB, M1 and M2, were investigated individually and for all possible combinations. We also performed a control simulation with no SIP mechanisms.

For the warmer cloud bases simulations with updraft speeds of 0.5 and 2 m s$^{-1}$, the control ICNCs were relatively constant at $6 \times 10^{-4}$ and $2 \times 10^{-3}$, respectively. The simulation with an updraft speed of 10 m s$^{-1}$ had a peak maximum at 92 min of 0.09 L$^{-1}$. In contrast, the control ICNCs of the colder cloud base simulations exhibited a maximum peak at approximately 91, 93 and 99 min for updraft speeds of 0.5, 2 and 10 m s$^{-1}$ respectively. The maximum ICNCs were 0.4, 1.3 and 1.6 L$^{-1}$ for updraft speeds of 0.5, 2 and 10 m s$^{-1}$, approximately 2–3 orders of magnitude greater than their corresponding warmer cloud base simulations. For reference, Fig. S3 of the Supplement shows the ICNCs of the control simulations for a natural aerosol size distribution for both cloud base temperatures and all updraft speeds.

For both cloud base temperatures, the maximum cloud drop number concentration (CDNC) for simulations with updraft speeds of 0.5, 2 and 10 m s$^{-1}$ was 400, 900 and 1200 cm$^{-3}$ respectively, and these maximums occurred within the first few minutes of the simulations. The CDNCs decreased gradually, by a factor of 2–3, throughout the simulations. There was no observable difference between the control simulations and simulations with SIP mechanisms. For reference, Fig. S4 of the Supplement shows the CDNC for all simulations with a natural aerosol size distribution.

Figure 2 shows the ice enhancement (i.e. the difference between the SIP ICNC and control ICNC) as a function of simulation time for a shallower convective cloud with a natural aerosol size distribution. The initial updraft speed remained constant until a threshold time ($t_w$) and was dependent on the updraft speed and then set to zero to simulate a cloud at the desired depth.

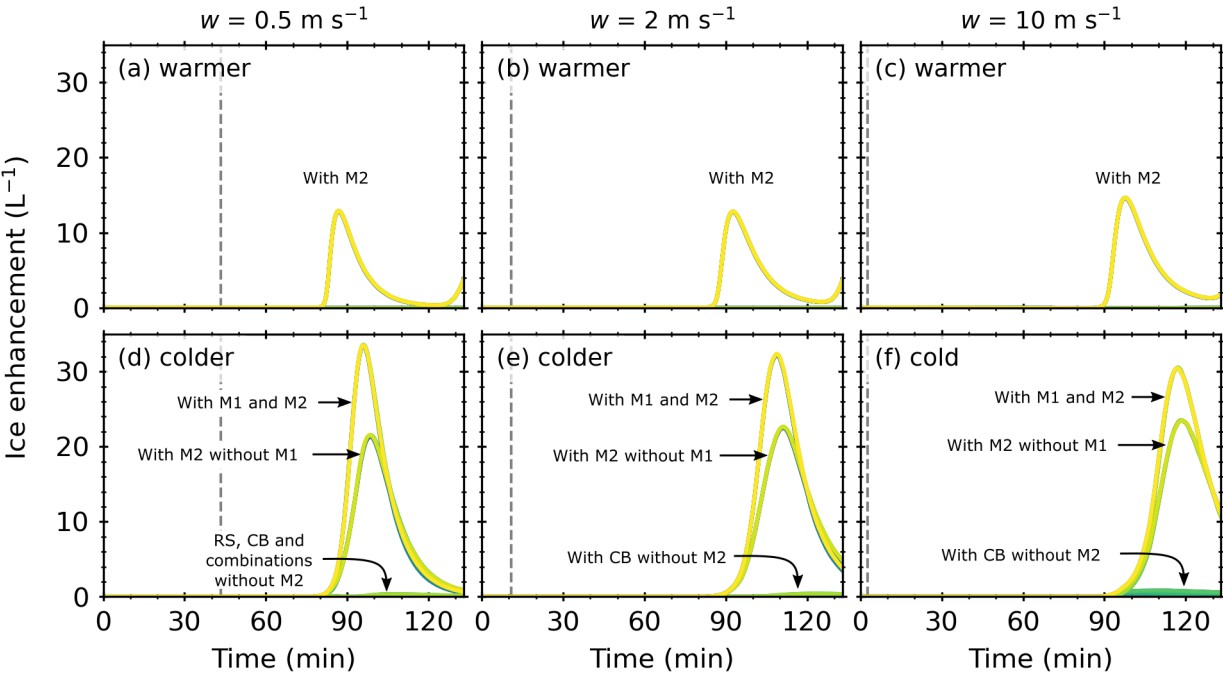

**Figure 2.** Ice enhancement against simulation time for a shallower (1.3 km) cloud with a natural aerosol size distribution. Warmer refers to cloud base temperatures of 7 °C, and colder refers to cloud base temperatures of 0 °C. The grey dashed lines indicate the threshold time where the updraft was turned off. Plots are annotated to indicate the processes that were active.

In Fig. 2, $t_w$ is represented by a dashed line at approximately 21, 11 and 2 min for updraft speeds of 0.5, 2 and 10 m s$^{-1}$

respectively.

For all updraft speeds with a warmer cloud base , ice enhancement occurred in simulations with M2 (i.e. M2, M1+M2, M2+CB, RS+M2, M1+M2+CB, RS+M1+M2, RS+M2+CB & RS+M1+M2+CB). For each updraft speed, the ice enhancement profiles were identical for all simulations with M2, indicating that only M2 was active in the warmer cloud base simulations. All updraft speeds exhibited at least one ice enhancement peak, which shifted to higher simulation times from 99 to 111 to

250 118 min as the updraft speeds increased from 0.5 to 2 to 10 m s$^{-1}$ respectively. Simulations with updraft speeds of 0.5 m s$^{-1}$ showed further ice enhancement at the end of the simulation. For all updraft speeds, the maximum ice enhancements were similar, around 23 L$^{-1}$.

For simulations with a colder cloud base and updraft speeds of 0.5 m s$^{-1}$, ice enhancement occurred in simulations where RS, M2 or CB was present. Simulations with M2 without M1 (i.e. M2, RS+M2, M2+CB, RS+M2+CB) exhibited one ice

enhancement peak at 99 min with a maximum of 22 L$^{-1}$. There were interaction effects for combinations of SIP mechanisms. Simulations with M1 and M2 (i.e. M1+M2, RS+M1+M2, M1+M2+CB & RS+M1+M2+CB) had one ice enhancement peak

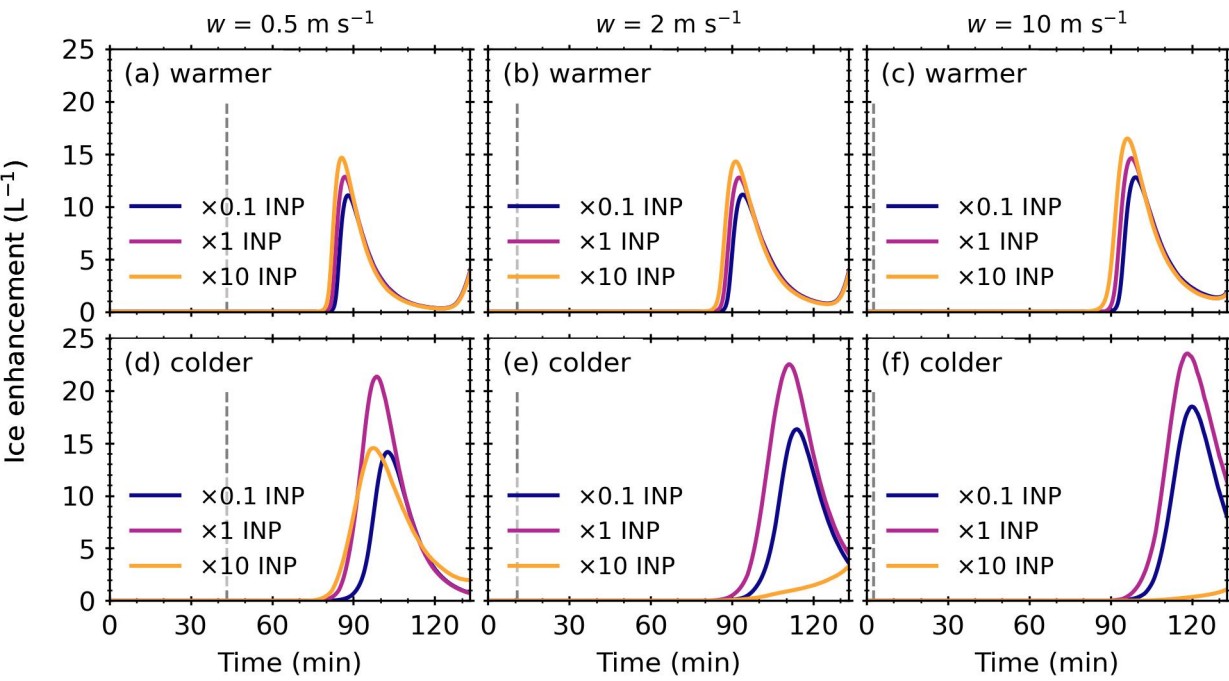

**Figure 3.** M2 ice enhancement against simulation time for three INP concentrations ($\times 0.1$, $\times 1$ & $\times 10$) for a shallower (1.3 km) cloud with a natural aerosol size distribution. Warmer refers to cloud base temperatures of 7 °C, and colder refers to cloud base temperatures of 0 °C.

with a maximum of 33 L$^{-1}$ at 96 min. Individually, the RS and CB SIP mechanisms were only slightly active with maximum ice enhancements between 0.1–0.2 L$^{-1}$, and the M1 SIP mechanism was not active.

For simulations with a colder cloud base and updraft speeds of 2 and 10 m s$^{-1}$, ice enhancement occurred in simulations
where M2 or CB was present. Simulations with M2 without M1 (i.e. M2, RS+M2, M2+CB, RS+M2+CB) exhibited one ice enhancement peak at 111 and 118 min for updraft speeds of 2 and 10 m s$^{-1}$ respectively. The maximum ice enhancement was approximately 23 L$^{-1}$, similar to the simulations with updraft speeds of 0.5 m s$^{-1}$. Again, there were interaction effects for combinations of SIP mechanisms. Simulations with M1 and M2 (i.e. M1+M2, RS+M1+M2, M1+M2+CB & RS+M1+M2+CB) had ice enhancements of $\sim$31 L$^{-1}$ for simulations with updraft speeds of 2 and 10 m s$^{-1}$ occurring at 109 and 118 min,
respectively. Simulations with CB except for M2 (i.e. CB, M1+CB, RS+CB, RS+M1+CB) were only slightly active with maximum ice enhancements between 0.4–1 L$^{-1}$, and the M1 and RS SIP mechanisms were not active.

### 4.1.2 Sensitivity test: Initial INP concentration

We used the DeMott et al. (2010) INP parameterisation scheme to initiate ice formation in the BMM. To test the sensitivity of the INP parameterisation scheme to ice enhancement, we multiplied the initial INP concentrations by 0.1 and 10. For reference,

Fig. S5 of the Supplement shows the control ICNCs for the INP concentrations over different cloud base temperatures and updraft speeds.

Figure 3 shows the ice enhancement for M2 simulations with INP concentrations of ×0.1, ×1 and ×10. The M2 ice enhancement profiles of the warmer cloud base simulations were similar for all the INP concentrations. For all updraft speeds, there was a small offset between the maximum ice enhancement peaks of approximately 1 min between ×0.1 and ×1 INP concentrations and a further 1 min between ×1 and ×10 INP concentrations. The maximum ice enhancement for all the INP concentrations was between 11–15 $L^{-1}$, with the ×10 INP concentration having the highest enhancement and the ×0.1 having the lowest enhancement.

For the colder cloud base with updraft speeds of 0.5 m s$^{-1}$, the ×1 INP and ×10 concentrations peaked at similar times and the ×0.1 INP concentration peaked last. The ×1 INP concentration had the highest ice enhancement. The ×0.1 and ×10 INP concentrations had similar maximum ice enhancements, approximately 10 $L^{-1}$ less than the ×1 INP concentration. However, compared to the ×1 INP concentration, the ×10 INP concentration had a broader ice enhancement profile. For simulations with updraft speeds of 2 and 10 m s$^{-1}$, the ×1 INP concentration had the highest ice enhancement, followed by the ×0.1 INP concentration. Very little ice enhancement occurred in the simulations with the ×10 INP concentration.

In addition to ice enhancement due to M2, ice enhancement occurred due to CB and M1+M2. The M1+M2 simulations followed similar ice enhancement trends to the M2 simulations (see Fig. S6 of the Supplement). However, the CB simulations followed different ice enhancement trends (see Fig. S7 of the Supplement). There was a non-linear effect on ice enhancement due to INP concentration in simulations with a colder cloud base. For example, in simulations with updraft speeds of 0.5 m s$^{-1}$, the ×10 INP concentration had the highest ice enhancement, followed by the ×1 INP then the ×10 INP concentrations. However, in simulations with updraft speeds of 2 m s$^{-1}$, the ×1 INP concentration had the highest ice enhancement with little to no ice enhancement from the ×10 and ×0.1 INP concentrations. In simulations with updraft speeds of 10 m s$^{-1}$, the ×10 INP concentration had the highest ice enhancement, followed by the ×1 and ×0.1 INP concentrations.

### 4.1.3 Sensitivity test: M2 freezing fraction ($\Phi$)

The M2 parameterisation given in Eq. 9 requires the fraction of secondary drops that will freeze, denoted as $\Phi$. We used a value of 0.3, based on our recent laboratory measurements (James et al., 2021). To test the sensitivity of ice enhancement to $\Phi$ during M2 simulations, we ran simulations for warmer and colder cloud bases with natural aerosol size distribution and updraft speeds of 2 m s$^{-1}$ using the following $\Phi$ values: 0.5, 0.1, 0.01, 0.001 & 0.0001. The results are plotted in Figure 4.

For the warmer cloud base temperatures, as the value of $\Phi$ decreased from 0.5 to 0.3, the maximum ice enhancement decreased by $\sim$8 $L^{-1}$. When the value of $\Phi$ decreased from 0.3 to 0.1, the maximum ice enhancement decreased by $\sim$9 $L^{-1}$. Where $\Phi$ had values $\leq 0.1$, very little to no ice enhancement was observed, between 0–4 $L^{-1}$. Compared to lower values of $\Phi$, higher values of $\Phi$ had earlier maximum ice enhancement peaks. For example, ice enhancement occurred 1 min earlier for $\Phi$ values of 0.5 compared to $\Phi$ values of 0.3 and 5 min earlier for $\Phi$ values of 0.5 compared to $\Phi$ values of 0.1.

For the colder cloud base, decreasing the value of $\Phi$ from 0.5 to 0.3 reduced the maximum ice enhancement by 10 $L^{-1}$. When the values of $\Phi$ decreased from 0.3 to 0.1, the maximum ice enhancement decreased by $\sim$13 $L^{-1}$. When $\Phi$ was between 0.1

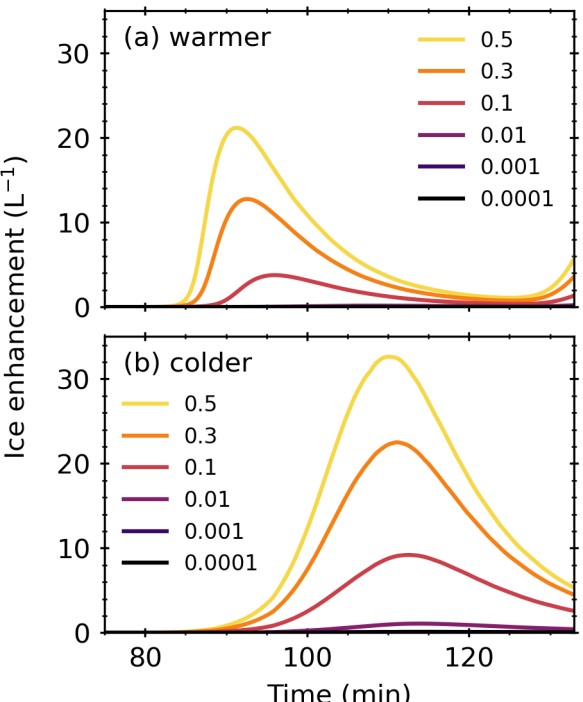

**Figure 4.** M2 ice enhancement against simulation time for five $\Phi$ values (0.5, 0.3, 0.1, 0.01, 0.001 and 0.0001) for a shallower (1.3 km) cloud with natural aerosol size distributions and updraft speed of 2 m s$^{-1}$. Warmer refers to cloud base temperatures of 7 °C, and colder refers to cloud base temperatures of 0 °C.

and 0.01, the maximum ice enhancement decreased by 9 L$^{-1}$. There was very little to no ice enhancement when $\Phi$ had values
of 0.001 or below. Similar to the warmer cloud base simulations, lower values of $\Phi$ delayed the maximum ice enhancement for
the colder cloud bases.

### 4.1.4  Sensitivity test: M2 DE$_{Crit}$

The M2 parameterisation given in Eq. 9 requires the onset of splashing, denoted as DE$_{crit}$. We used the value of 0.2 given in
Phillips et al. (2018) based on laboratory data of drops colliding on roughened copper hemispheres from Levin et al. (1971).
To test the sensitivity of ice enhancement to DE$_{crit}$ during M2 simulations, we ran simulations for warmer and colder cloud
bases with natural aerosol size distribution and updraft speeds of 2 m s$^{-1}$ using the following DE$_{crit}$ values: 0.2, 3 & 6. The
results are plotted in Figure 5.

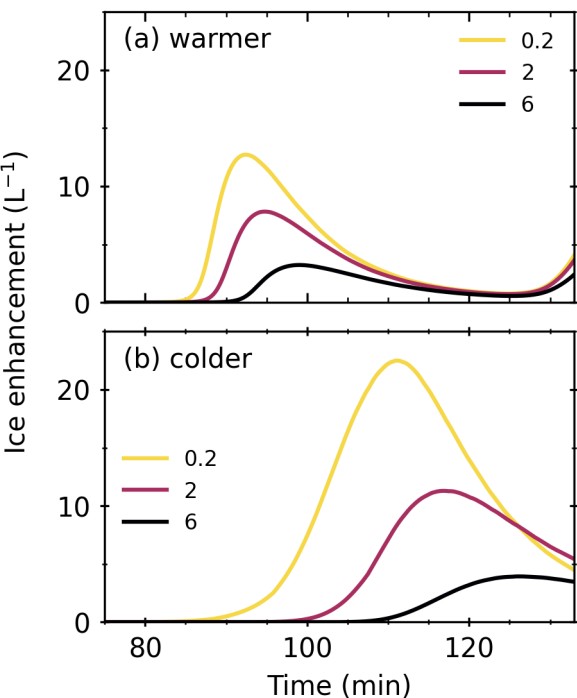

**Figure 5.** M2 ice enhancement against simulation time for three $DE_{crit}$ values (0.2, 3 and 6) for a shallower (1.3 km) cloud with natural aerosol size distributions and updraft speed of 2 m s$^{-1}$. Warmer refers to cloud base temperatures of 7 °C, and colder refers to cloud base temperatures of 0 °C.

For the warmer cloud base temperatures, as the value of $DE_{crit}$ increased from 0.2 to 3, the maximum ice enhancement decreased by $\sim$6 L$^{-1}$. When the value of $\Phi$ increased from 3 to 6, the maximum ice enhancement decreased by $\sim$4 L$^{-1}$.

Compared to higher values of $DE_{crit}$, lower values of $DE_{crit}$ had earlier maximum ice enhancement peaks. For example, ice enhancement occurred $\sim$2 min earlier for $DE_{crit}$ values of 0.2 compared to $DE_{crit}$ values of 3 and 6 min earlier for $DE_{crit}$ values of 0.2 compared to $DE_{crit}$ values of 6.

For the colder cloud base, increasing the value of $DE_{crit}$ from 0.2 to 3 reduced the maximum ice enhancement by 11 L$^{-1}$. When the values of $DE_{crit}$ increased from 3 to 6, the maximum ice enhancement decreased by $\sim$7 L$^{-1}$. Similar to the warmer

cloud base simulations, lower values of $DE_{crit}$ delayed the maximum ice enhancement for the colder cloud bases.

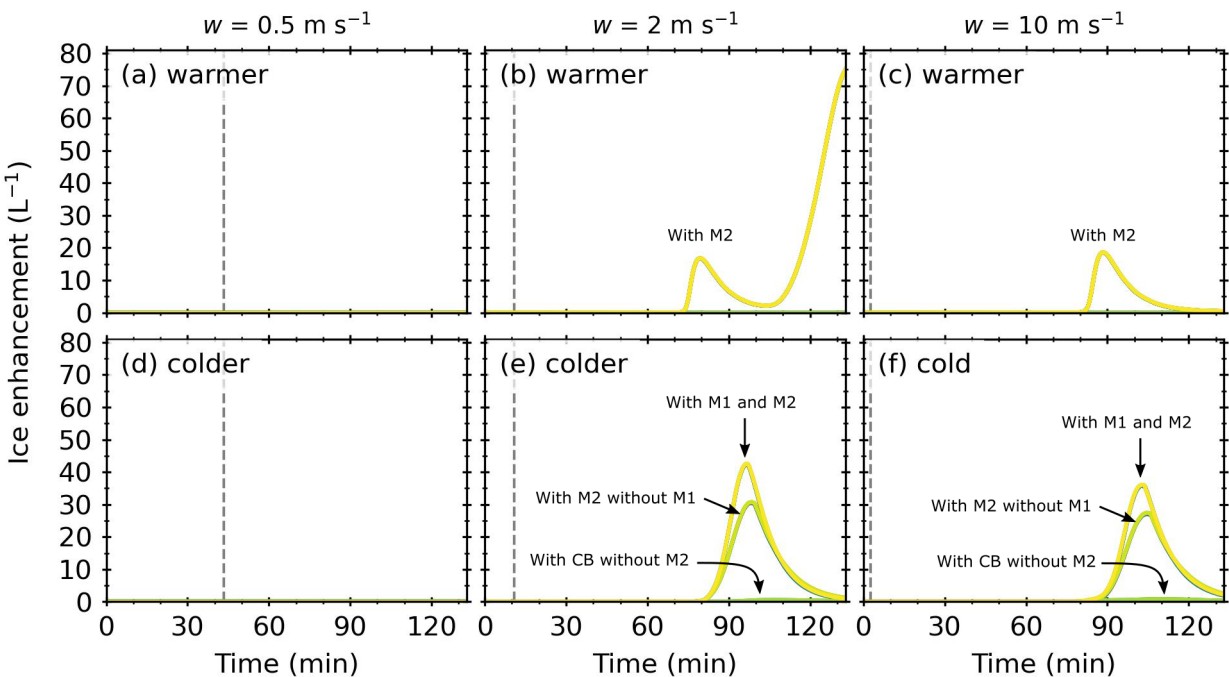

**Figure 6.** Ice enhancement against simulation time for a shallower (1.3 km) cloud with a near-city aerosol size distribution. Warmer refers to cloud base temperatures of 7 °C, and colder refers to cloud base temperatures of 0 °C. The grey dashed lines indicated the threshold time where the updraft was turned off. Plots are annotated to indicate the processes that were active.

## 4.2 Shallower cloud: Near-city aerosol size distribution

### 4.2.1 Sensitivity test: Cloud base temperature and updraft speed

Similar to the natural aerosol size distribution, we performed simulations using a near-city aerosol size distribution for two cloud base temperatures, 7 and 0 °C, and three updraft speeds, 0.5, 2 and 10 m s$^{-1}$. The SIP mechanisms, RS, M1, M2 and CB, were investigated individually and for all possible combinations. We also performed a control simulation with no SIP mechanisms.

For the warmer cloud base simulations, the maximum control ICNCs were $3 \times 10^{-4}$ L$^{-1}$, $2 \times 10^{-3}$ L$^{-1}$ and 0.01 L$^{-1}$ for updraft speeds of 0.5, 2 and 10 m s$^{-1}$, respectively. For the colder cloud base simulations, the maximum control ICNCs were 0.3, 1 and 1.5 L$^{-1}$ for updraft speeds of 0.5, 2 and 10 m s$^{-1}$, respectively. For reference, Fig. S8 of the Supplement shows the ICNC of the control simulations for a near-city aerosol size distribution.

For both cloud base temperatures, the maximum CDNC for simulations with updraft speeds of 0.5, 2 and 10 m s$^{-1}$ was 100, 900 and 2100 cm$^{-3}$ respectively. These maxima occurred within the first few minutes of the simulations. For both cloud base

temperatures, in the simulations with updraft speeds of 0.5 m s$^{-1}$, the CDNC remained constant throughout the simulation; hence there was no active collision-coalescence process (and therefore no strongly active secondary ice production). In simulations with a warmer cloud base temperature and updraft speeds of 10 m s$^{-1}$, the CDNC decreased sharply, almost to zero, for all simulations around 80 min. For simulations with a warmer cloud base temperature and updraft speeds of 2 m s$^{-1}$, this decrease occurred around 90 min in simulations with M2. For simulations without M2, this occurred around 95 min. Sharp drops in the CDNC, almost zero, also happened in simulations with a colder cloud base and updraft speeds of 2 and 10 m s$^{-1}$ around 90 and 80 min, respectively. For reference, Fig. S9 shows the CDNCs for all SIP mechanism simulations with a natural aerosol size distribution.

Figure 6 shows the ice enhancement as a function of simulation time for a shallower (1.3 km) convective cloud with a near-city aerosol size distribution. The vertical dashed line represents $t_w$ and is dependent on the updraft speed. For simulations with updraft speeds of 0.5 m s$^{-1}$, no significant ice enhancement was observed for any SIP mechanism or combination due to the lack of a strong collision-coalescence process for this case. For simulations with a warmer cloud base and updraft speeds of 2 and 10 m s$^{-1}$, ice enhancement occurred in simulations with M2 (i.e. M2, M1+M2, M2+CB, RS+M2, M1+M2+CB, RS+M1+M2, RS+M2+CB & RS+M1+M2+CB). For both updraft speeds, the ice enhancement profiles were identical for all simulations with M2, indicating that only M2 was active. Both updraft speeds exhibited one ice enhancement peak, which shifted to higher simulation times as the updraft speed increased from 79 to 88 min for updraft speeds of 2 and 10 m s$^{-1}$ respectively. A further ice enhancement occurred in simulations with updraft speeds of 2 m s$^{-1}$ towards the end of the simulation. For both updraft speeds, the maximum of the ice enhancement peak was similar, around 17 L$^{-1}$. However, the ice enhancement, which occurred towards the end of the simulations with updraft speeds of 2 m s$^{-1}$, was approximately 50 L$^{-1}$ more.

For simulations with a colder cloud base and updraft speeds of 2 and 10 m s$^{-1}$, ice enhancement occurred in simulations with M2 or CB. Simulations with M2 without M1 (i.e. M2, M2+RS, M2+CB & RS+M2+CB) exhibited one ice enhancement peak at 98 and 104 min for updraft speeds of 2 and 10 m s$^{-1}$ respectively. The maximum ice enhancement was similar for both updraft speeds, around 29 L$^{-1}$. There was an interaction effect in simulations with M1 and M2 (i.e. M1+M2, RS+M1+M2, M1+M2+CB, RS+M1+M2+CB). Compared to the M2 simulations, these simulations had higher ice enhancements, 44 and 38 L$^{-1}$ for simulations with updraft speeds of 2 and 10 m s$^{-1}$ respectively, which occurred at slightly earlier at 97 and 103 min. Simulations with CB without M2 (i.e. CB, M1+CB, RS+CB, RS+M1+CB) were only slightly active with maximum ice enhancements between 0.4–1 L$^{-1}$, and the M1 and RS SIP mechanisms were not active.

### 4.2.2 Sensitivity test: INP concentration

Figure 7 shows the ice enhancement for M2 simulations with INP concentrations of ×0.1, ×1 and ×10. For reference, Fig. S10 of the Supplements shows the ICNCs from the control simulations. No ice enhancement occurred in simulations with updraft speeds of 0.5 m s$^{-1}$. For warmer cloud base simulations with updraft speeds of 2 and 10 m s$^{-1}$, the M2 ice enhancement profiles were similar for all the INP concentrations, exhibiting one ice enhancement peak and, for the simulations with updraft speeds of 2 m s$^{-1}$ a further increase towards the end of the simulation. There was a small offset between the maximum ice

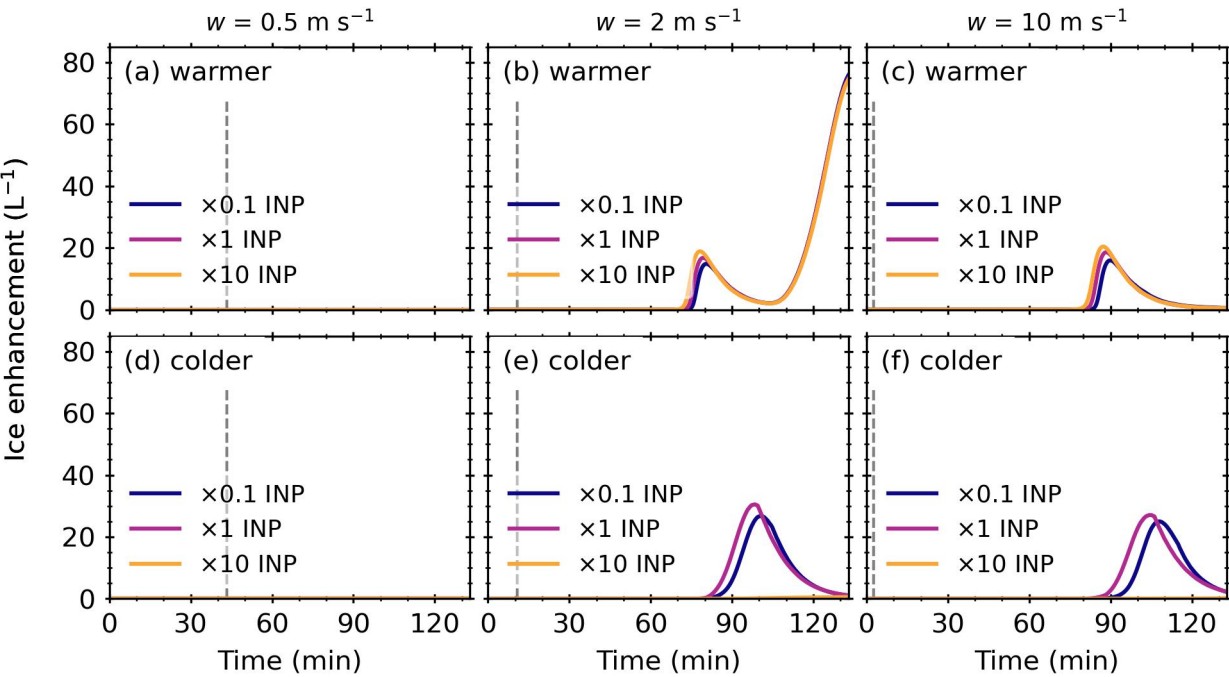

**Figure 7.** M2 ice enhancement against simulation time for three INP concentrations (×0.1, ×1 & ×10) for a shallower (1.3 km) cloud with a near-city aerosol size distribution. Warmer refers to cloud base temperatures of 7 °C, and colder refers to cloud base temperatures of 0 °C.

enhancement peaks of approximately 1 min between ×0.1 and ×1 INP concentrations and a further 1 min between ×1 and ×10 INP concentrations. The maximum ice enhancement of the peak between 80–90 min was between 15–20 $L^{-1}$, with the ×10 INP concentration having the highest enhancement and the ×0.1 having the lowest enhancement. For the colder cloud base (273 K) with updraft speeds of 2 and 10 m $s^{-1}$, the ×1 INP concentration had the highest ice enhancement, followed by the ×0.1 INP concentration. Very little to no ice enhancement occurred in the simulation with ×10 INP concentration.

In addition to M2 ice enhancement, ice enhancement occurred due to CB and M1+M2. The M1+M2 simulations followed similar ice enhancement trends to the M2 simulations (see Fig. S11 of the Supplement). The CB simulations followed different trends (see Fig. S12 of the Supplement). For the warmer cloud bases, a very slight ice enhancement due to CB occurred in the simulation with updraft speeds of 10 m $s^{-1}$ and ×10 INP concentration. For the colder cloud base, slight ice enhancements due to CB occurred in simulations with updraft speeds of 2 and 10 m $s^{-1}$. Very little ice enhancement occurred in the simulations with updraft speeds of 0.5 m $s^{-1}$. In simulations with updraft speeds of 2 m $s^{-1}$, the ×1 INP concentration had the highest ice enhancement with little to no ice enhancement from the ×0.1 and ×10 INP concentrations. In simulations with updraft speeds of 10 m $s^{-1}$, the ×10 INP concentration had the highest ice enhancement, peaking around 75 min. The ×1 INP concentration

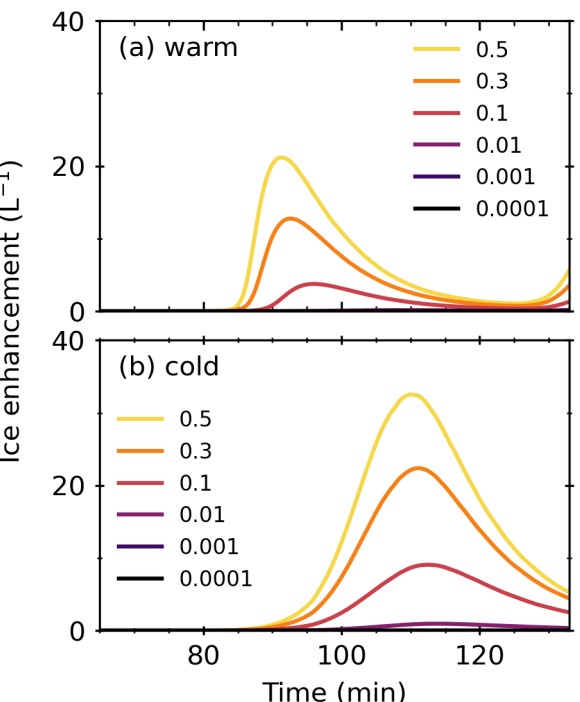

**Figure 8.** M2 ice enhancement against simulation time for five $\Phi$ values (0.5, 0.3, 0.1, 0.01, 0.001 and 0.0001) for a shallower (1.3 km) cloud with near-city aerosol size distributions and updraft speed of 2 m s$^{-1}$. Warmer refers to cloud base temperatures of 7 °C, and colder refers to cloud base temperatures of 0 °C.

peaked later, around 110 min, with little to no ice enhancement from the ×0.1 INP concentrations. Hence, the clouds exhibited some sensitivity to INP concentrations when INP concentrations were high.

### 4.2.3 Sensitivity test: M2 $\Phi$ parameter

Figure 8 shows the M2 ice enhancement against simulation time for six $\Phi$ values, 0.5, 0.3, 0.1, 0.01, 0.001 and 0.0001, for shallower clouds with a near-city aerosol size distribution. Similar to simulations with a natural aerosol size distribution, little

to no ice enhancement occurred when the $\Phi$ value decreased below 0.1 for both cloud base temperatures. In general, where ice enhancement occurred ($\Phi \geq 0.1$), higher values of $\Phi$ had earlier maximum ice enhancement peaks compared to lower values of $\Phi$.

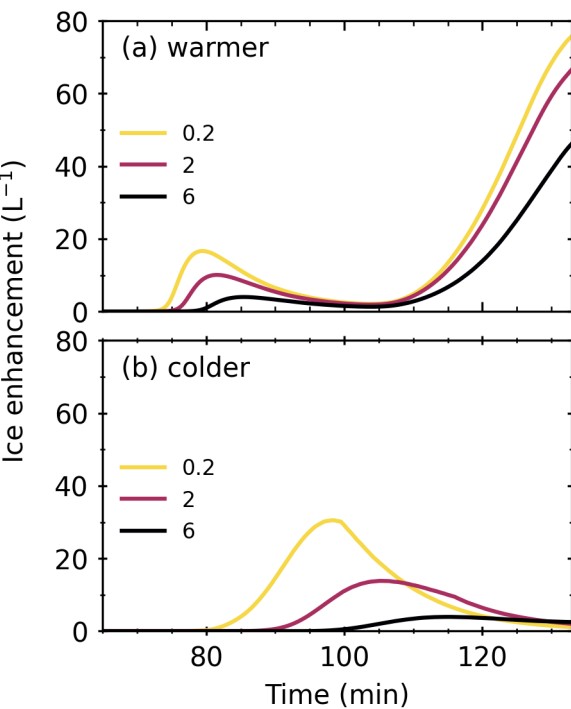

**Figure 9.** M2 ice enhancement against simulation time for three DE$_{crit}$ values (0.2, 3 and 6) for a shallower (1.3 km) cloud with near-city aerosol size distributions and updraft speed of 2 m s$^{-1}$. Warmer refers to cloud base temperatures of 7 °C, and colder refers to cloud base temperatures of 0 °C.

### 4.2.4 Sensitivity test: M2 DE$_{crit}$

Figure 9 shows the M2 ice enhancement against simulation time for three DE$_{crit}$ values, 0.2, 3 and 6, for shallower clouds with
a near-city aerosol size distribution. In general, lower values of DE$_{crit}$ had earlier maximum ice enhancement peaks compared to higher values of DE$_{crit}$. The maximum ice enhancement peaks were greater for lower DE$_{crit}$ values.

### 4.3 Deeper cloud: natural aerosol size distribution

### 4.3.1 Sensitivity test: Cloud base temperature and updraft speed

We performed simulations using a natural aerosol size distribution for two cloud base temperatures, 7 and 0 °C, and three
updraft speeds, 0.5, 2 and 10 m s$^{-1}$. Similar to the shallower clouds presented in Section 4.1 & 4.2, we investigated the SIP

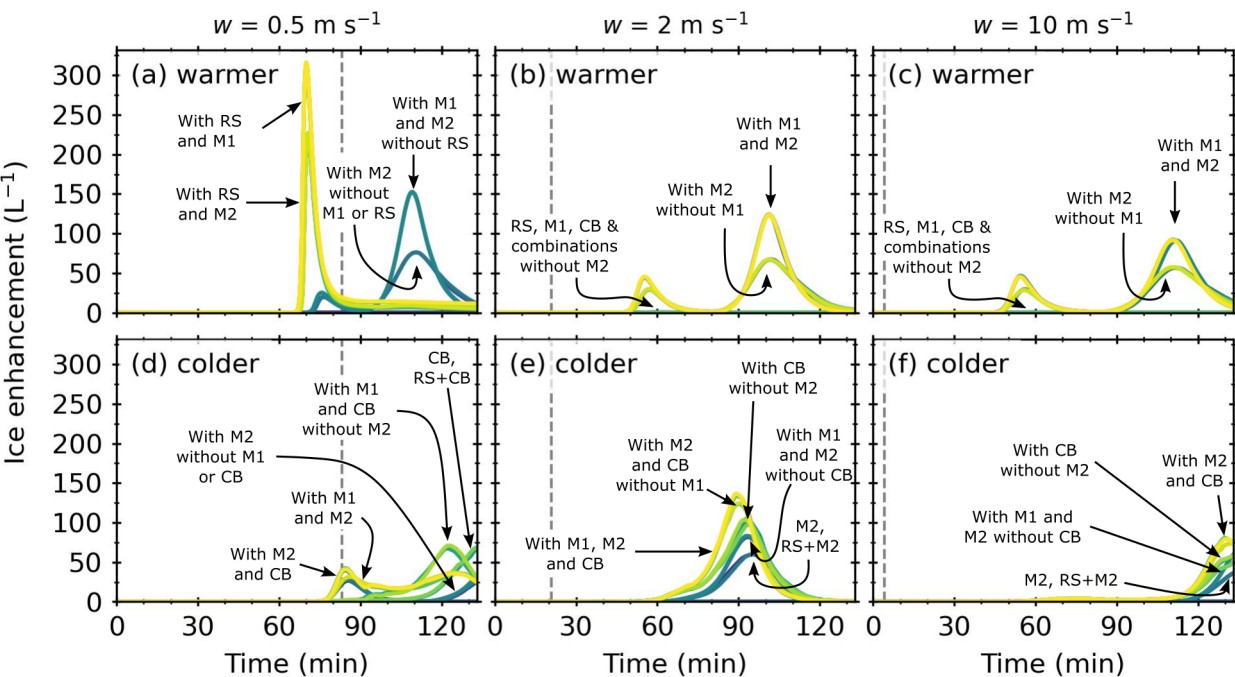

**Figure 10.** Ice enhancement against simulation time for a deeper (2.4 km) cloud with a natural aerosol size distribution. Warmer refers to cloud base temperatures of 7 °C, and colder refers to cloud base temperatures of 0 °C. The grey dashed lines indicated the threshold time where the updraft was turned off. Plots are annotated to indicate the processes that were active.

mechanisms RS, CB, M1 and M2 individually and for all possible combinations. We also performed a control simulation with no SIP mechanisms.

For reference, Fig. S13 of the Supplement shows the ICNC of the control simulations for natural aerosol size distributions. The warmer cloud base temperature simulations had maximum control ICNCs of 0.7, 1 and 2 $L^{-1}$ for updraft speeds of 0.5, 2 and 10 m $s^{-1}$, respectively. The colder cloud base temperature simulations had maximum control ICNCs of 3, 5 and 8 $L^{-1}$ for updraft speeds of 0.5, 2 and 10 m $s^{-1}$, respectively. Fig. S14 of the Supplement shows the CDNCs for all simulations. The maximum CDNCs for simulations with updraft speeds of 0.5, 2 and 10 m $s^{-1}$ were approximately 420, 1000 and 1300 $cm^{-1}$.

Figure 10 shows the ice enhancement as a function of simulation time for a deeper (2.4 km) convective cloud with a natural aerosol size distribution. The initial updraft speed remained constant until a threshold time ($t_w$), dependent on the updraft speed, after which it was set to zero to simulate a cloud at the desired depth. In Fig. 10 $t_w$ is represented by a dashed line at approximately 83, 21 and 4 min for updraft speeds of 0.5, 2 and 10 m $s^{-1}$, respectively.

For simulations with a warmer cloud base and updraft speeds of 0.5 m $s^{-1}$, ice enhancement occurred in simulations with RS or M2 active. Simulations with RS and M2 (i.e. RS+RS, RS+M1+M2, RS+M2+CB, RS+M1+M2+CB) exhibited one

ice enhancement peak at 71 min with a maximum ice enhancement of 223 $L^{-1}$. Simulations with RS and M1 (i.e. RS+M1, RS+M1+CB, RS+M1+M2, RS+M1+M2+CB) also had one ice enhancement peak at 71 min with a maximum ice enhancement of 316 $L^{-1}$. Simulations with M2 without RS or M1 exhibited two ice enhancement peaks at 77 and 111 min with maximums of 20 and 76 $L^{-1}$. Simulations with M1 and M2 without RS (i.e. M1+M2, M1+M2+CB) also two ice enhancement peaks at 76 and 109 min with maximums of 25 and 152 $L^{-1}$. Individually, the M1 and CB SIP mechanisms were not active.

For simulations with a warmer cloud base and updraft speeds of 2 and 10 m s$^{-1}$, ice enhancement occurred in simulations with M2 and CB and all combinations of these SIP mechanisms. Simulations with M2 without M1+M2 or M1+M2+RS (i.e. M2, RS+M2, M2+CB, RS+M2+CB, M1+M2+CB, RS+M1+M2) exhibited two ice enhancement peaks at 57 and 102 min for updraft speeds of 2 m s$^{-1}$ and 56 and 112 min for updraft speeds of 10 m s$^{-1}$. The maximum ice enhancement for the first peak was 29 $L^{-1}$ for simulations with updraft speeds of 2 and 10 m s$^{-1}$ respectively, and 68 and 57 $L^{-1}$ for the second ice enhancement peak. A higher ice enhancement occurred in the M1+M2 and RS+M1+M2 simulations. The first peaks had maximums of 44 and 46 $L^{-1}$ at 55 min. The second peaks had maximums of 123 and 91 $L^{-1}$ at 101 and 112 min for updraft speeds of 2 and 10 m s$^{-1}$. On their own, the RS, M1 and CB SIP mechanisms were only slightly active with maximum ice enhancements between 0.1–0.7 $L^{-1}$.

For simulations with a colder cloud base and updraft speeds of 0.5 m s$^{-1}$, ice enhancement occurred in simulations with M2 or CB active. Simulations with M2 without M1 or CB (i.e. M2, RS+M2) had one ice enhancement peak at 86 min with a maximum of 27 $L^{-1}$, and a further ice enhancement towards the end of the simulation. Simulations with M1 and M2 (i.e. M1+M2, RS+M1+M2, M1+M2+CB, RS+M1+M2+CB) had similar profiles but with higher maximum ice enhancements of 41 $L^{-1}$. Simulations with M2 and CB (i.e. M2+CB, RS+M2+CB, M1+M2+CB, RS+M1+M2+CB) had two overlapping peaks at 87 and 125 min with ice enhancements of 31 and 40 $L^{-1}$. Simulations with CB without M1 or M2 (i.e. CB, RS+CB) exhibited ice enhancement towards the end of the simulation with maximum values of ~67 $L^{-1}$. Simulations with CB and M1 without M2 (i.e. M1+CB, RS+M1+CB) exhibited an ice enhancement peak towards the end of the simulation at 122 min with a maximum ice enhancement of 67 $L^{-1}$. Individually, the M1 and RS SIP mechanisms were not active.

For simulations with a colder cloud base and updraft speeds of 2 m s$^{-1}$, ice enhancement occurred in simulations with M2 or CB active. All active SIP mechanisms exhibited one ice enhancement peak. The M2 and RS+M2 simulations had a maximum ice enhancement peak at 95 min of 60 $L^{-1}$. Simulations with M1 and M2 without CB (i.e. M1+M2, RS+M1+M2) had slightly higher ice enhancements of 82 $L^{-1}$. Simulations with CB without M2 (i.e. CB, M1+CB, RS+CB, RS+M1+CB) had a maximum ice enhancement peak of 103 $L^{-1}$ at 93 min. Simulations with M1, M2 and CB (i.e. M1+M2+CB, RS+M1+M2+CB) had an ice enhancement peak at 89 min of 136 $L^{-1}$. Simulations with M2 and CB without M1 (i.e. M2+CB, RS+M2+CB) had an ice enhancement peak of 124 $L^{-1}$ at 90 min. Individually, the RS and M1 SIP mechanisms were not active.

For simulations with a colder cloud base and updraft speed of 10 m s$^{-1}$, an ice enhancement occurred towards the end of the simulation where M2 or CB was active. The M2 and RS+M2 simulations had a maximum ice enhancement of 40 $L^{-1}$ at the end of the simulation. Simulations with M1 and M2 without CB (i.e. M1+M2, RS+M1+M2) had a slightly higher ice enhancement of 61 $L^{-1}$ at the end of the simulation. Simulations with CB without M2 (i.e. CB, RS+CB, M1+CB, RS+M1+CB) had an ice

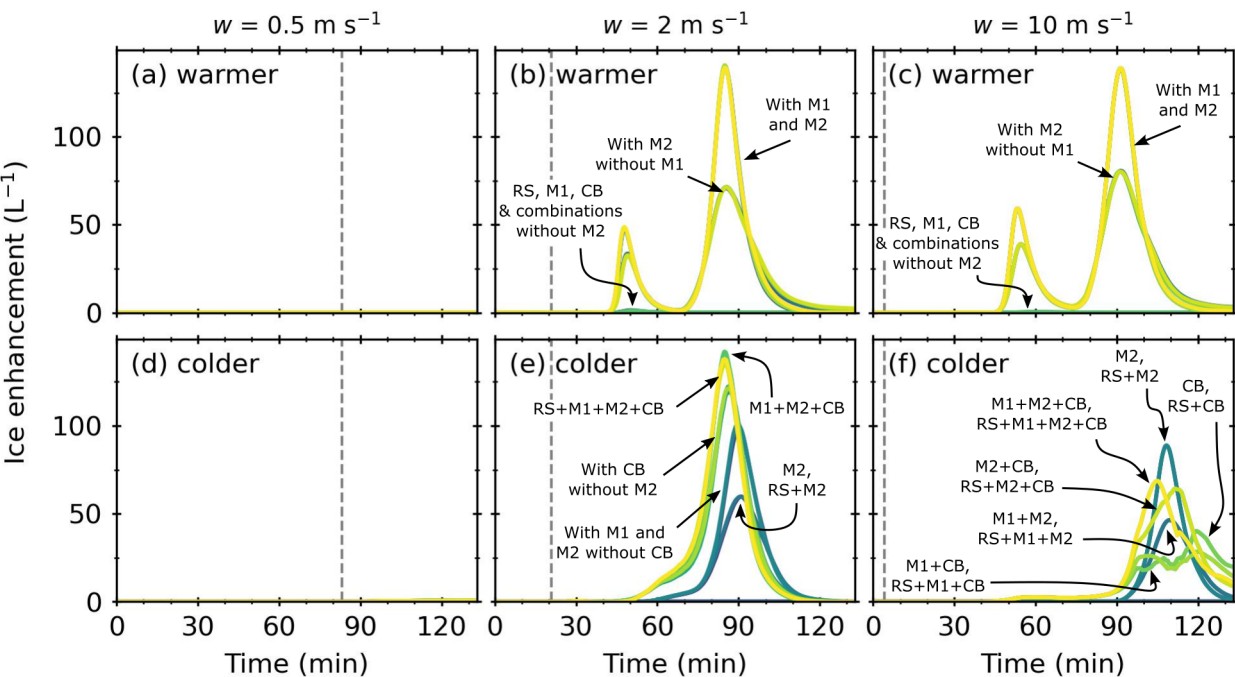

**Figure 11.** Ice enhancement against simulation time for a deeper (2.4 km) cloud with a near-city aerosol size distribution. Warmer refers to cloud base temperatures of 7 °C, and colder refers to cloud base temperatures of 0 °C. The grey dashed lines indicated the threshold time where the updraft was turned off. Plots are annotated to indicate the processes that were active.

enhancement of 51 $L^{-1}$ at the end of the simulation. Simulations with M2 and CB (i.e. M2+CB, RS+M2+CB, M1+M2+CB, RS+M1+M2+CB) had an ice enhancement peak at 131 min of 80 $L-1$.

## 445 4.4 Deeper cloud: near-city aerosol size distribution

### 4.4.1 Sensitivity test: Cloud base temperature and updraft speed

For reference, Fig. S15 of the Supplement shows the ICNC of the control simulations for a near-city aerosol size distribution. The warmer cloud base temperature simulations had maximum control ICNCs of 0.3, 1 and 2 $L^{-1}$ for updraft speeds of 0.5, 2 and 10 m s$^{-1}$, respectively. The colder cloud base temperature simulations had maximum control ICNCs of 7, 10 and 12 $L^{-1}$

for updraft speeds of 0.5, 2 and 10 m s$^{-1}$, respectively. Fig. S16 of the Supplement shows the CDNCs for all simulations. The maximum CDNCs for simulations with updraft speeds of 0.5, 2 and 10 m s$^{-1}$ were approximately 100, 1200 and 2400 cm$^{-1}$.

For simulations with updraft speeds of 0.5 m s$^{-1}$, no significant ice enhancement was observed for any SIP mechanism or combination. For simulations with a warmer cloud base and updraft speeds of 2 and 10 m s$^{-1}$, ice enhancement occurred

in simulations with M2 or CB active. Simulations with M2 without M1 (i.e. M2, RS+M2, M2+CB, RS+M2+CB) exhibited two ice enhancement peaks. The first peaks occurred at 49 and 55 min for simulations with updraft speeds 2 and 10 m s$^{-1}$ respectively, and the second peaks at 85 and 91 min. The maximum ice enhancement for the first peak was 33 and 39 L$^{-1}$ for simulations with updraft speeds of 2 and 10 m s$^{-1}$ respectively, and 71 and 80 L$^{-1}$ for the second ice enhancement peak. For M2+M1 and M2+M1+RS simulations, higher ice enhancements occurred. Maximums of 47 and 139 L$^{-1}$ for simulations with updraft speeds of 2 m s$^{-1}$ and 59 and 138 L$^{-1}$ for updraft speeds of 10 m s$^{-1}$. Individually, the CB, RS, and M1 SIP mechanisms were only slightly active.

For simulations with a colder cloud base and updraft speeds of 2 m s$^{-1}$, ice enhancement occurred in simulations with M2 or CB. All active SIP mechanisms exhibited one ice enhancement peak. The M1+M2+CB simulation had a maximum ice enhancement of 142 L$^{-1}$ at 85 min. The M2+CB, RS+M2+CB and RS+M1+M2+CB simulations had maximum ice enhancements of 136 L$^{-1}$. Simulations with M1 and M2 without CB (i.e. M1+M2, RS+M1+M2) had maximum ice enhancements of 101 L$^{-1}$ at 90 min. The M2 and RS+M2 simulations had a maximum ice enhancement peak at 91 min of 60 L$^{-1}$. Individually, M1 and RS SIP mechanisms were not active.

For simulations with a colder cloud base and updraft speeds of 10 m s$^{-1}$, ice enhancement occurred in simulations with M2 or CB. Individually, M1 and RS SIP mechanisms were not active. The M2 and RS+M2 simulations exhibited one ice enhancement peak at 110 min of 46 L$^{-1}$. The M1+M2 and RS+M1+M2 simulations had ice enhancements of 88 L$^{-1}$ at 108 min. The M2+CB and RS+M2+CB simulations had an ice enhancement of 64 L$^{-1}$ at 112 min. The RS+M2+CB and RS+M1+M2+CB simulations had an ice enhancement peak of 69 L$^{-1}$ at 105 min. The CB and RS+CB simulations a maximum ice enhancement peak of 40 L$^{-1}$ at 120 min. The M1+CB and RS+M1+CB simulation had two overlapping peaks with similar maximum ice enhancements of 28 L$^{-1}$ between 100–120 min.

## 5 Discussion

In our simulations, ice enhancement occurs after the collision-coalescence process, which takes time to initiate due to drop growth, usually within the latter half of the simulations ( > 60 min). The lifetime of shallow convective clouds is variable. For example, Öktem and Romps (2021) showed that shallow cumulus clouds forming over the Southern Great Plains in the USA over a observational period of three years had lifetimes from as little as 30 min to over 9 hours, with shorter duration clouds occurring in the spring. There have been observations of rapid ice enhancement within shallow convective clouds(e.g. Rangno and Hobbs, 1991; Hobbs and Rangno, 1990). Collision-coalescence is key to initiate ice enhancement within our simulations and there are several processes which may accelerate collision-coalescence which are not modelled. For example, entrainment of dry air may cause inhomogeneous mixing and preferential growth of larger drops during ascent (e.g. Baker et al., 1980; Telford and Chai, 1980) or turbulence (e.g. Pinsky et al., 2008; Grabowski and Wang, 2013; Chen et al., 2018) See Morrison et al. (2022) for more discussion of these processes.

Table 4 shows a summary of active SIP mechanisms and combinations in the idealised shallow convective clouds investigated. Overall, M2 is a significant contributor to ice enhancement, individually and in combination with other SIP mechanisms,

**Table 4.** Summary of active SIP mechanisms. ↑ indicates a SIP mechanism or SIP mechanism combination where significant ice enhancement occurred.

| Cloud | Updraft speed (m s$^{-1}$) | Active SIP mechanism |
|---|---|---|
| Shallow | 0.5 | M2 ↑ |
| Natural | 2 | M2 ↑ |
| Warm | 10 | M2 ↑ |
| Shallow | 0.5 | RS, M2 ↑, CB, M1+M2 ↑ |
| Natural | 2 | M2 ↑, CB, M1+M2 ↑ |
| Cold | 10 | M2 ↑, CB, M1+M2 ↑ |
| Shallow | 0.5 | – |
| Near-city | 2 | M2 ↑ |
| Warm | 10 | M2 ↑ |
| Shallow | 0.5 | – |
| Near-city | 2 | CB, M2 ↑, M1+M2 ↑ |
| Cold | 10 | CB, M2 ↑, M1+M2 ↑ |
| Deep | 0.5 | RS ↑, M2↑, RS+M1 ↑, RS+M2 ↑, M1+M2 ↑ |
| Natural | 2 | RS, M1, M2 ↑, CB, M1+M2 ↑ |
| Warm | 10 | RS, CB, M1, M2 ↑, M1+M2 ↑ |
| Deep | 0.5 | M2 ↑, CB ↑, M1+M2 ↑, M1+CB ↑ M2+CB ↑ |
| Natural | 2 | M2 ↑, CB ↑, M1+M2 ↑, M2+CB ↑, M1+M2+CB ↑ |
| Cold | 10 | CB ↑, M2 ↑, M1+M2 ↑, M2+CB ↑ |
| Deep | 0.5 | – |
| Near-city | 2 | RS, M1, M2 ↑, CB, M1+M2 ↑ |
| Warm | 10 | RS, M1, M2 ↑, CB, M1+M2 ↑ |
| Deep | 0.5 | – |
| Near-city | 2 | M2 ↑, CB ↑, M1+M2 ↑, CB+M2 ↑, RS+M1+M2 ↑ |
| Cold | 10 | M2 ↑, CB ↑, M1+M2 ↑, M1+CB↑, M2+CB ↑, M1+M2+CB↑ |

for most idealised shallow convective clouds. The exception being clouds with a near-city aerosol size distribution with updraft speeds of 0.5 m s$^{-1}$. There were also several clouds where other SIP mechanisms significantly contributed to ice enhancement. Rime-splintering in the deeper cloud with a warmer cloud base with a natural aerosol size distribution and updraft speeds of

0.5 m s$^{-1}$, and ice-ice collisional breakup for some deeper clouds with colder cloud bases.

     To understand the prevalence of the M2 SIP mechanism, we first consider why the other SIP mechanisms may not have been as active. For RS to be active, according to the RS parameterisation scheme used in the BMM, the cloud must have temperatures between -2.5 °C and -7.5 °C. This rules out the shallower clouds with warmer cloud bases as the cloud tops are too warm, with temperatures of -1 °C. The temperature region is only one requirement of the RS mechanism. In addition, the cloud drops

must been sufficiently large and have a broad size distribution. Drop size distributions are broadest in simulations with lower updraft speeds (see Figs. S17–S20 of the Supplement). Compared to deeper clouds, shallower clouds have a narrower drop size distribution; hence we only observe ice enhancement due to RS in the shallower idealised clouds with a natural aerosol size distribution and colder cloud base with updraft speeds of 0.5 m s$^{-1}$. While RS was active for the deeper clouds with warmer cloud bases for all updraft speeds due to the overall broader drop size distribution within the RS temperature region, only the

lower updraft speeds of 0.5 m s$^{-1}$ had significant ice enhancement due to RS. This is due to the increased residence time in the RS temperature region for the simulations with the lowest updraft speed.

     In contrast to RS, there is no narrow temperature region restriction for the M1 SIP mechanism. However, it does have a thermal peak for fragmentation around -15 °C and a minimum drop diameter requirement of 50 μm (Phillips et al., 2018). Where nucleation occurs due to ice particles, these ice particles must be less massive than the drop. Furthermore, fragmentation

was greater in larger drops compared to smaller drops (see Fig. 12). For the shallower clouds with warmer cloud bases, no fragmentation occurred for any drop sizes given in Fig. 12 due to the relatively warm cloud tops of -1 °C. For the shallower clouds with colder cloud bases and the deeper clouds with warmer cloud bases, the cloud tops had temperatures of -9 °C. Some of the largest drops formed in these simulations were approximately 1 mm, where according to Fig. 12(c) fragmentation at -9 ° C would still result in very few fragments. While the deeper clouds with colder cloud bases have some part of the cloud at

the thermal peak of -15 °C (see Fig. S2 of the Supplement), this occurs relatively early on in the simulations when most drop diameters were below the minimum diameter threshold of 50 μm. Furthermore, these clouds remain at -19 °C for most of the simulation, away from the thermal peak, resulting in very few fragments.

     Similar to M1, CB also has a thermal peak for fragmentation around -15 °C for collisions of graupel with graupel or hail and hail with hail. Collisions of dendrites with any ice particles and spatial planar with any ice particle are not temperature

dependent according to the Phillips et al. (2017) parameterisation. The BMM can resolve ice habits based on aspect ratio according to Chen and Lamb (1994), and as indicated in Table 1, an aspect ratio of < 1 and bulk density of < 400 kg m$^{-3}$ defines dendrites and an aspect ratio of < 1 and bulk density of ≥ 400 kg m$^{-3}$ defines spatial planar ice habits. Both aerosol size distributions of the shallower clouds have average aspect ratios close to or above one (see Fig. S21 and S22 of the Supplement). This suggested that any CB occurring in the shallower clouds was temperature dependent. CB was only slightly

active in the colder cloud base simulations with cloud tops of -9 °C, away from the thermal peak of -15 °C, and ice collision frequency, initial kinetic energy, and rime fraction will also determine the degree of ice enhancement. For the deeper clouds,

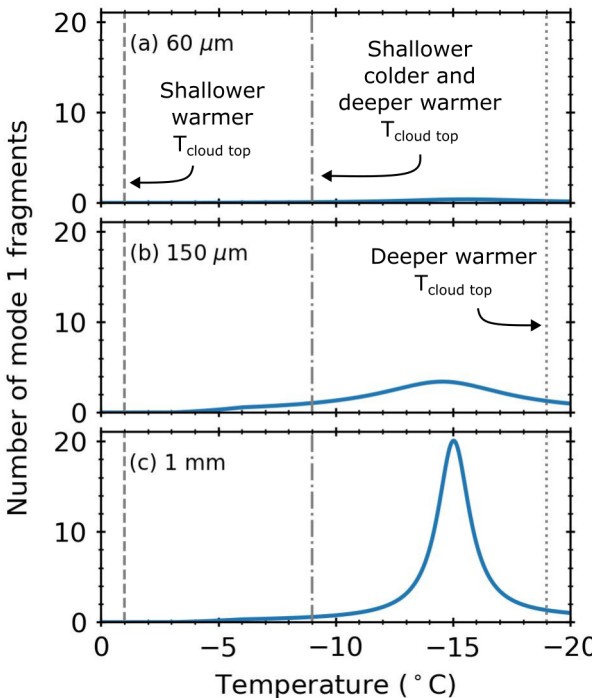

**Figure 12.** The number of fragments due to mode 1 for drop diameters of (a) 60 μm, (b) 150 μm and (c) 1 mm. Shallower refers to clouds with a depth of 1.3 km, and deeper refers to clouds with a depth of 2.4 km. Warmer refers to cloud base temperatures of 7 °C, and colder refers to cloud base temperatures of 0 °C.

the warmer cloud bases had aspect ratios > 1 suggesting columnar habits (see Fig. S23 of the Supplement) and cloud tops of -9, therefore suggesting little ice enhancement from CB. However, significant ice enhancement from CB occurred in the deeper clouds with colder cloud bases where the aspect ratio value is < 1 (plates) (see Fig. S24 of the Supplement) and cloud tops are

525    -19 °C. The aspect ratio for the natural aerosol size distribution with updraft speeds of 0.5 m s$^{-1}$ dropped below one in the latter half of the simulation, hence ice enhancement from CB occurred towards the end of the simulation.

On the other hand, M2 does not have a strict temperature region or thermal peaks like RS, M1 and CB, instead increasing linearly with dimensionless energy and supercooling (Phillips et al., 2018). However, the drop diameter must be greater than 0.15 mm and the ice particle more massive than the drop. Therefore, M2 takes time to become active in the simulated clouds

due to the requirement of large drops and ice particles, hence ice enhancement due to M2 occurred more in the latter half of the simulations.

While M2 is the most prevalent SIP mechanism responsible for ice enhancement in our simulations, there is still a large uncertainty in the M2 freezing fraction (Φ). Before we studied M2 in the laboratory (James et al., 2021), the theoretical work of Phillips et al. (2018) suggested that a reasonable value of Φ would be 0.5. The results from our laboratory study suggested

that the value of Φ was closer to 0.3, given the limitations of the experimental setup used. The results from our sensitivity

tests on the M2 Φ values in Figs. 4 and 8 show only a small difference in ice enhancement between using these two Φ values, approximately $10\,\text{L}^{-1}$. However, ice enhancement was more significantly affected when values of Φ ≤0.1 were used, with little to no ice enhancement observed in values of Φ below 0.01. Phillips et al. (2018) suggested that based on a study by Latham and Warwicker (1980) '*Φ must be somewhere between about 0.01 and 1, with the range 0.1–1 seeming more likely than 0.01*'.

If this is correct, then even with the lower Φ value of 0.1, ice enhancement values would not be trivial. For the examples shown in Figs. 4 and 8 the ice enhancement was close to $10\,\text{L}^{-1}$ more for the colder cloud base simulations and $4\,\text{L}^{-1}$ for the warmer cloud base simulations.

Other uncertainties in the M2 parameterisation include the critical value for onset of splashing ($DE_{cit}$). For splashing to occur, the dimensionless energy must be greater than $DE_{crit}$. Phillips et al. (2018) estimated a value for $DE_{crit}$ of ∼0.2

based on room temperature experiments of colliding drops by Low and List (1982) and acknowledged that this was a source of uncertainty. The equation used to describe number of secondary liquid drops was further constrained using laboratory experiments by Levin et al. (1971). These experiments consisted of 2.5 mm drops impacting on a rough copper sphere at room temperature. The number of secondary drops formed in collision experiments is sensitive both to the geometry and impact material; hence the expression used to describe the number of liquid fragments presents another source of uncertainty.

We increased the $DE_{crit}$ by a factor of 10 and 30 to demonstrate the sensitivity of M2 ice enhancement on the $DE_{crit}$, as shown in Figs. 5 and 9. When the $DE_{crit}$ was increased by a factor of 10 or 30, the ice enhancement decreased compared to the stated $DE_{crit}$ in Phillips et al. (2018), but ice enhancement still occurred. Despite the uncertainties within the M2 parameterisation, and in agreement with Phillips et al. (2018), it is more erroneous not to include M2 within simulations. Our laboratory experiments strongly suggest that the M2 SIP mechanism could be active (James et al., 2021), and further laboratory

studies into this mechanism will help reduce the uncertainties listed.

In simulations with near-city aerosol size distributions and updraft speeds of $0.5\,\text{m s}^{-1}$, no SIP mechanisms were active. In these clouds, the CDNCs remained constant at approximately $100\,\text{cm}^{-3}$, considerably lower than the corresponding natural aerosol size distribution simulations. In addition to the low CDNCs, the drop size distribution in these clouds was significantly narrower than in simulations with a natural aerosol size distribution (see Figs. S14 and S16 of the Supplement). Usually,

we expect a more active collision-coalescence process in simulations with lower updraft speeds due to the activation of fewer aerosol particles resulting in fewer cloud drops. However, this result demonstrates the importance of the aerosol size distribution and updraft speed. In the near-city aerosol size distributions, activation only occurs in aerosol particles > 400 nm diameter, whereas in the natural aerosol size distributions, activation occurs in aerosols > 250 nm. A narrow range of aerosol activation sizes results in a narrower drop size distribution suppressing the collision-coalescence process, which will prevent or hinder

most of the SIP mechanisms investigated.

The effect of aerosol particle number, size and hygroscopicity on activation of cloud condensation nuclei has been investigated by Reutter et al. (2009) under pyro-convective conditions. They linked the ratio of pyro-convective aerosol particle number concentration and updraft speed to cloud drop formation noting three distinct regimes: aerosol-limited, updraft-limited and aerosol- and updraft- sensitive. IAerosol particle concentrations strongly determined the CDNC in the aerosol-limited

regime. Updraft speed strongly determined the CDNC in the updraft-limited regime. A combination of aerosol particle con-

centrations and updraft speed determined the aerosol- and updraft-sensitive regime. In the updraft-limited regime, the CDNC is strongly dependent on the updraft speed. In general, these regimes are not specific to the pyro-convective conditions of Reutter et al. (2009) (e.g. Gunthe et al., 2009), but the values used to define the boundaries between these regimes are. In fact, Reutter et al. (2009) suggested that shallow convection over polluted regions is more likely to be updraft limited, whereas; shallow convection in moderately polluted air is more likely to be aerosol and updraft sensitive. We appear to see something similar with the two aerosol size-distributions used with the near-city aerosol size distribution with updraft speeds of 0.5 m s$^{-1}$ potentially in an updraft-limited regime and the natural aerosol size distribution with updraft speeds of 0.5 m s$^{-1}$ potentially in the aerosol- and updraft-sensitive regime.

As SIP mechanisms rely on the presence of pre-existing ice particles, such as those formed via the action of INPs, we need to consider how the initial concentration of ice particles affects SIP. In our study, we varied the initial INP concentrations. For the warmer cloud bases, where SIP mechanisms were active for either aerosol size distribution, there was a linear effect on INP concentrations such that the $\times 0.1$ INP had the lowest ice enhancement and the $\times 10$ INP had the highest ice enhancement. Phillips et al. (2003) also observed results where increasing the initial INP increases the SIP ICNC, due to rime-splintering in their case, using a bulk microphysics model for deep convective clouds. However, for the colder cloud bases there was a non-linear effect with the $\times 10$ INP concentration reducing or even suppressing ice enhancement. The effect of increasing the INP concentration, which in turn decreases SIP ICNC, has been observed before. For example, simulations using a 1-D column model with bin-microphysics of shallow wintertime cumulus clouds showed that when the INP was multiplied by 100, the RS mechanism was 'turned off' due to the Wegener-Bergeron Findeisen (WBF) process (Crawford et al., 2012). Figures S.25 & 26 of the Supplement show the liquid and ice water contents for M2 simulations with a natural aerosol size distribution, respectively, with extended simulation runtimes, demonstrating that the WBF process also occurred in our colder cloud base simulations. However, the WBF process occurred earlier, with a different profile, in the $\times 10$ INP concentration simulation compared to the $\times 1$ or $\times 0.1$ INP concentration simulations as initially there was a higher ice water content in the simulations, which allowed more water vapour to condense onto the ice suppressing the coalescence process, hence reducing SIP. Whereas, for the $\times 0.1$ and $\times 1$ INP simulations, coalescence occurred earlier due to the lower ice water content allowing SIP formation, which increased the ice enhancement.

These results are important as increasing the INP concentration does not always increase ICNC, especially given the uncertainty within the DeMott et al. (2010) INP parameterisation. We also note that as these are idealised clouds, the DeMott et al. (2010) INP parameterisation works adequately as a 'global' ambient representation of aerosol particles. However, the DeMott et al. (2010) INP parameterisation is not suitable for marine environments, and the aerosol particle size dependency may be less sensitive at temperatures warmer than -15 °C, which is the temperature region of our idealised clouds. How best to parameterise INPs is an active area of research (e.g. see review by Burrows et al., 2022) and outside the scope of this work.

Several studies have included M2 in their simulations, although usually combined with M1 (Phillips et al., 2018; Sotiropoulou et al., 2020; Qu et al., 2020; Zhao et al., 2021; Zhao and Liu, 2021; Georgakaki et al., 2022; Zhao and Liu, 2022; Huang et al., 2022; Karalis et al., 2022) which is equivalent to our M1+M2 simulations. For our idealised cloud conditions, the contribution from our M1+M2 simulations appears to be derived from the M2 aspect. In fact it is only when we combine M1 with M2

that we see any significant ice enhancement from M1. On its own, M1 was not a strong ice enhancement mechanism in our idealised shallow convective clouds. In contrast, Phillips et al. (2018) (see their Fig. 6) showed that in their simulation of a deep tropical maritime convective cloud with temperatures between 0 to -20 °C, over 90 % of the ice came from M1 and M2, of which M1 contributed to approximately double that of M2. M1 was more significant in Phillips et al. (2018) simulations compared to M2, probably due to simulation temperatures which covered the thermal peak of M1 at -15 °C. As shown in Fig. 12, very few fragments were formed away from the thermal peak. Most of our simulations were warmer than the thermal peak and the deeper clouds with colder cloud base temperatures of 0 °C only briefly had temperatures near the thermal peak early on in the simulation (see Fig. S2(b) of the Supplementary). Similar results were also observed by Qu et al. (2020) who modelled similar deep tropical maritime convective cloud conditions that Phillips et al. (2018) simulated. Zhao et al. (2021) simulated four types of Arctic mixed-phase clouds based on Atmospheric Radiation Measurement Mixed-Phase Arctic Cloud Experiment observations. They showed that approximately 80 % of all ice particles came from M1 in their single-layer boundary layer stratus simulations. The largest contribution of M2 came from the transition simulations but this only contributed a small fraction compared to M1. Other studies which modelled the M2 SIP mechanism with the M1 SIP mechanism did not provide a breakdown of the contribution from M1 and M2 (Zhao and Liu, 2021, 2022; Huang et al., 2022; Karalis et al., 2022). Two studies found that M2 combined with M1 was not an effective SIP mechanism in the simulated conditions (Sotiropoulou et al., 2020; Georgakaki et al., 2022). Sotiropoulou et al. (2020) stated that this was due to their thermodynamic conditions with relatively cold cloud base temperatures and Georgakaki et al. (2022) stated that this was due to the drops being too small to initiate M1 + M2 in their simulated conditions.

## 6 Conclusions

To summarise our key results:

- No SIP mechanisms were active in simulations with a near-city aerosol size distribution and updraft speeds of 0.5 m s$^{-1}$ indicating a significant sensitivity of ice enhancement within these simulated clouds to the aerosol particle size distribution when updraft speeds are low.

- Across all simulations where SIP mechanisms were active, M2 was the most prevalent SIP mechanism, especially for shallower clouds. While M2 was still prevalent in the deeper clouds, there were cases where RS or CB contributed significantly to ice enhancement in the colder cloud bases.

- Ice enhancement from M2 was particularly sensitive to the freezing fraction value ($\Phi$).

- There was a high sensitivity to INP concentration at the highest INP concentrations ($\times10$) which delayed the collision-coalescence process and, hence ice enhancement.

Our results suggest that M2 may be a significant ice enhancement mechanism in shallow convective clouds where large drops are present especially when cloud tops are warmer than -15 °C, the thermal peak of M1 and M2. It may also be significant for

clouds where large drops do not reside within the RS region. However, there are still many areas within the M2 parameterisation that need to be addressed via further laboratory work as detailed in James et al. (2021), a theme which can be applied to all SIP parameterisations. Our results also show that the parameterisation of INPs can have an effect on SIP mechanisms, and certainly
with modelling studies these should be varied.

*Code and data availability.* The University of Manchester bin microphysics model is available upon request. The model outputs are deposited in Figshare, a FAIR-aligned (findable, accessible, interoperable and re-usable) data repository, and can be accessed at

https://doi.org/10.48420/c.6238311

## Appendix A

**Table A1.** List of SIP combinations

| List of SIP combinations |
|---|
| RS |
| M1 |
| M2 |
| CB |
| RS+M1 |
| RS+M2 |
| RS+CB |
| M1+M2 |
| M1+CB |
| M2+CB |
| RS+M1+M2 |
| RS+M2+CB |
| RS+M1+CB |
| M1+M2+CB |
| RS+M1+M2+CB |

**Appendix B**

Table B1: List of symbols

| Symbol | Description | Value and units |
|--------|-------------|-----------------|
| $A$ | Number density of the breakable asperities in the region of contact | - |
| $c_w$ | Specific heat capacity of liquid water | $4200 \text{ J kg}^{-1} \text{ K}^{-1}$ |
| $C$ | Asperity fragility coefficient | - |
| $D_a$ | Diameter of aerosol particle | m |
| $D_d$ | Diameter of drop in mode 2 | m |
| $D_{a,m}$ | Median aerosol particle diameter of mode | (m) |
| $D_{i,m}$ | Median aerosol particle diameter of mode A, B or C | nm |
| $D_s$ | Diameter of the smaller colliding particle in ice-ice collisional breakup | m |
| $DE$ | Dimensionless energy | - |
| $DE_{crit}$ | Critical value of dimensionless energy for onset of splashing | 0.2, unless otherwise stated |
| $f$ | Mass fraction of a drop frozen by the end of stage 1 freezing | - |
| $f_{RS}$ | Function of rime splintering | - |
| $F$ | Interpolation function for the onset of fragmentation | - |
| $K_{0(CB)}$ | Collisional kinetic energy at impact | J |
| $L_f$ | Specific latent heat of freezing | $3.3 \times 10^5 \text{ J kg}^{-1}$ |
| $m_r$ | Mass of rime | kg |
| $m_1, m_2$ | Mass of colliding ice particles | kg |
| $m_d$ | Mass of drop in mode 2 | kg |
| $m_i$ | Mass of ice particle in mode 2 | kg |
| $N$ | Number density of aerosol particles | $\text{kg}^{-1}$ |
| $N_{CB}$ | Number of ice particles due to ice-ice collisional breakup | - |
| $N_i$ | Total number density of aerosol particles of mode A, B or C | $\text{cm}^{-3}$ |
| $N_{M1T}$ | Total number of ice particles due to mode 1 | - |
| $N_{M1L}$ | Total number of large ice particles due to mode 1 | - |
| $N_{M2}$ | Number of ice particles per drop accreted due to mode 2 | - |
| $N_{RS}$ | Number of ice particles due to rime splintering | - |
| $N_T$ | Total number of aerosol particles of mode | $\text{kg}^{-1}$ |
| $R_{FL}$ | Rime fraction of the larger colliding particle | - |
| $R_{FS}$ | Rime fraction of the smaller colliding particle | - |
| $t$ | Time | s |
| $T$ | Freezing temperature of water drop | °C |
| $T_0$ | Value of T at maximum of Lorentzian for Eq.5 | °C |

| | | |
|---|---|---|
| $T_{B0}$ | Value of T at maximum of Lorentzian for Eq.6 | °C |
| $v_1, v_2$ | Fall speed of colliding ice particles | m s$^{-1}$ |
| $v_d$ | Fall speed of drop in mode 2 | m s$^{-1}$ |
| $v_i$ | Fall speed of ice particle in mode 2 | m s$^{-1}$ |
| $w$ | Updraft speed | m s$^{-1}$ |
| $\alpha$ | Equivalent spherical surface area of the smaller colliding particle | m$^2$ |
| $\beta$ | Parameter in Eq. 5 | K$^{-1}$ |
| $\beta_B$ | Parameter in Eq. 6 | K$^{-1}$ |
| $\gamma$ | Parameter related to riming intensity | - |
| $\gamma_l$ | Surface tension of liquid water | 0.073 J m$^{-2}$ |
| $\zeta$ | Intensity of Lorentzian in Eq. | - |
| $\zeta_B$ | Intensity of Lorentzian in Eq. | - |
| $\eta$ | Half width of Lorentzian in Eq. 5 | °C |
| $\eta_B$ | Half width of Lorentzian in Eq. 6 | °C |
| $\kappa$ | Hygroscopicity parameter | 0.61 |
| $\rho_L$ | Density of the larger colliding particle | kg m$^{-3}$ |
| $\sigma_g$ | Standard geometric deviation of the logarithmic distribution | - |
| $\sigma_i$ | Standard geometric deviation of the logarithmic distribution of mode A, B or C | - |
| $\Phi$ | Probability of any drop in the mode 2 splash containing ice | 0.3, unless otherwise stated |
| $\Phi_s$ | Aspect ratio of the smaller colliding particle | - |
| $\Omega$ | Interpolating function for the onset of fragmentation | - |

*Author contributions.* RLJ and PJC conceived the original study. PJC developed the model code. RLJ performed the simulations, analysed the data and wrote the paper. PJC and JC contributed to scientific discussions and provided comments on the paper.

*Competing interests.* The authors declare that they have no conflict of interest.

*Acknowledgements.* This research has been supported the Natural Environment Research Council (grant no. NE/T001496/1).

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

**SECTION S1:** Temperature profiles

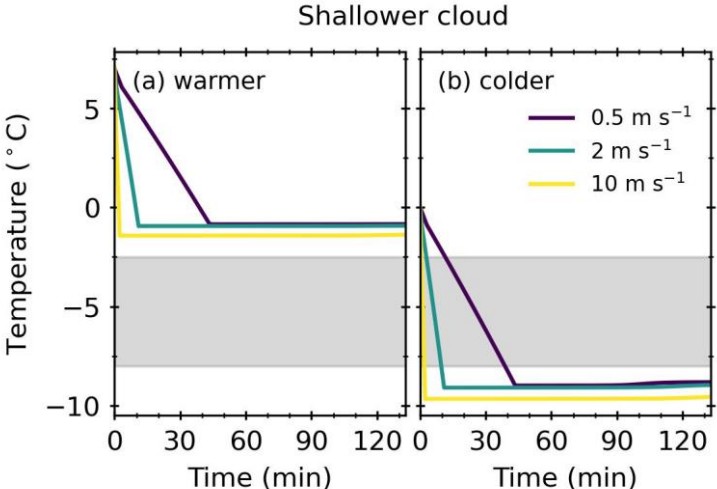

**Figure S1** Temperature profiles of a shallower (1.3 km deep) cloud. Warmer refers to cloud base temperatures of 7 °C, and colder refers to cloud base temperatures of 0 °C. The grey shaded regions indicate the temperature region in which rime-splintering could be active.

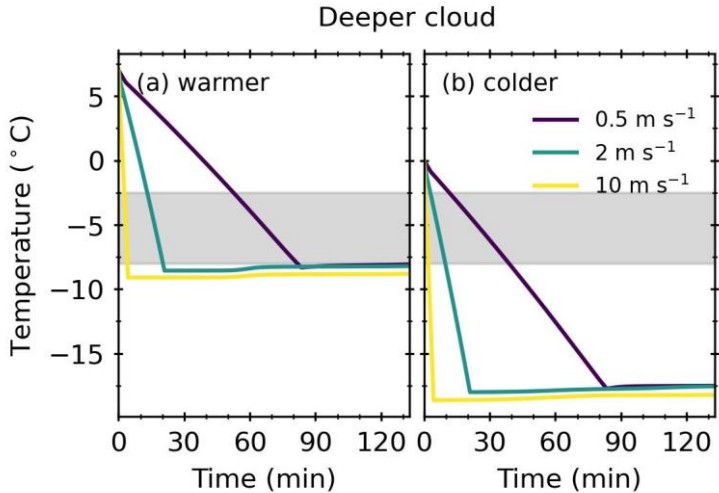

**Figure S2** Temperature profiles of a deeper (2.4 km) cloud. Warmer refers to cloud base temperatures of 7 °C, and colder refers to cloud base temperatures of 0 °C. The grey shaded regions indicate the temperature region in which rime-splintering could be active.

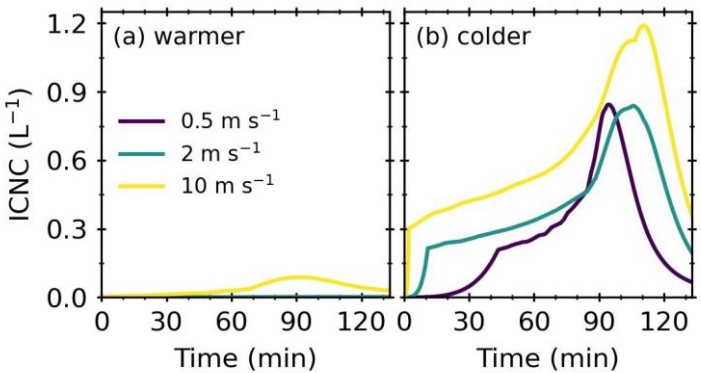

**Figure S3** Control simulation ice crystal number concentrations for shallower clouds (1.3 km deep) with a natural aerosol size distribution. Warmer refers to cloud base temperatures of 7 °C, and colder refers to cloud base temperatures of 0 °C.

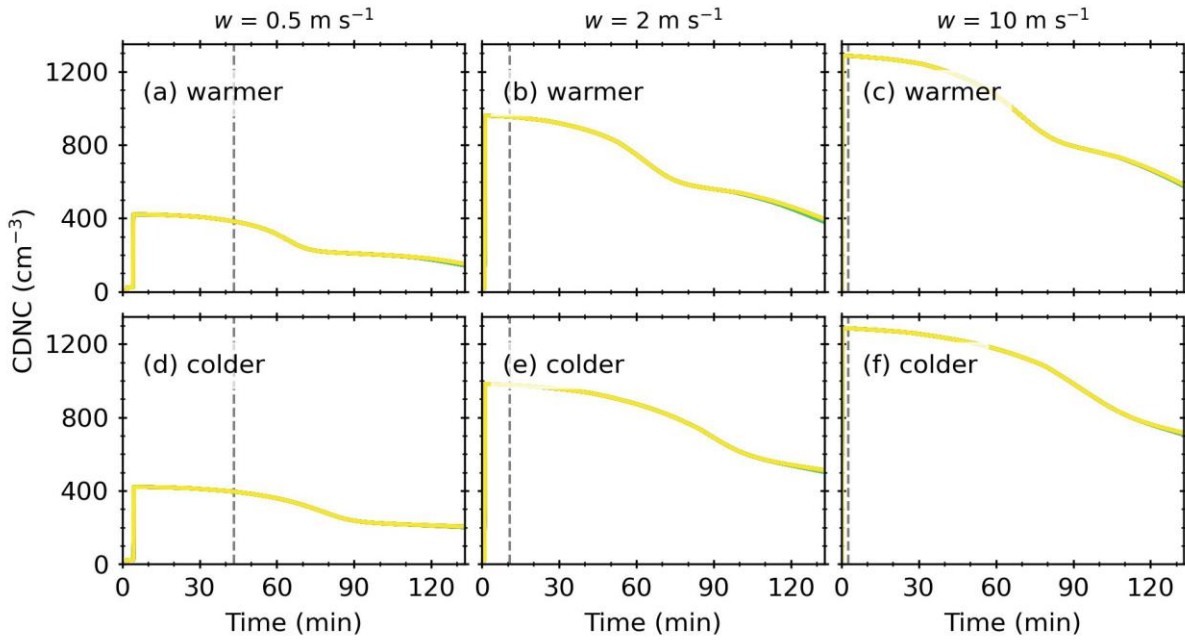

**Figure S4** Cloud drop number concentrations for a shallower cloud (1.3~km deep) with a natural aerosol size distribution. Warmer refers to cloud base temperatures of 7 °C, and colder refers to cloud base temperatures of 0 °C.

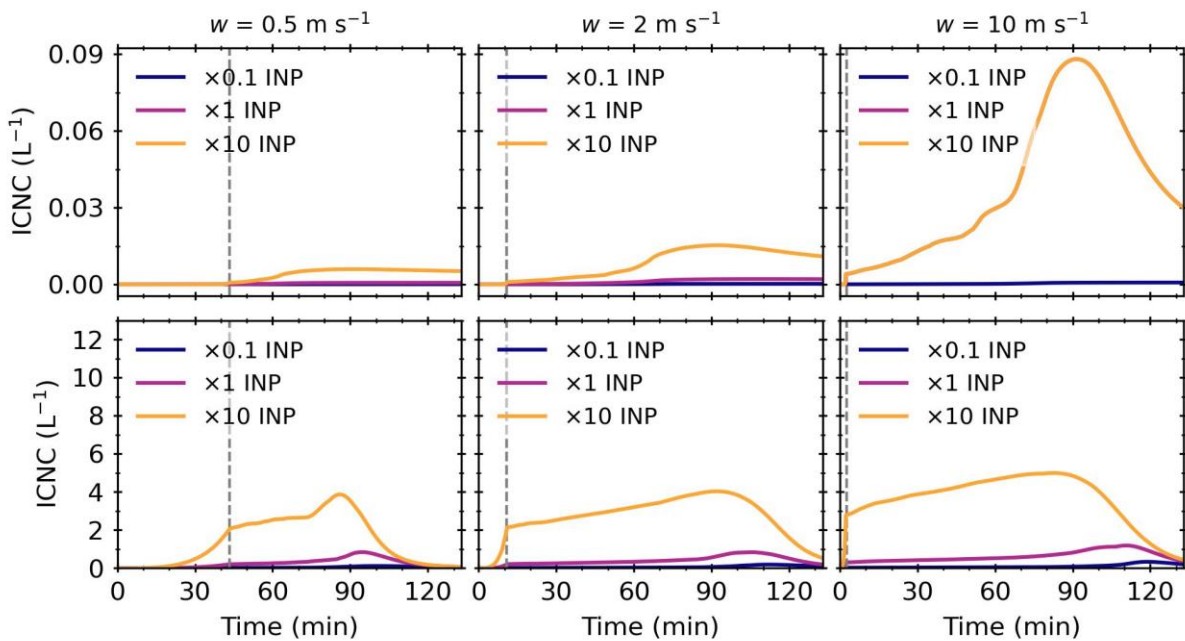

**Figure S5** Ice crystal number concentrations for control simulations against simulation time for three initial INP concentrations (×0.1, ×1 and ×10) for a shallower cloud (1.3 km deep) with a natural aerosol size distribution. Warmer refers to cloud base temperatures of 7 °C, and colder refers to cloud base temperatures of 0 °C.

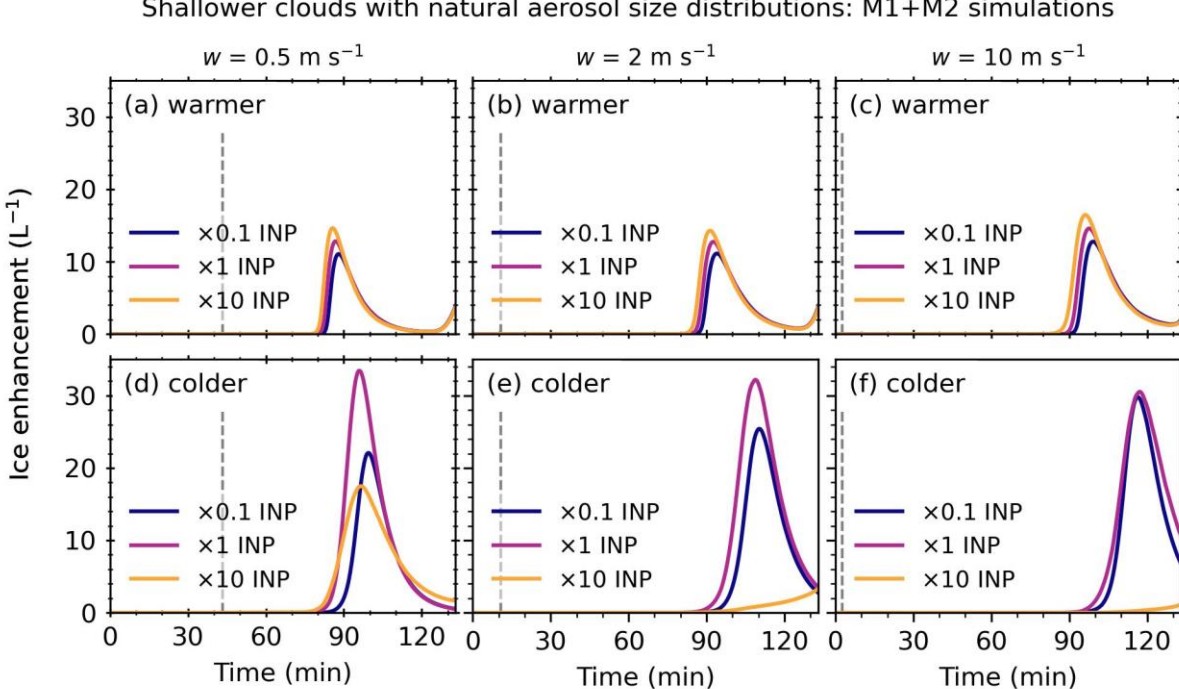

**Figure S6** Mode 1 and mode 2 ice enhancement against simulation time for three INP concentrations (×0.1, ×1 and ×10) for a shallower (1.3 km deep) cloud with a natural aerosol size distribution. Warmer refers to cloud base temperatures of 7 °C, and colder refers to cloud base temperatures of 0 °C.

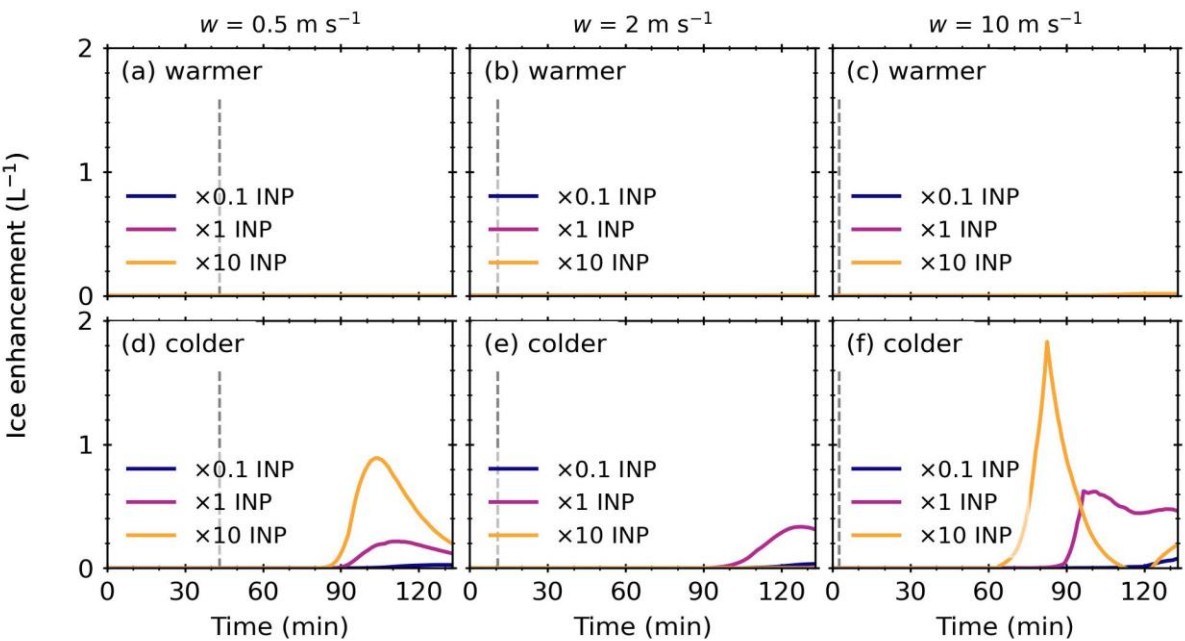

**Figure S7** Ice-ice collisional breakup ice enhancement against simulation time for three INP concentrations (×0.1, ×1 and ×10) for a shallower (1.3 km deep) cloud with a natural aerosol size distribution. Warmer refers to cloud base temperatures of 7 °C, and colder refers to cloud base temperatures of 0 °C.

**SECTION S3:** Additional figures for shallower clouds with a near-city size distribution

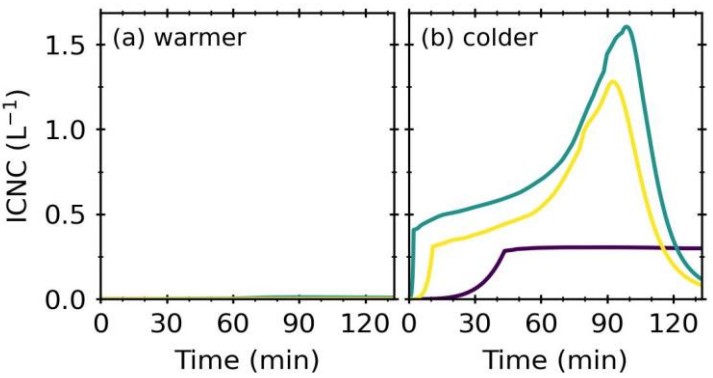

**Figure S8** Control simulation ice crystal number concentrations for shallower clouds (1.3 km deep) with a natural aerosol size distribution. Warmer refers to cloud base temperatures of 7 °C, and colder refers to cloud base temperatures of 0 °C.

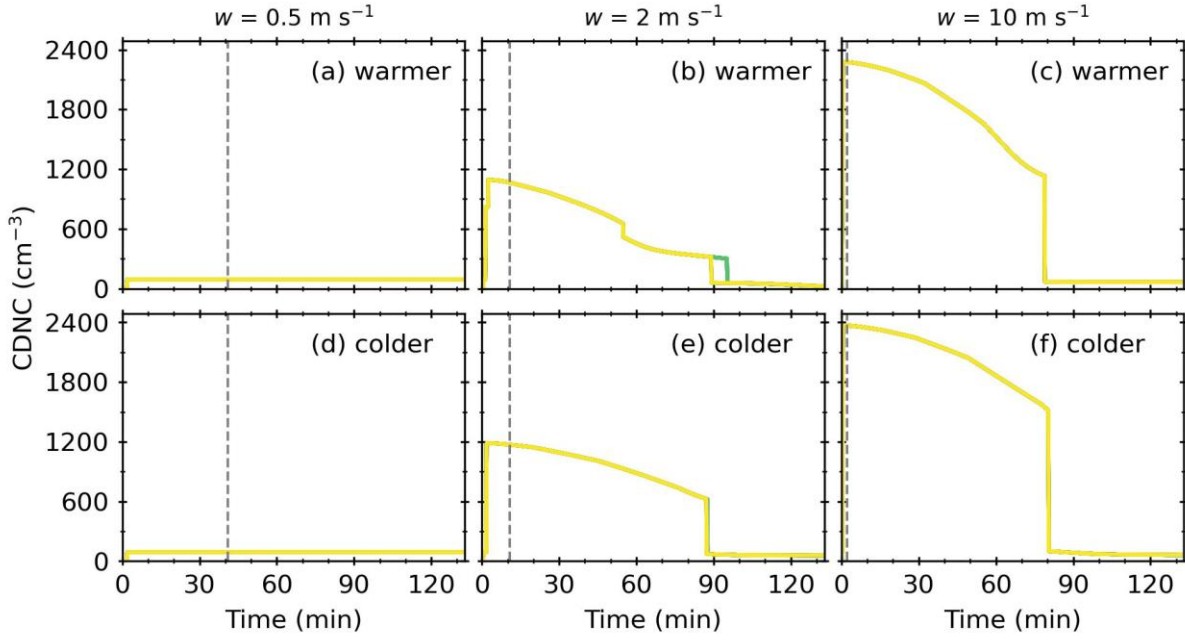

**Figure S9** Cloud drop number concentrations for a shallower cloud (1.3~km deep) with a near-city aerosol size distribution. Warmer refers to cloud base temperatures of 7 °C, and colder refers to cloud base temperatures of 0 °C.

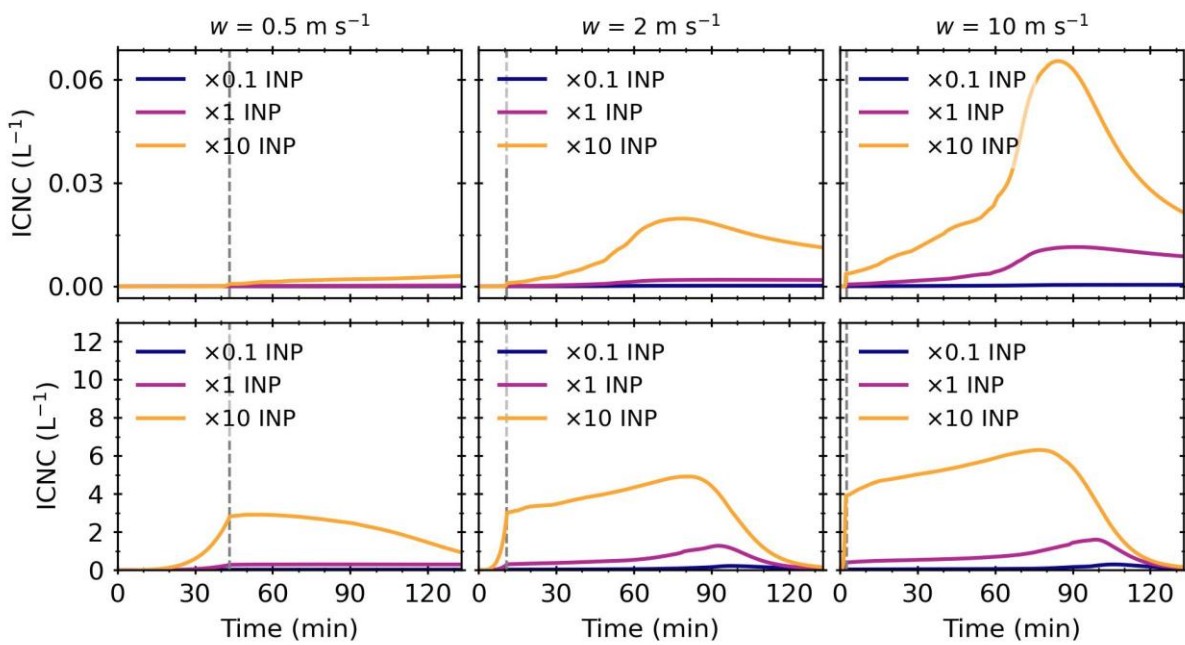

**Figure S10** Ice crystal number concentrations for control simulations against simulation time for three initial INP concentrations (×0.1, ×1 and ×10) for a shallower cloud (1.3 km deep) with a near-city aerosol. Warmer refers to cloud base temperatures of 7 °C, and colder refers to cloud base temperatures of 0 °C.

Shallower clouds with near-city aerosol size distributions: M1+M2 simulations

$w = 0.5 \text{ m s}^{-1}$     $w = 2 \text{ m s}^{-1}$     $w = 10 \text{ m s}^{-1}$

(a) warmer (b) warmer (c) warmer
(d) colder (e) colder (f) colder

×0.1 INP
×1 INP
×10 INP

Ice enhancement (L⁻¹)

Time (min)

**Figure S11** Mode 1 and mode 2 ice enhancement against simulation time for three INP concentrations (×0.1, ×1 and ×10) for a shallower (1.3 km deep) cloud with a near-city aerosol size distribution. Warmer refers to cloud base temperatures of 7 °C, and colder refers to cloud base temperatures of 0 °C.

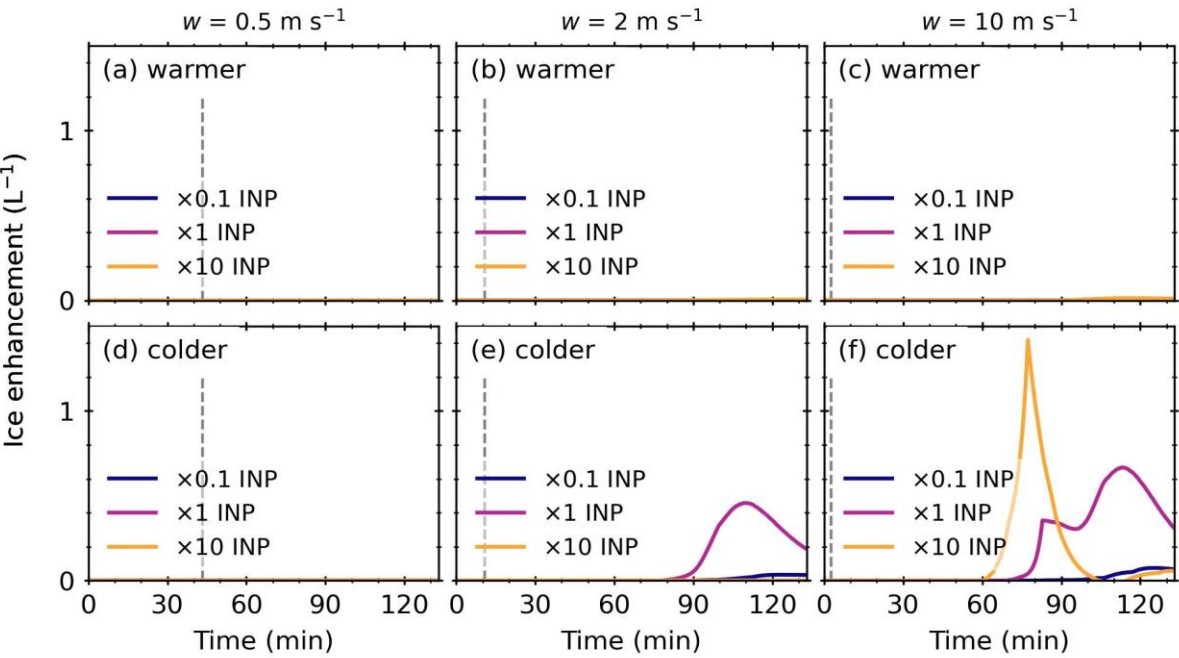

**Figure S12** Ice-ice collisional breakup ice enhancement against simulation time for three INP concentrations (×0.1, ×1 and ×10) for a shallower (1.3 km deep) cloud with a natural aerosol size distribution. Warmer refers to cloud base temperatures of 7 °C, and colder refers to cloud base temperatures of 0 °C.

**SECTION S4:** Additional figures for deeper clouds with a natural aerosol size distribution

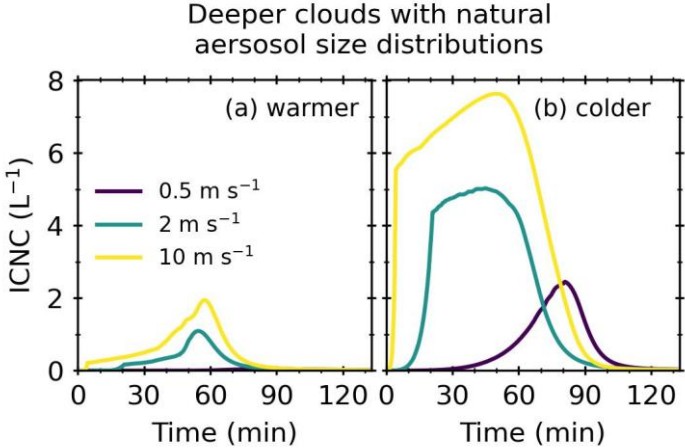

**Figure S13** Control simulation ice crystal number concentrations for deeper clouds (1.3 km deep) with a natural aerosol size distribution. Warmer refers to cloud base temperatures of 7 °C, and colder refers to cloud base temperatures of 0 °C.

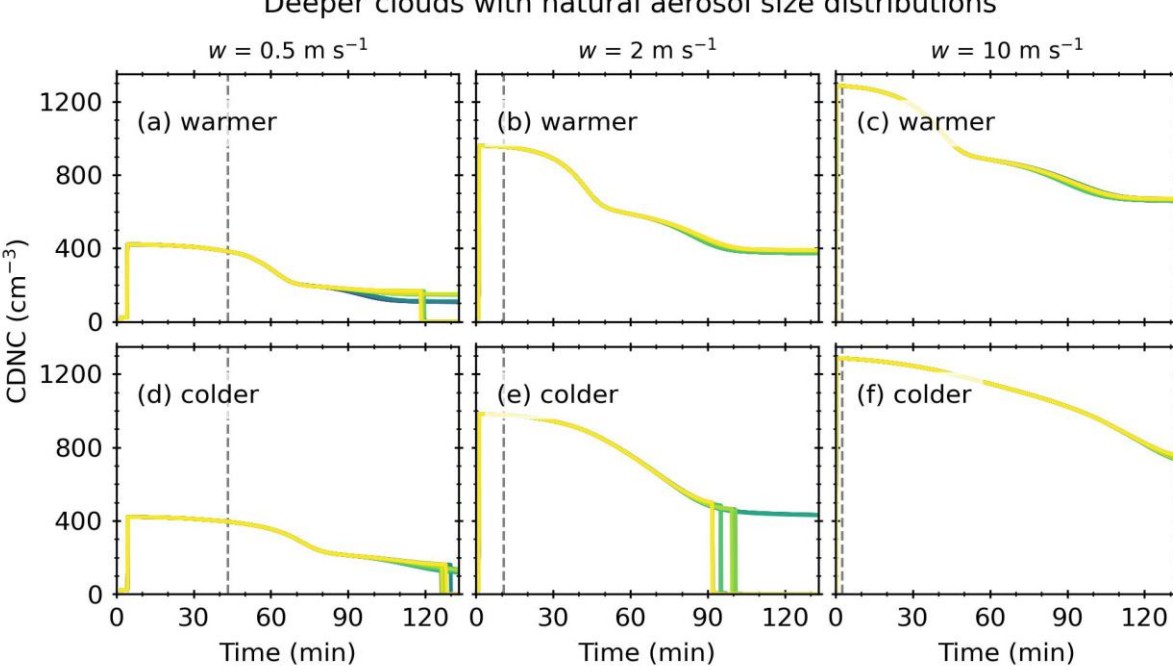

**Figure S14** Cloud drop number concentrations for a deeper cloud (1.3 km deep) with a natural aerosol size distribution. Warmer refers to cloud base temperatures of 7 °C, and colder refers to cloud base temperatures of 0 °C.

**SECTION S5:** Additional figures for deeper clouds with a near-city size distribution

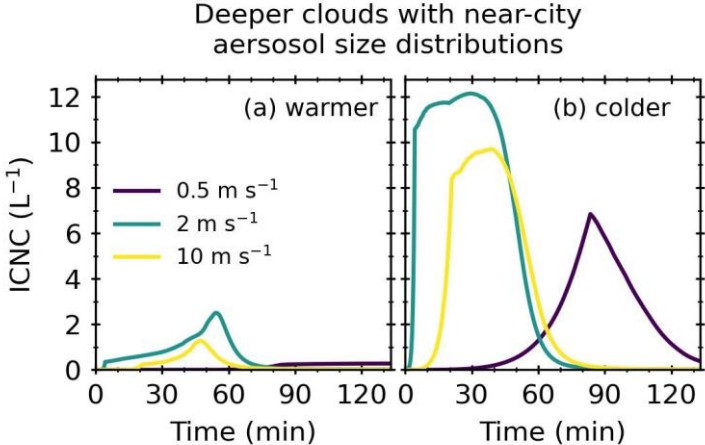

**Figure S15** Control simulation ice crystal number concentrations for deeper clouds (2.4 km deep) with a near-city aerosol size distribution. Warmer refers to cloud base temperatures of 7 °C, and colder refers to cloud base temperatures of 0 °C.

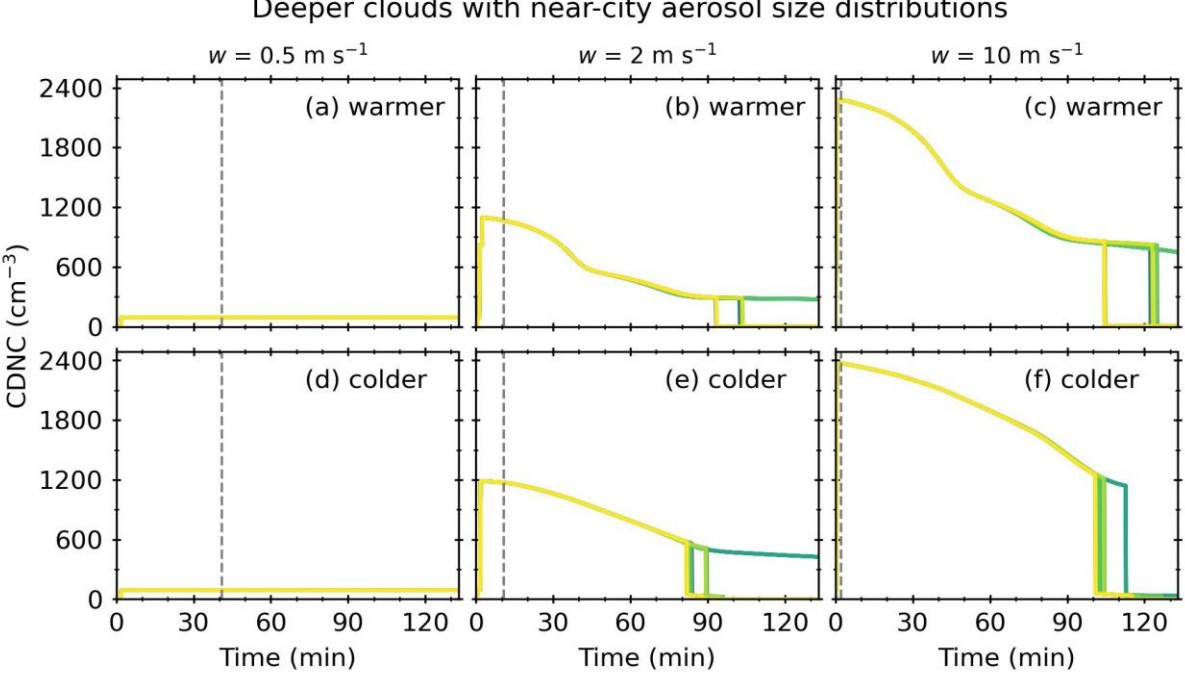

**Figure S16** Cloud drop number concentrations for a deeper cloud (2.4 km deep) with a near-city aerosol size distribution. Warmer refers to cloud base temperatures of 7 °C, and colder refers to cloud base temperatures of 0 °C.

**SECTION S6:** Particle and ice size distribution contour plots

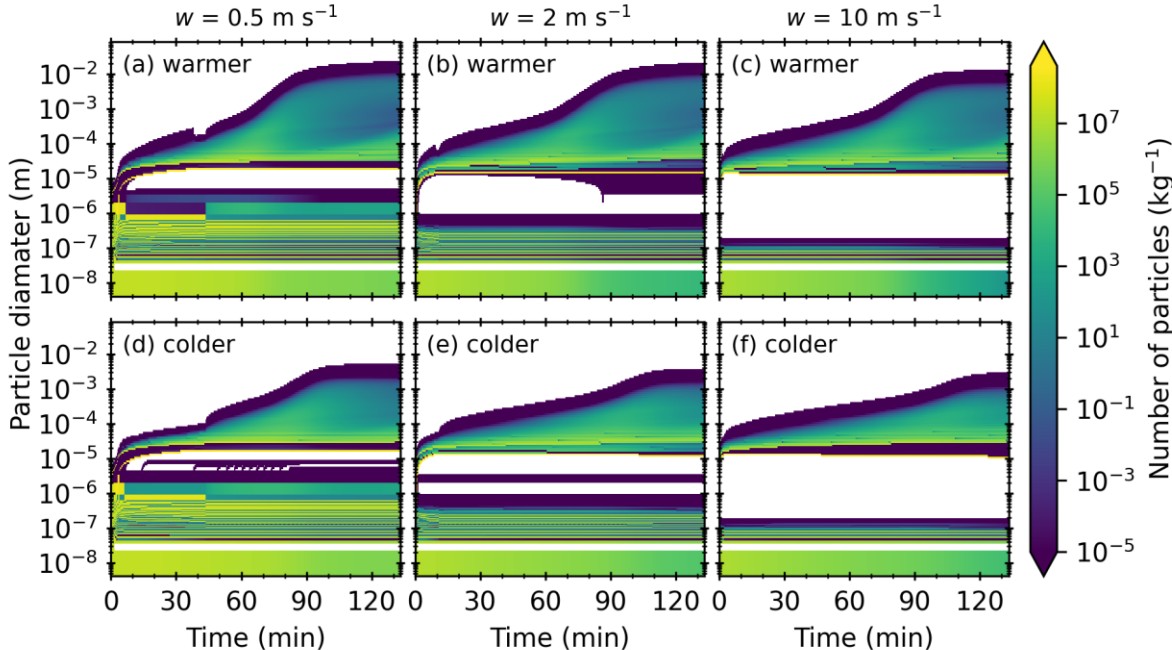

**Figure S17** Contour plot showing the particle size distribution as a function of simulation time for shallower clouds (1.3 km deep) with natural aerosol size distributions. Warmer refers to cloud base temperatures of 7 °C, and colder refers to cloud base temperatures of 0 °C.

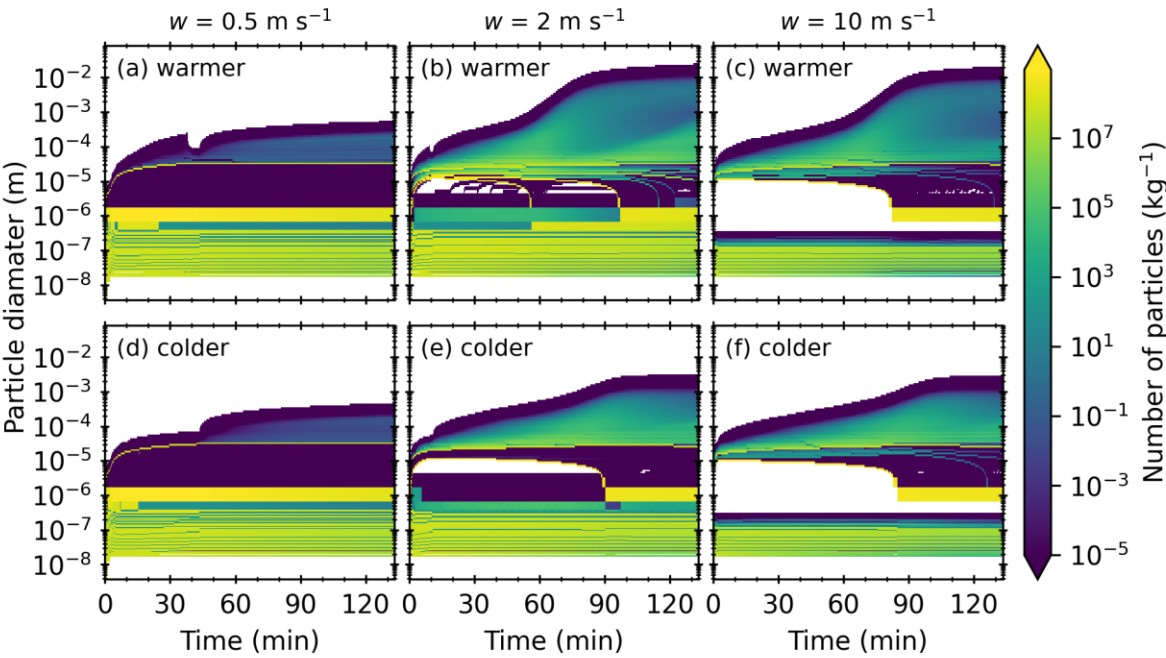

**Figure S18** Contour plot showing the particle size distribution as a function of simulation time for shallower clouds (1.3 km deep) with near-city aerosol size distributions. Warmer refers to cloud base temperatures of 7 °C, and colder refers to cloud base temperatures of 0 °C.

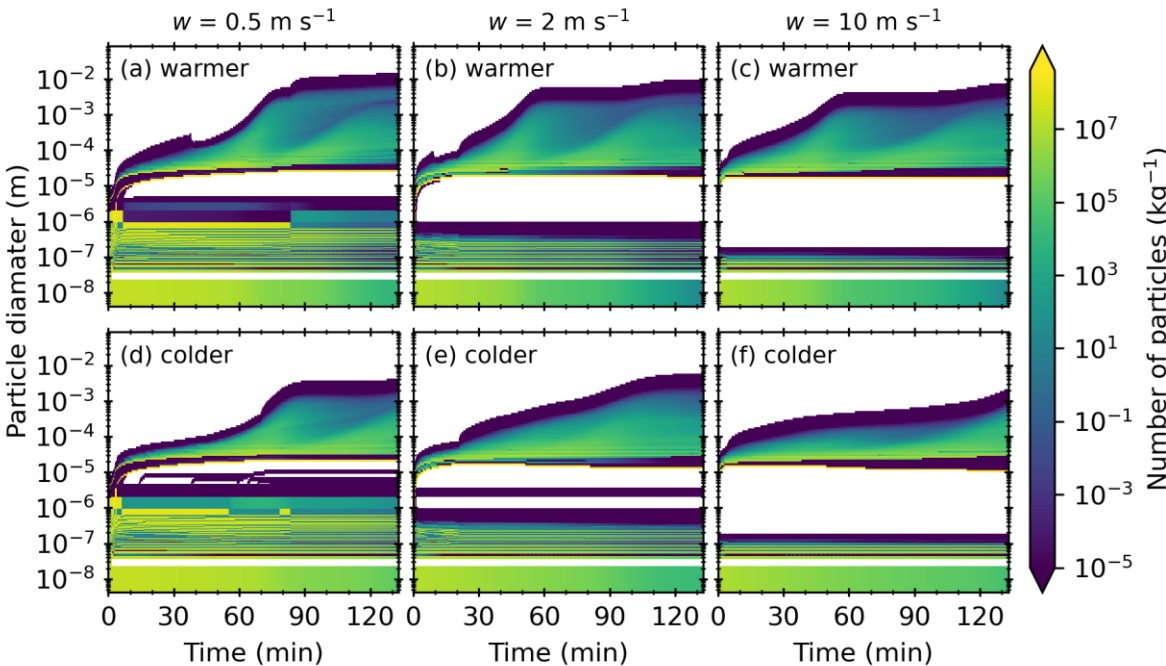

**Figure S19** Contour plot showing the particle size distribution as a function of simulation time for deeper clouds (2.4 km deep) with natural aerosol size distributions. Warmer refers to cloud base temperatures of 7 °C, and colder refers to cloud base temperatures of 0 °C.

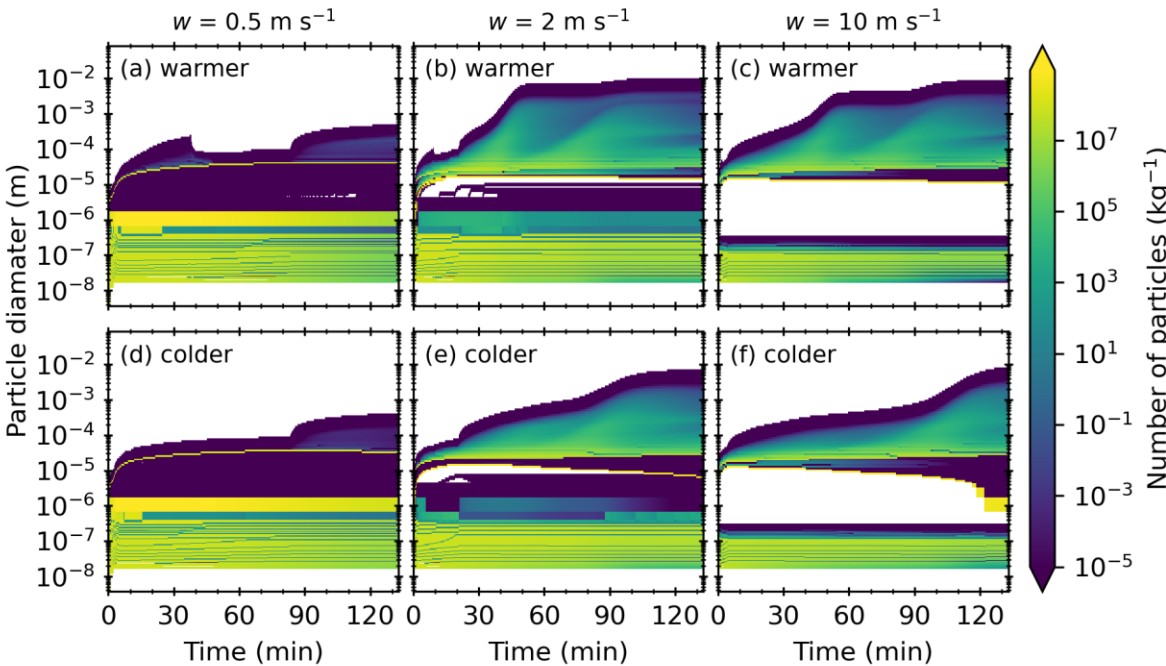

**Figure S20** Contour plot showing the particle size distribution as a function of simulation time for deeper clouds (2.4 km deep) with natural aerosol size distributions. Warmer refers to cloud base temperatures of 7 °C, and colder refers to cloud base temperatures of 0 °C.

# SECTION S7: Ice particle aspect ratio figures

## Shallower clouds with natural aerosol size distributions

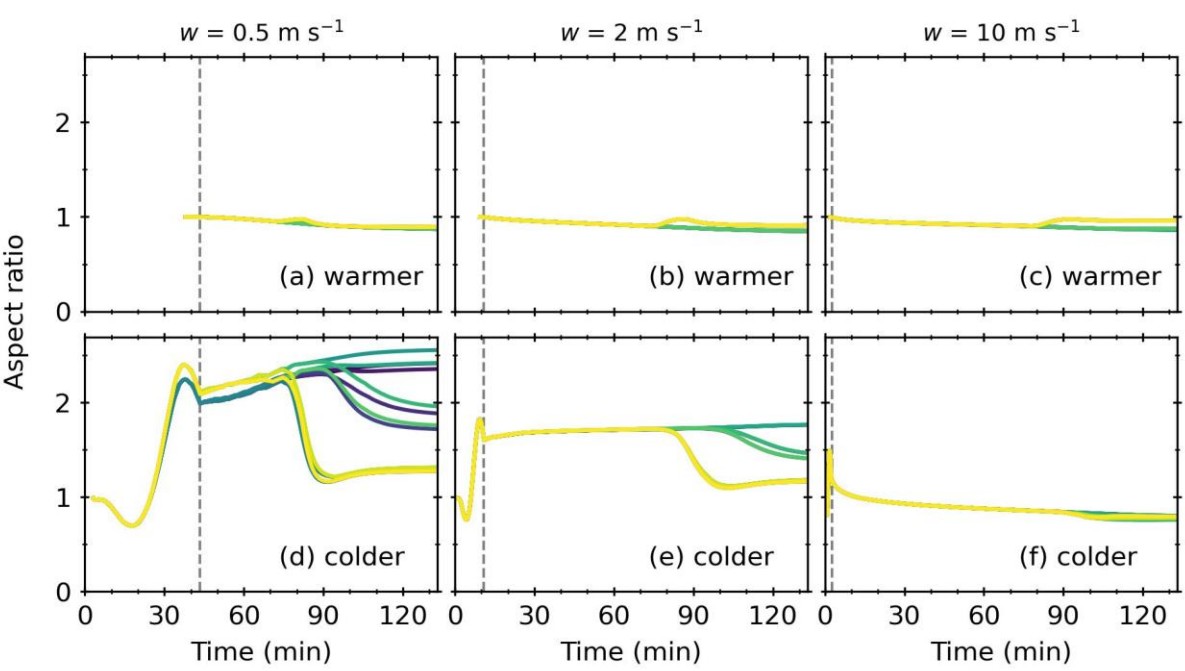

**Figure S21** Ice particle aspect ratio for a shallower cloud (1.3 km deep) with a natural aerosol size distribution. Warmer refers to cloud base temperatures of 7 °C, and colder refers to cloud base temperatures of 0 °C.

## Shallower clouds with near-city aerosol size distributions

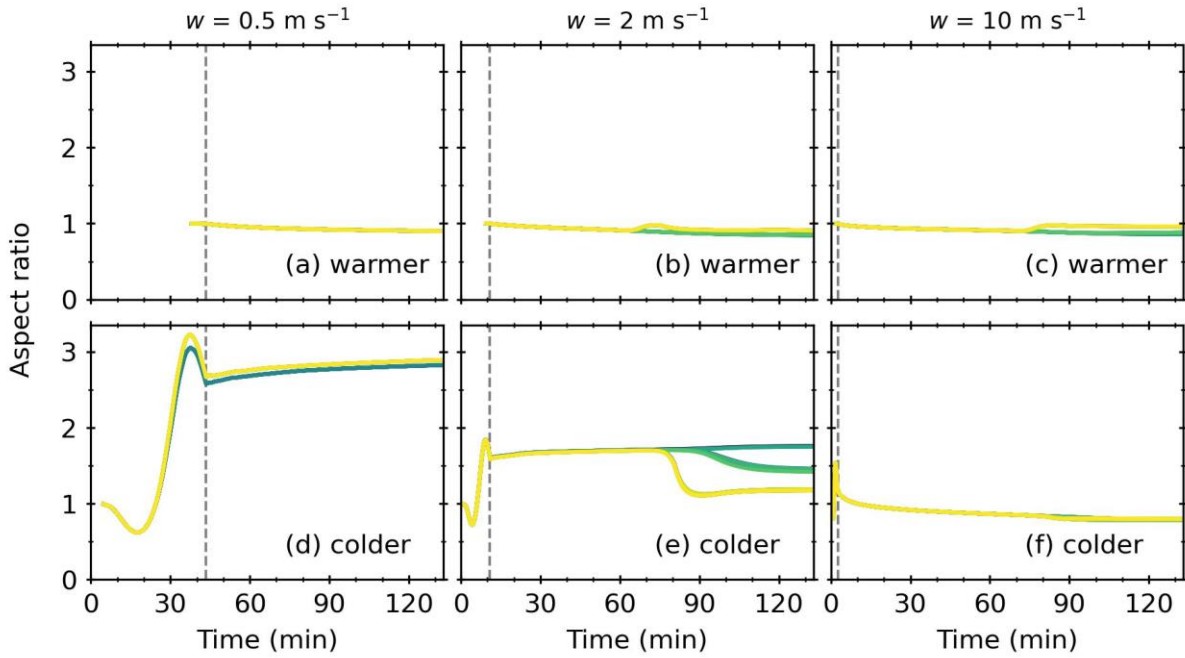

**Figure S22** Ice particle aspect ratio for a shallower cloud (1.3 km deep) with a near-city aerosol size distribution. Warmer refers to cloud base temperatures of 7 °C, and colder refers to cloud base temperatures of 0 °C.

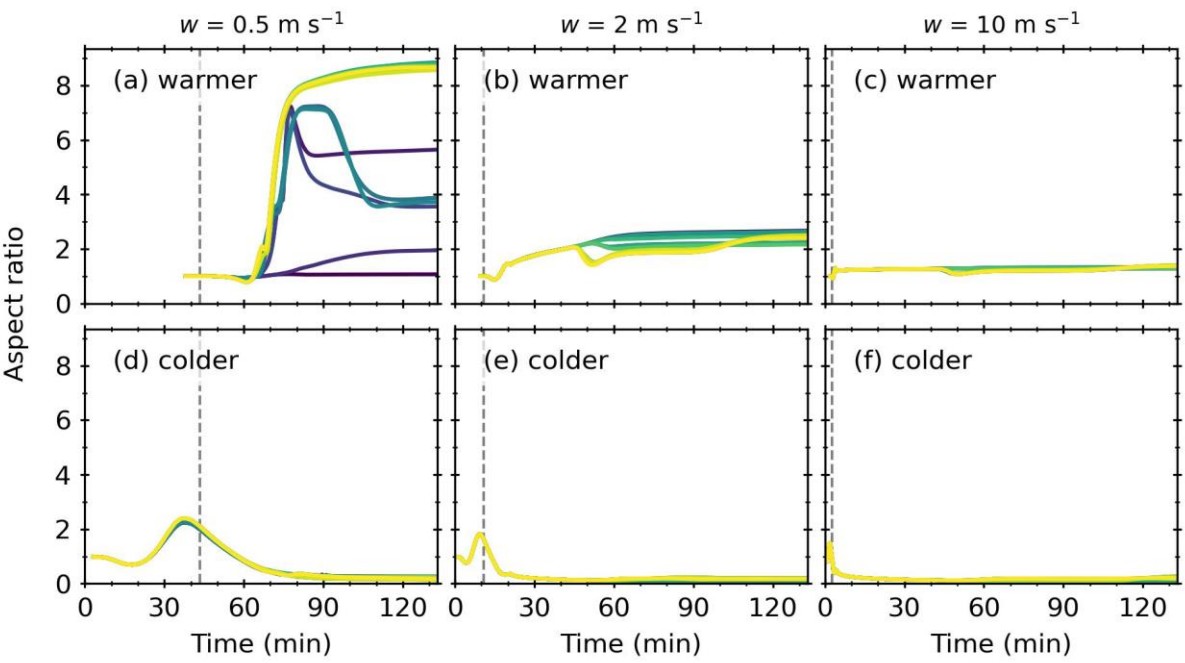

**Figure S23** Ice particle aspect ratio for a deeper cloud (2.4 km deep) with a natural aerosol size distribution. Warmer refers to cloud base temperatures of 7 °C, and colder refers to cloud base temperatures of 0 °C.

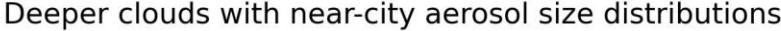
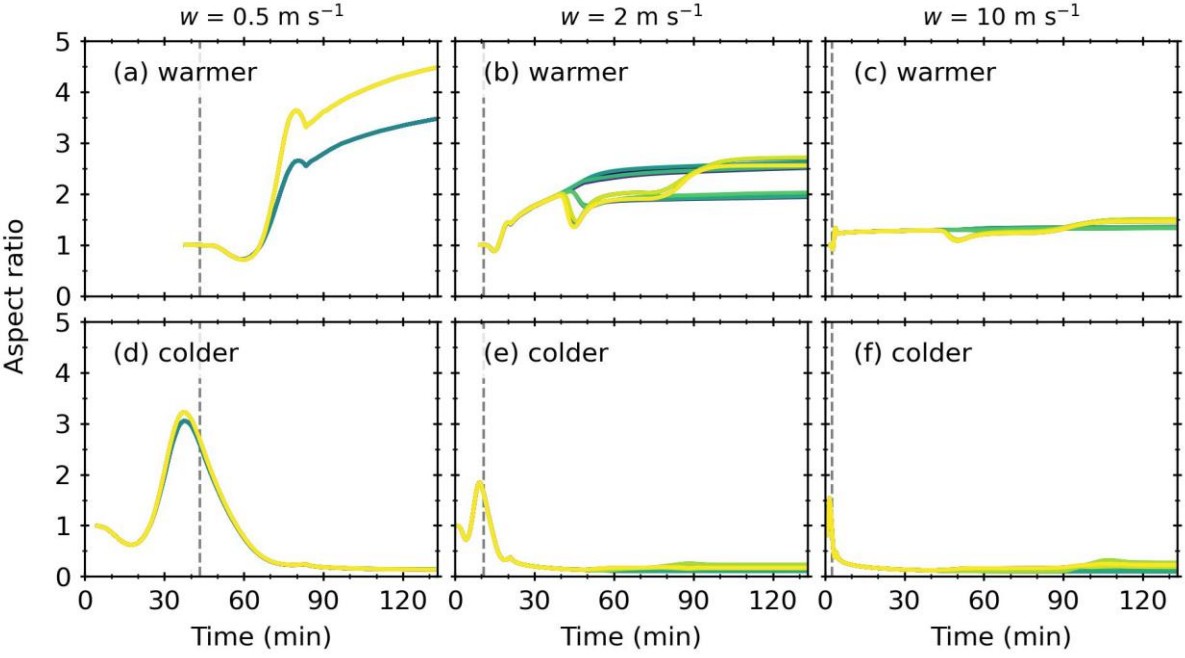

**Figure S24** Ice particle aspect ratio for a deeper cloud (2.4 km deep) with a near-city aerosol size distribution. Warmer refers to cloud base temperatures of 7 °C, and colder refers to cloud base temperatures of 0 °C.

# SECTION S8: Liquid and ice water contents

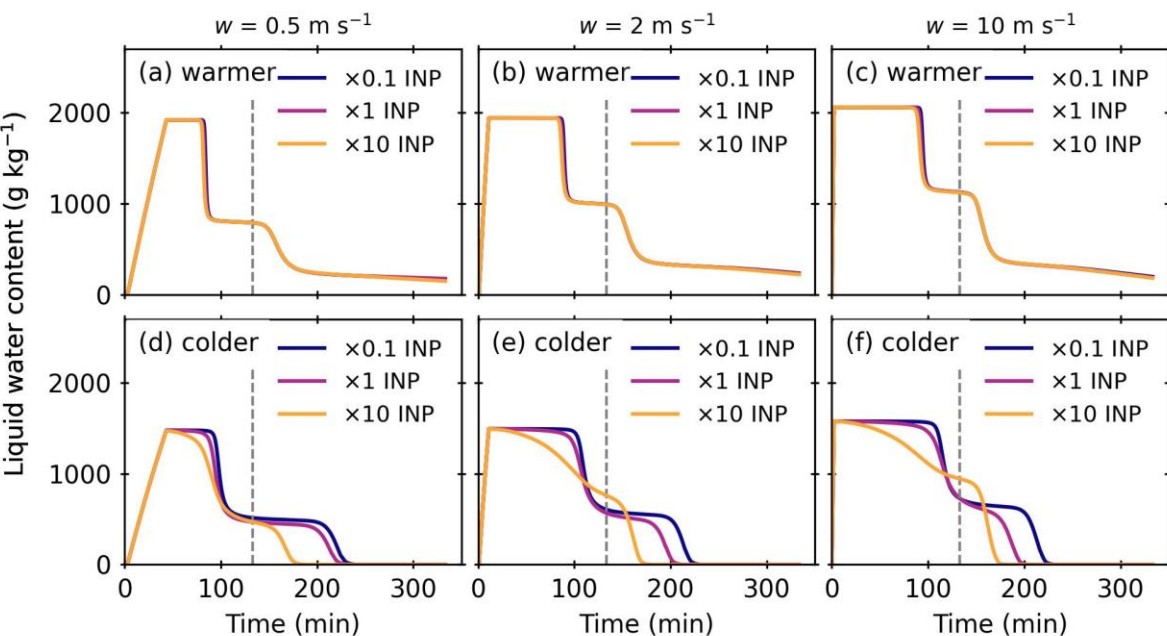

**Figure S25** Liquid water content for a shallower cloud (1.3 km deep) with a natural aerosol size distribution. Warmer refers to cloud base temperatures of 7 °C, and colder refers to cloud base temperatures of 0 °C. These simulations were extended past the 133.3 min runtime, indicated by a grey dashed line, to demonstrate the effects of the Wegener-Bergeron Findeisen process.

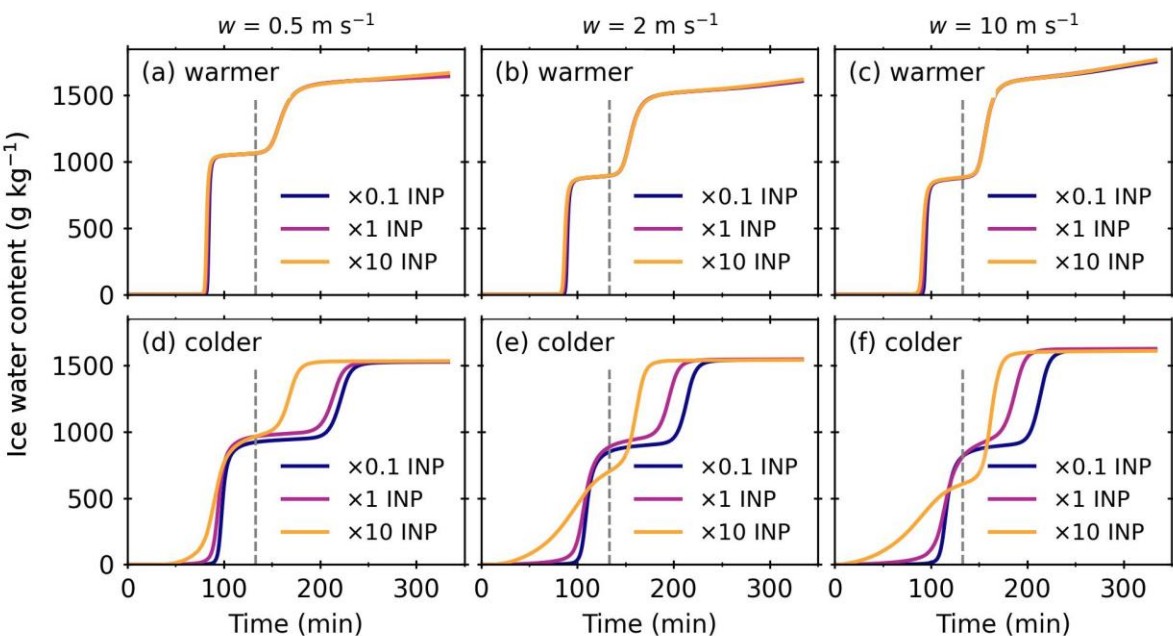

**Figure S26** Ice water content for a shallower cloud (1.3 km deep) with a natural aerosol size distribution. Warmer refers to cloud base temperatures of 7 °C, and colder refers to cloud base temperatures of 0 °C. These simulations were extended past the 133.3 min runtime, indicated by a grey dashed line, to demonstrate the effects of the Wegener-Bergeron Findeisen process.