# Peer review of "A bin-microphysics parcel model investigation of secondary ice formation in an idealised shallow convective cloud"

_Atmospheric Chemistry and Physics, 2022_

## Referee Comment (RC1)

**Reviewer comments on the manuscript „A bin-microphysics parcel model investigation of secondary ice formation in an idealised shallow convective cloud" by R. James et al.**

The secondary ice production (SIP) mechanisms receive exceedingly more attention in cloud research, as their importance in atmospheric ice formation becomes evident. There is, however, a strong disbalance between modelling efforts, field research, and laboratory experiments dedicated to the elucidation of SIP mechanisms, with the modelling studies clearly dominating the research landscape. Whereas in-situ aircraft-based research was recently gaining momentum thanks to new cloud particle imaging probes becoming available, the experimental efforts are scarce and are not yet capable of delivering physically meaningful and quantitative description of the SIP mechanisms so badly needed by cloud modelling community. Quite understandably, the SIP modelling scientists cannot wait for very slow advance of the state-of-the-art experimental activity and are therefore dependent on sieving the experimental data from past literature sources or venturing an extrapolation from recent but very preliminary experimental data.

The manuscript by R. James et al. reports on the modelling of various SIP mechanisms contributing to the enhancement of ice crystal number concentration (ICNC) in shallow convective clouds featuring cloud top temperature above -10°C. The main focus of the study lies in evaluation of the secondary ice formation upon collision between a rain droplet (diameter more than 150 μm) and an ice particle resulting in "splashing" of the droplet. In this mechanism, called "freezing fragmentation of droplets – mode 2" and subsequently referred to as "M2", the secondary water droplets produced in such a splashing event may contain fragments of ice crystals forming at the moment of droplet freezing initiation upon contact with ice. This mechanism was recently suggested on a basis of theoretical considerations in Phillips et al. (2018) and was not seriously considered being relevant for SIP, until the recent study of James et al. (2021) which produced some evidence of its potential importance. My comments are therefore mostly focused on the parametrization and implementation of this mechanism, because this is where my doubts are the strongest.

**General comments.**

**A.** According to (Phillips et al., 2018), the main criterium for M2 being active is the value of dimensionless energy $DE$ that has to be higher than the critical value $DE_{crit}$ (eqs. 7 to 9 of the manuscript). If this is the case, secondary droplets are produced upon collision of water droplet with solid ice surface (splashing condition). This criterium is accepted in the study as a basis for parametrization of secondary ice production rate according to M2 mechanism. (Phillips et al., 2018) derived this criterium analysing the results of droplet-droplet collision reported in (Low and List, 1982) and using the dimensionless collision energy formalism of (Testik et al., 2011), hinting at the absence of other laboratory data.

I seriously doubt that the critical value of $DE$ obtained by applying the formalism of (Testik et al., 2011) to the droplet-droplet collision data can be immediately used for parameterizing splashing due to droplet-solid surface collision. Actually, this is also not the most straightforward approach as there is quite a number of studies describing splashing of droplets upon collision with curved solid surfaces. In many studies, a combination of dimensionless numbers (Weber, *We*, and Ohnesorge, *Oh*) is used to describe the onset of splashing when the critical collision velocity is reached for the droplet of given diameter. For example, (Charalampous and Hardalupas, 2017) have found that splashing of water droplets colliding with a larger spherical target is initiated for *We* larger than 400, whereas this value is around 120 for droplets colliding with the flat surface. Sykes et al. (2022) has used a more complex splashing parameter based on a combination of kinetic energy of collision, surface energy, and viscosity, but even there splashing does not occur for We less than 300. Sykes et al. (2022) also contains a nice overview of previous experimental results on droplet-surface collisions and is definitely worth discussing. Apparently, many factors influence the splashing behaviour upon collision, including roughness of the solid surface, amount of water adsorbed on the surface, and the curvature of the surface. And yet, none of these factors are included into the M2 parameterization of (Phillips et al., 2018).

It is straightforward to compare the critical Weber number with the $DE_{crit}$ introduced by Phillips et al. (2018), because $DE$ is just the so called modified Weber number (the ratio of kinetic energy of collision to the surface energy):

$$We = \frac{\rho_w \cdot D_{drop} \cdot v_{col}^2}{\gamma_l}, DE = \frac{\rho_w \cdot D_{drop} \cdot v_{col}^2}{12 \cdot \gamma_l} = \frac{We}{12}$$

Here I take for simplicity that the droplet is much smaller than the ice particle and therefore the mass term in equation (7) is just the mass of the droplet $m_d$. If the droplet and the ice particle are approximately of the same mass, then $DE = \frac{We}{24}$. From here one can easily define the relationship between the droplet size $D_{drop}$ and the relative collision velocity $v_{col}$ required to initiate splashing (see Figure 1 below). One immediately notices that according to splashing criteria based on $We > 120$ (collision with flat surface, the red dash line) the relative collision velocity for a given droplet size must be almost 10x higher than the velocity estimated from the condition of $DE_{crit} = 0.2$ (the blue dash-dot line). It is even worse if the condition of $We > 400$ must be satisfied to initiate splashing. The argument that the splashing was observed in the experiments reported by James et al. (2021) remains valid, because the collision conditions were indeed well above the splashing limit (the solid diamond data point at $D_{drop} = 5\ mm$ and $v_{col} = 5.3\ m/s$).

[Figure]

**Figure 1. Splashing condition for droplet-solid surface collision.**

So, the central question becomes, would the M2 mechanism contribute to any secondary ice production if the condition of $We > 120$ or even $We > 400$ has to be fulfilled to ensure splashing upon collision? Could you extend the sensitivity study into this range or is it rather pointless because neither droplets nor ice crystals grow to the sizes required to fulfil this condition?

**B.** Examination of the Figure 1 brings some follow-up questions. If the less restrictive condition of $DE_{crit} > 0.2$ can be used, what are the collision velocities that can be achieved assuming the free fall of droplets and ice crystals? Since the bin model handles the collisions explicitly and the significant enhancement of ICNC due to M2 is achieved, the collisions are obviously efficient in the model; however, it would be very useful to show the distribution of sedimentation velocities for droplets and ice particles as a function of cloud evolution. I would be especially interested to see the sedimentation velocities of ice particles with the aspect ratios and perhaps even effective densities predicted by the ice growth model. A comparison of the modelled sedimentation velocities for both ice particles and rain droplets with the values observed in the real shallow convective clouds would also be very instructive.

**C.** I was surprised to learn that apparently, this study is not the first attempt to include M2 into the cloud modelling. As mentioned in the Discussion section (lines 556 to 569) Zhao et al. (2021) and others have modelled the contribution of M2 mechanism using essentially the same parameterization provided by

(Phillips et al., 2018). However, no significant contribution to SIP has been demonstrated. So much is clear from the text of this manuscript but the detailed discussion of the possible reasons is missing. Given that the M2 is in the focus of this study, I feel that the manuscript would strongly benefit from such discussion.

**D.** On a side note, a few experimental studies of the freezing fragmentation of droplets (M1 mode) under more realistic conditions of free fall has been published after 2018 (Kleinheins et al., 2021; Keinert et al., 2020; Lauber et al., 2018). It might be worth checking the parameterization of (Phillips et al., 2018) aginst the data of these recent studies.

**Summarizing**, I find the presented study a useful exercise in modelling the SIP contributions to ice enhancement in a shallow convective cloud and as such well worth publishing. Not being a part of modelling world myself, I can only presume that the model framework itself is flawless and can handle the standard microphysics properly, given the validity of underlying parameterizations. In case of M2, I have strong doubts that the splashing condition is fulfilled in a shallow convective cloud and that the M2 SIP mechanism will be important under the circumstances. I do not doubt, however, that if the splashing condition were fulfilled, the M2 would become a valid SIP mechanism, but the validity of parameterization should be convincingly shown, through experiments or via small-scale physical modelling of a collision-freezing event. One possible suggestion would be to choose the modelling case differently (deep convective cloud?) to reproduce these conditions, but this would be a different study. I feel this is a major critical point since the study is dedicated to the evaluation of the M2 mechanism and therefore request "major revisions".

**Minor comments:**

Line 73: Reference to (Hobbs and Rangno, 1990) is given twice.

Table 1: why is the effective density of *larger* colliding particle and the aspect ratio of the *smaller* colliding ice particle is given? Aren't both parameters for both colliding particles are important to calculate the relative collision velocity?

Equation (9) should include $\max(DE - DE_{crit}, 0)$ as in (Phillips et al., 2018), to avoid getting negative values for subcritical dimensionless energy (if $DE < DE_{crit}$).

Line 156: $f(T)$ is the mass FRACTION of water converted to ice after the end of the first freezing stage, not the "mass of frozen drop…".

Figure 4: the line colours for $\Phi$ values of 0.3 and 0.1 should be swapped. The way it is shown, the lower value of $\Phi$ results in a stronger ice enhancement.

Supplement, figures S17 to S20: The colour bar on the right supposedly gives the number of particles per unit volume; the units of the number concentration are not given.

**Cited literature**

Phillips, V. T. J., Patade, S., Gutierrez, J., and Bansemer, A.: Secondary Ice Production by Fragmentation of Freezing Drops: Formulation and Theory, Journal of the Atmospheric Sciences, 75, 3031-3070, 10.1175/jas-d-17-0190.1, 2018.

James, R. L., Phillips, V. T. J., and Connolly, P. J.: Secondary ice production during the break-up of freezing water drops on impact with ice particles, Atmos. Chem. Phys., 21, 18519-18530, 10.5194/acp-21-18519-2021, 2021.

Low, T. B. and List, R.: Collision, Coalescence and Breakup of Raindrops. Part I: Experimentally Established Coalescence Efficiencies and Fragment Size Distributions in Breakup, Journal of Atmospheric Sciences, 39, 1591-1606, 10.1175/1520-0469(1982)039<1591:Ccabor>2.0.Co;2, 1982.

Testik, F. Y., Barros, A. P., and Bliven, L. F.: Toward a Physical Characterization of Raindrop Collision Outcome Regimes, Journal of the Atmospheric Sciences, 68, 1097-1113, 10.1175/2010jas3706.1, 2011.

Charalampous, G. and Hardalupas, Y.: Collisions of droplets on spherical particles, Physics of Fluids, 29, 103305, 10.1063/1.5005124, 2017.

Sykes, T. C., Fudge, B. D., Quetzeri-Santiago, M. A., Castrejón-Pita, J. R., and Castrejón-Pita, A. A.: Droplet splashing on curved substrates, J. Colloid Interface Sci., 615, 227-235, https://doi.org/10.1016/j.jcis.2022.01.136, 2022.

Zhao, X., Liu, X., Phillips, V. T. J., and Patade, S.: Impacts of secondary ice production on Arctic mixed-phase clouds based on ARM observations and CAM6 single-column model simulations, Atmos. Chem. Phys., 21, 5685-5703, 10.5194/acp-21-5685-2021, 2021.

Kleinheins, J., Kiselev, A., Keinert, A., Kind, M., and Leisner, T.: Thermal imaging of freezing drizzle droplets: pressure release events as a source of secondary ice particles, Journal of the Atmospheric Sciences, 10.1175/jas-d-20-0323.1, 2021.

Keinert, A., Spannagel, D., Leisner, T., and Kiselev, A.: Secondary Ice Production upon Freezing of Freely Falling Drizzle Droplets, Journal of the Atmospheric Sciences, 77, 2959-2967, 10.1175/jas-d-20-0081.1, 2020.

Lauber, A., Kiselev, A., Pander, T., Handmann, P. V. K., and Leisner, T.: Secondary Ice Formation during Freezing of Levitated Droplets, Journal of the Atmospheric Sciences, 75, 2815-2826, 10.1175/jas-d-18-0052.1

Hobbs, P. V. and Rangno, A. L.: Rapid Development of High Ice Particle Concentrations in Small Polar Maritime Cumuliform Clouds, Journal of Atmospheric Sciences, 47, 2710-2722, 10.1175/1520-0469(1990)047<2710:Rdohip>2.0.Co;2, 1990.

---

## Referee Comment (RC2)

**Review of "A bin-microphysics parcel model investigation of secondary ice formation in an idealised shallow convective cloud" by James and colleagues**

**Verdict**

This paper is lucid and intriguing. Minor modifications to the text are needed before it is published.

**Major comments**

It would be a good idea to provide a precise definition of the meaning, values and units of all symbols in a table. This would clarify things when the text does not specify all the needed details.

Regarding the treatment of Mode 1 of raindrop-freezing fragmentation by Phillips et al. (2018), it should be emphasized in the introduction of the present paper by James et al. that numbers of secondary ice fragments were based only on lab observations of freezing drops in free-fall. No observations of electrodynamically levitated drops were used. Thus, Phillips et al. (2018) were vindicated, since it has since been shown that free-fall is crucial for fragmentation. Regarding the treatment of Mode 2, I think there is still wide experimental uncertainty introduced by the treatment of numbers of secondary drops emitted per impact with a larger ice surface. Sensitivity tests should be done to explore the realm of uncertainty. Anyway, raindrop-freezing fragmentation does not appear to be very important in most recent studies using the Phillips et al. (2018) scheme.

I think a more serious issue of the present paper is the lack of dependency on mean droplet size of the H-M process of rime-splintering. This was seen in lab studies by Hallett and Mossop, and was also consistent with aircraft observations of convective clouds by Harris-Hobbs and Cooper (1987) and Blyth et al. (1993). Yes, those studies involved artificial shattering biases on optical probes, but the correction merely introduces a systematic bias and the qualitative correlation with H-M conditions still likely applies. Can the simulations be re-done with a factor depending on mean droplet size somehow, including this in Eq (2), to account for the numbers of cloud-droplets > 24 microns ?

Thus, in view of the empirical uncertainty about splashing in Mode 2 of randrop-freezing fragmentation, it would be prudent for James et al. to perform sensitivity tests of their cloud simulations by raising the onset threshold of DE (0.2 currently) by a factor of 10 or 30. Also, sensitivity tests on reducing the constant of proportionality by an order of magnitude would also explore the range of experiment uncertainty. Error-bars reflecting this uncertainty from Mode 2 are needed on the control simulation plots of IE ratio.

**Comments about the Review by Kiselev about treatment of Mode 2**

I am grateful for the lucid comments about raindrop-freezing fragmentation during drop collisions with a larger ice particle (Mode 2) from the reviewer, Kiselev. It is illuminating to see papers published in colloidal science and fluid dynamics journals of relevance to splashing.

Curiously, in a sense, the two papers cited by the review by Kiselev qualitatively confirm the validity of the general theoretical approach for the treatment of Mode 2 by Phillips et al. (2018) because they show that the splashing onset condition is related to the Weber number.   This Weber number is identical to the dimensionless energy ("DE") of the 2018 scheme, except for a factor of 12.   So, the main import of Kiselev's criticism of the treatment of Mode 2 concerns uncertainty in the empirical estimation of parameters by Phillips et al. as applied in the present paper by James et al.

I would like to clarify one or two details about Kiselev's critique of the treatment of Mode 2 of raindrop-freezing fragmentation by Phillips et al. (2018).

First, observations of drop-drop collisions were not the only lab observations used to constrain the splashing scheme in the treatment of Mode 2.  There was also comparison with the observations by Levin and Hobbs (1971) of drops (2.5 mm, 4.2 m/s) falling on a rough copper hemisphere.  Levin and Hobbs observed about 150 splash-drops produced per collision.  They reported that there was onset of splashing at about 0.5 m/s impact speed.   Thus, the dimensionless energy (DE = initial collision kinetic energy divided by drop surface energy) per drop impact was $500 \, D \, v^2 \, /(6 \times 0.073)$ = 50 (for about 150 splash-droplets per collision) and 0.7 (for onset of secondary droplet emission or splashing).  This critical onset value of DE (0.7) is between one and two orders of magnitude lower than the threshold claimed by Kiselev in the review, who analyzed the Sykes et al. and Charalampous data.

Moreover, Levin and Hobbs observed that target roughness is crucial for the production of secondary drops, and presumably must have selected a rough copper sphere with the aim of better representing the actual roughness of ice precipitation particles, in the quest to study charge separation in clouds. Inspection of the results from Levin and Hobbs seem to suggest an approximate proportionality between number of secondary droplets and DE.  Intriguingly, Levin and Hobbs report observing that a "crown" of fluid would be produced on impact, with the secondary droplets being emitted from the crown (e.g. by jets). On very smooth surfaces, they observed that there would be no crown, and presumably little or no secondary droplet emission (no splashing).

Thus, although Phillips et al. (2018) in the text cited the drop-drop collisions as the source of the critical DE value (0.2), their simultaneous use of the Levin and Hobbs observations of drops on a rough copper hemisphere also approximately support this value (0.7, the same order of magnitude as 0.2).  In fact, Phillips et al. used chiefly the Levin and Hobbs results for the most important parameter of the scheme, namely the coefficient of proportionality between drop number and excess DE, rather than the drop-drop collision data.

Secondly, the review by Kiselev claims that factors of surface roughness, water adsorbed onto the surface and curvature of the surface were not accounted for in the treatment by Phillips et al. (2018).   Is this claim true ?   Yes and no.   Naturally, these factors are greatly sensitive, as observed by Levin and Hobbs for roughness, as noted above.  Yet, Levin and Hobbs knew that these factors were influential and designed their copper sphere experiment so as to be representative for collisions with ice precipitation particles in natural clouds (their copper sphere was dry, as with ice during dry growth by riming, and rough, as is true of most riming;  their ratio of the radii of curvature between the incident drop and target was about 10%, a plausible order of magnitude for Mode 2 collisions in natural clouds).  Their focus was on charge separation in electrified clouds.  So the average conditions of these three factors may be implicitly factored into the Levin and Hobbs results and hence into the Phillips et al. (2018) scheme.

By contrast, the two laboratory studies cited by Kiselev involve observations of extremely smooth spherical surfaces. Sykes et al. (2022) used "smooth untreated borosilicate-glass substrates" while Charalampous and Hardalupas (2017) used similarly smooth glass (smoothness of 35 nm). While such studies are theoretically illuminating (they nicely demonstrate that the ratio of drop to target radius of curvature is a sensitive parameter rather than the target radius per se), the observations of Levin and Hobbs about the crucial role of target roughness for the splashing process indicates that they cannot be used quantitatively for constraining Mode 2.

Finally, Schremb et al. (2017) observed splash-droplets from drops of 3 mm in diameter falling at about 2 m/sec on a larger ice target, with a reported Weber number of about 200 (DE value of about 15). Although they did not report splash-drop numbers, analysis of the published photos in their paper and my personal communication with Schremb shows that dozens of splash-drops were emitted per collision. This suggests that for ice, the critical DE value for onset of splashing must be much less than about 15, contrary to the lab results cited by the review. Perhaps onset of splashing in Schremb's experiment would be expected to have occurred at a critical DE value of about unity, given the proportionality noted above inferred from the Levin and Hobbs results.

On the one hand, the Mode 2 treatment by Phillips et al. might have been an over-estimate, since Levin and Hobbs (1971) observed 10 times more splash-drops for a rough curved copper surface than for similar values of drop size, DE and Weber number compared to Schremb et al. with a larger ice surface (a vertical columnar shape). Alternatively, one could argue that the roughness of the surface is more realistic in the case of Levin and Hobbs than for Schremb et al., such that Schremb et al. may have observed too few (unreported) splash-droplets per collision if their ice surface was smoother than for ice precipitation in natural clouds. There is much uncertainty still, but roughness appears to be crucial.

**In summary, there seems to be variability of the splashing among drops among different types of larger solid surface for different lab experiments, even for similar macroscopic dynamics of collision. The observations by Levin and Hobbs of drops falling on a rough copper sphere were designed for representativeness when studying charging in collision of ice particles in natural clouds. They found target roughness is crucial for secondary droplet emission (and for splashing in Mode 2). Ice precipitation in natural clouds is almost instantly roughened by riming. Hence, the smooth-glass-target observations cited in the review by Kiselev, though useful theoretically for the physics of splashing, must greatly under-estimate the real splashing in natural clouds during Mode 2 by ice precipitation. The smooth-glass-target observations cannot be supposed to be more reliable than observations by Levin and Hobbs.**

**Of course, future lab experiments on this topic will be invaluable to elucidate Mode 2 of raindrop-freezing fragmentation in a representative manner.**

**Specific comments**

Line 90: More details of the model description are needed in the text. What are the microphysical species represented ? What processes of growth are treated ? Is raindrop-freezing treated ?

Line 120: Why is the observed dependency on cloud-droplet size not represented in Eq (2) ? Drops larger than 24 microns are needed for rime-splintering, as shown by Hallett and Mossop's later work.

One can simply multiply the formula shown in Eq (2) by an extra factor that increases with mean droplet size from zero to unity over a certain range.

Line 148:  It might be good to clarify that Mode 2 is for raindrops.

Line 155:  Need to say that NM2 is the number of fragments per drop accreted.

Line 167:  It is unclear what is meant here: "We ran all possible combinations using the methodology described by Montgomery (2013); Teller and Levin (2008). For k factors there are $2^k$ different combinations".  What is k ?  What factors are being mentioned ?  What combinations are being mentioned ?

Line 170: as noted above, there is a need to perform sensitivity tests with respect to the number of splash droplets per collision predicted for Mode 2 of raindrop-freezing fragmentation.  This is due to the experimental uncertainty.

Line 180:  There is over-loading of the terms "Mode 1" and "Mode 2" in Table 3 to refer to aerosol modes.  Can a different label be used for aerosol modes (e.g., Modes A, B, C) ?

Line 248:  A problem with the DeMott (2010) parameterization is that there is no dependency on aerosol chemistry and that on size is only represented in a basic way, using a threshold.  There is no direct dependency on dust surface area.  If the aerosol chemical composition was observed for the simulated cases, then other schemes are possible (e.g. Phillips et al. 2008, 2013).

Line 298:  the plural of "maximum" is "maxima", as it is a Latin word.

Figure 5:  I think it would be wise to include an error-bar in the simulations arising from errors in the parameters of Mode 2, in view of the discussions above and by Kiselev's review.   Also, a logarithmic y-axis would be more appropriate in view of the uncertainty.

Line 568:  the sentence does not make grammatical sense.  Did Sotiropolou et al. explicitly treat Mode 2 ?

Line 578:  Need to explain the mechanism for how high IN concentrations in the model suppress SIP and coalescence.  Do they cause subsaturation with respect to liquid, from vapour growth of ice, and is this what curbs the droplet growth, inhibiting raindrop-freezing fragmentation ?  Or is the liquid depleted by riming of the primary ice ?

As argued by Yano and Phillips (2011), there is an upper limit to the ice concentration that depends on vertical velocity and temperature, corresponding to the onset of subsaturation with respect to liquid. When the cloud-liquid evaporates and the cloud becomes ice-only, all SIP ceases.   Waman et al. (2021) nicely showed this happening, so that paper needs to be cited.   Could the lack of SIP when INPs are numerous be due to this upper ceiling being reached by the primary ice, or even approached ?

---

## Author Comment (AC1)

**AC Response to RC1**

We thank Alexei for taking the time to review our manuscript and appreciate the constructive feedback provided.

The secondary ice production (SIP) mechanisms receive exceedingly more attention in cloud research, as their importance in atmospheric ice formation becomes evident. There is, however, a strong disbalance between modelling efforts, field research, and laboratory experiments dedicated to the elucidation of SIP mechanisms, with the modelling studies clearly dominating the research landscape. Whereas in-situ aircraft-based research was recently gaining momentum thanks to new cloud particle imaging probes becoming available, the experimental efforts are scarce and are not yet capable of delivering physically meaningful and quantitative description of the SIP mechanisms so badly needed by cloud modelling community. Quite understandably, the SIP modelling scientists cannot wait for very slow advance of the state-of-the-art experimental activity and are therefore dependent on sieving the experimental data from past literature sources or venturing an extrapolation from recent but very preliminary experimental data.

The manuscript by R. James et al. reports on the modelling of various SIP mechanisms contributing to the enhancement of ice crystal number concentration (ICNC) in shallow convective clouds featuring cloud top temperature above -10°C. The main focus of the study lies in evaluation of the secondary ice formation upon collision between a rain droplet (diameter more than 150 μm) and an ice particle resulting in "splashing" of the droplet. In this mechanism, called "freezing fragmentation of droplets – mode 2" and subsequently referred to as "M2", the secondary water droplets produced in such a splashing event may contain fragments of ice crystals forming at the moment of droplet freezing initiation upon contact with ice. This mechanism was recently suggested on a basis of theoretical considerations in Phillips et al. (2018) and was not seriously considered being relevant for SIP, until the recent study of James et al. (2021) which produced some evidence of its potential importance. My comments are therefore mostly focused on the parametrization and implementation of this mechanism, because this is where my doubts are the strongest.

**General comments.**

**RC: A.** According to (Phillips et al., 2018), the main criterium for M2 being active is the value of dimensionless energy $DE$ that has to be higher than the critical value $DE_{\text{crit}}$ (eqs. 7 to 9 of the manuscript). If this is the case, secondary droplets are produced upon collision of water droplet with solid ice surface (splashing condition). This criterium is accepted in the study as a basis for parametrization of secondary ice production rate according to M2 mechanism. (Phillips et al., 2018) derived this criterium analysing the results of droplet-droplet collision reported in (Low and List, 1982) and using the dimensionless collision energy formalism of (Testik et al., 2011), hinting at the absence of other laboratory data.

I seriously doubt that the critical value of $DE$ obtained by applying the formalism of (Testik et al., 2011) to the droplet-droplet collision data can be immediately used for parameterizing splashing due to droplet-solid surface collision.

**AC:** As stated in Phillips et al. (2018), observations of drops on rough copper hemispheres from Levin and Hobbs (1971) were also used to constrain the Mode 2 splashing parameterisation and the available

data from Levin and Hobbs (1971) also fit the parameterisation. Admittedly there are not many data points, but the data we have fit the parameterisation.

RC: Actually, this is also not the most straightforward approach as there is quite a number of studies describing splashing of droplets upon collision with curved solid surfaces. In many studies, a combination of dimensionless numbers (Weber, *We*, and Ohnesorge, *Oh*) is used to describe the onset of splashing when the critical collision velocity is reached for the droplet of given diameter. For example, (Charalampous and Hardalupas, 2017) have found that splashing of water droplets colliding with a larger spherical target is initiated for *We* larger than 400, whereas this value is around 120 for droplets colliding with the flat surface. Sykes et al. (2022) has used a more complex splashing parameter based on a combination of kinetic energy of collision, surface energy, and viscosity, but even there splashing does not occur for We less than 300.

AC: Both Charalampous and Hardalupas (2017) and Sykes et al. (2022) show that the splashing onset is related to the Weber number. The dimensionless energy (DE) is related to the Weber number as shown below except by a factor of 12. The results from Charalampous and Hardalupas (2017) and Sykes et al. (2022) therefore validate the general theoretical approach used in Phillips et al. (2018) for parameterising Mode 2. Charalampous and Hardalupas (2017) have not formulated an SIP parameterisation, so we have resorted to using the Phillips et al. (2018) formulation.

RC: Sykes et al. (2022) also contains a nice overview of previous experimental results on droplet-surface collisions and is definitely worth discussing. Apparently, many factors influence the splashing behaviour upon collision, including roughness of the solid surface, amount of water adsorbed on the surface, and the curvature of the surface. And yet, none of these factors are included into the M2 parameterization of (Phillips et al., 2018).

AC: We agree that roughness of the solid surface ought to affect the splashing behaviour. Roughness is not included in the M2 parameterisation, but then again there are no cloud microphysical models, that we are aware of, that calculate the roughness of the ice particle surfaces. Therefore, it does not help to include roughness in the parameterisation.

RC: It is straightforward to compare the critical Weber number with the $DE$ introduced by Phillips et al. (2018), because *DE* is just the so called modified Weber number (the ratio of kinetic energy of collision to the surface energy):

$$We = \frac{\rho_w \cdot D_{drop} \cdot v_{col}^2}{\gamma_l}, DE = \frac{\rho_w \cdot D_{drop} \cdot v_{col}^2}{12 \cdot \gamma_l} = \frac{We}{12}$$

Here I take for simplicity that the droplet is much smaller than the ice particle and therefore the mass term in equation (7) is just the mass of the droplet $m_d$. If the droplet and the ice particle are approximately of the same mass, then $DE = \frac{We}{24}$. From here one can easily define the relationship between the droplet size $D_{drop}$ and the relative collision velocity $v_{col}$ required to initiate splashing (see Figure 1 below). One immediately notices that according to splashing criteria based on $We > 120$ (collision with flat surface, the red dash line) the relative collision velocity for a given droplet size must be almost 10x higher than the velocity estimated from the condition of $DE_{crit} = 0.2$ (the blue dash-dot line). It is even worse if the condition of $We > 400$ must be satisfied to initiate splashing. The argument that the splashing was observed in the experiments reported by James et al. (2021) remains valid,

because the collision conditions were indeed well above the splashing limit (the solid diamond data point at $D_{drop} = 5\ mm$ and $v_{col} = 5.3\ m/s$).

[Figure]

**Figure 1. Splashing condition for droplet-solid surface collision.**

So, the central question becomes, would the M2 mechanism contribute to any secondary ice production if the condition of $We > 120$ or even $We > 400$ has to be fulfilled to ensure splashing upon collision? Could you extend the sensitivity study into this range or is it rather pointless because neither droplets nor ice crystals grow to the sizes required to fulfil this condition?

**AC:** Data from Levin and Hobbs (1971) was used to constrain the M2 parameterisation of Phillips et al. (2018). Levin and Hobbs (1971) specifically designed their experiments to represent cloud conditions. The amount of water adsorbed on the surface and roughness were accounted for by using a dry, roughened copper surface and the curvature between the drop and target had a ratio of 10 %, not untypical for clouds. These results are implicitly factored into the Mode 2 parameterisation of Phillips et al. (2018).

The issue with saying that We > 300 or 400 must be fulfilled to ensure splashing are the experimental conditions in which this value was determined. Charalampous and Hardalupas (2017) used smooth particles composed of glass to reduce the *'influence of the surface roughness… thus reducing the complexity of the interpretation of the measurements'*. Sykes et al. (2022) used *'smooth untreated borosilicate-glass substrates'*. Ice in clouds is unlikely to be as smooth as the substrates in Charalampous and Hardalupas (2017) and Sykes et al. (2022) and there is no reason to take their critical Weber numbers over the roughened substrates of Levin and Hobbs (1971).

However, we do accept that there is large uncertainty in the DE$_{crit}$ value, which is influenced by all of the factors mentioned above and have performed sensitivity studies on the DEcrit value by increasing it by a factor of 10 and 30. The results are given in Sections 4.1.4 and 4.2.4 for a natural and near-city aerosol size distribution respectively.

**4.1.4 Sensitivity test: M2 DE$_{Crit}$**

The M2 parameterisation given in Eq. 9 requires the onset of splashing, denoted as DE$_{crit}$. We used the value of 0.2 given in Phillips et al. (2018) based on laboratory data of drops colliding on roughened copper hemispheres from Levin et al. (1971). To test the sensitivity of ice enhancement to DE$_{crit}$ during M2 simulations, we ran simulations for warmer and colder cloud bases with natural aerosol size distribution and updraft speeds of 2 m s$^{-1}$ using the following DE$_{crit}$ values: 0.2, 3 & 6. The results are plotted in Figure 5.

[Figure]

**Figure 5.** M2 ice enhancement against simulation time for three DE$_{crit}$ values (0.2, 3 and 6) for a shallower (1.3 km) cloud with natural aerosol size distributions and updraft speed of 2 m s$^{-1}$. Warmer refers to cloud base temperatures of 7 °C, and colder refers to cloud base temperatures of 0 °C.

**4.2.4   Sensitivity test: M2 DE$_{Crit}$**

Figure 9 shows the M2 ice enhancement against simulation time for three DE$_{crit}$ values, 0.2, 3 and 6, for shallower clouds with a near-city aerosol size distribution. In general, lower values of DE$_{crit}$ had earlier maximum ice enhancement peaks compared to higher values of DE$_{crit}$. The maximum ice enhancement peaks were greater for lower DE$_{crit}$ values.

[Figure]

**Figure 9.** M2 ice enhancement against simulation time for three DE$_{crit}$ values (0.2, 3 and 6) for a shallower (1.3 km) cloud with near-city aerosol size distributions and updraft speed of 2 m s$^{-1}$. Warmer refers to cloud base temperatures of 7 °C, and colder refers to cloud base temperatures of 0 °C.

We have also expanded the discussion of DE$_{crit}$ to the following:

Other uncertainties in the M2 parameterisation include the critical value for onset of splashing ($DE_{cit}$). For splashing to occur, the dimensionless energy must be greater than $DE_{crit}$. Phillips et al. (2018) estimated a value for $DE_{crit}$ of ~0.2 based on room temperature experiments of colliding drops by Low and List (1982) and acknowledged that this was a source of uncertainty. The equation used to describe number of secondary liquid drops was further constrained using laboratory experiments by Levin et al. (1971). These experiments consisted of 2.5 mm drops impacting on a rough copper sphere at room temperature. The number of secondary drops formed in collision experiments is sensitive both to the geometry and impact material; hence the expression used to describe the number of liquid fragments presents another source of uncertainty. We increased the $DE_{crit}$ by a factor of 10 and 30 to demonstrate the sensitivity of M2 ice enhancement on the DE$_{crit}$, as

shown in Figs. 5 and 9. When the $DE_{crit}$ was increased by a factor of 10 or 30, the ice enhancement decreased compared to the stated $DE_{crit}$ in Phillips et al. (2018), but ice enhancement still occurred. Despite the uncertainties within the M2 parameterisation, and in agreement with Phillips et al. (2018), it is more erroneous not to include M2 within simulations. Our laboratory experiments strongly suggest that the M2 SIP mechanism could be active (James et al., 2021), and further laboratory studies into this mechanism will help reduce the uncertainties listed.

**B.** Examination of the Figure 1 brings some follow-up questions. If the less restrictive condition of $DE$ > 0.2 can be used, what are the collision velocities that can be achieved assuming the free fall of droplets and ice crystals? Since the bin model handles the collisions explicitly and the significant enhancement of ICNC due to M2 is achieved, the collisions are obviously efficient in the model; however, it would be very useful to show the distribution of sedimentation velocities for droplets and ice particles as a function of cloud evolution. I would be especially interested to see the sedimentation velocities of ice particles with the aspect ratios and perhaps even effective densities predicted by the ice growth model. A comparison of the modelled sedimentation velocities for both ice particles and rain droplets with the values observed in the real shallow convective clouds would also be very instructive.

**AC:** Although the bin model treats the collisional speeds, it has not been output in the model simulations and it is not straightforward to us how to display this. We also did not observe sedimentation velocities in the real clouds so do not feel this would be particularly instructive. We used Pruppacher and Klett (2010) to calculate sedimentation speeds of droplets over the whole range. The main aim of the paper was to try to show, given what we currently know about SIP, how does the 'mode 2' parameterisation behave in a cloud model simulation compared to other SIP mechanisms. We feel that the findings in the paper show that it is potentially very important, and that it warrants further quantification of the mechanism. There are outstanding questions, which are yet to be addressed, but can be the subject of further study.

**C.** I was surprised to learn that apparently, this study is not the first attempt to include M2 into the cloud modelling. As mentioned in the Discussion section (lines 556 to 569) Zhao et al. (2021) and others have modelled the contribution of M2 mechanism using essentially the same parameterization provided by (Phillips et al., 2018). However, no significant contribution to SIP has been demonstrated. So much is clear from the text of this manuscript but the detailed discussion of the possible reasons is missing.

Given that the M2 is in the focus of this study, I feel that the manuscript would strongly benefit from such discussion.

**AC:** The original phrasing of the paragraph gave the misrepresentation that M2 did not significantly contribute to SIP. We have rephrased so that it is clear that for some studies M2 did contribute significantly, but M1 may have had a larger contribution. We have also expanded the discussion to include reasons given from the studies at to why M1+M2 were not significant.

Several studies have included M2 in their simulations, although usually combined with M1 (Phillips et al., 2018; Sotiropoulou et al., 2020; Qu et al., 2020; Zhao et al., 2021; Zhao and Liu, 2021; Georgakaki et al., 2022; Zhao and Liu, 2022; Huang et al., 2022; Karalis et al., 2022) which is equivalent to our M1+M2 simulations. For our idealised cloud conditions, the contribution
560 from our M1+M2 simulations appears to be derived from the M2 aspect. In fact it is only when we combine M1 with M2 that we see any significant ice enhancement from M1. On its own, M1 was not a strong ice enhancement mechanism in our idealised shallow convective clouds. In contrast, Phillips et al. (2018) (see their Fig. 6) showed that in their simulation of a deep tropical maritime convective cloud with temperatures between 0 to -20 °C, over 90 % of the ice came from M1 and M2, of which M1 contributed to approximately double that of M2. M1 was more significant in Phillips et al. (2018) simulations
565 compared to M2, probably due to simulation temperatures which covered the thermal peak of M1 at -15 °C. As shown in Fig. 10, very few fragments were formed away from the thermal peak. Most of our simulations were warmer than the thermal peak and the deeper clouds with colder cloud base temperatures of 0 °C only briefly had temperatures near the thermal peak early on in the simulation (see Fig. S2(b) of the Supplementary). Similar results were also observed by Qu et al. (2020) who modelled similar deep tropical maritime convective cloud conditions that Phillips et al. (2018) simulated. Zhao et al. (2021)
570 simulated four types of Arctic mixed-phase clouds based on Atmospheric Radiation Measurement Mixed-Phase Arctic Cloud Experiment observations. They showed that approximately 80 % of all ice particles came from M1 in their single-layer bound-ary layer stratus simulations. The largest contribution of M2 came from the transition simulations but this only contributed a small fraction compared to M1. Other studies which modelled the M2 SIP mechanism with the M1 SIP mechanism did not provide a breakdown of the contribution from M1 and M2 (Zhao and Liu, 2021, 2022; Huang et al., 2022; Karalis et al., 2022).
575 Two studies found that M2 combined with M1 was not an effective SIP mechanism in the simulated conditions (Sotiropoulou et al., 2020; Georgakaki et al., 2022). Sotiropoulou et al. (2020) stated that this was due to their thermodynamic conditions with relatively cold cloud base temperatures and Georgakaki et al. (2022) stated that this was due to the drops being too small to initiate M1 + M2 in their simulated conditions.

**D.** On a side note, a few experimental studies of the freezing fragmentation of droplets (M1 mode) under more realistic conditions of free fall has been published after 2018 (Kleinheins et al., 2021; Keinert et al., 2020; Lauber et al., 2018). It might be worth checking the parameterization of (Phillips et al., 2018) against the data of these recent studies.

**AC:** We acknowledge that there have been several interesting studies on the freezing fragmentation of droplet (M1). However, the selection criteria of the laboratory data used in the Phillips et al. (2018) mode 1 parameterisation specifically states that it only included studies where the drops were in free fall. Thus excluding Keinert et al. (2020) and Lauber et al. (2018), both of whom used an electrodynamic balance to levitate drops. We note that Keinert et al. (2020) simulate free fall by passing an airflow around the drops. We also note that Keinert et al. (2021) shows that compared to Lauber et al. (2018), having an air flow around the levitated drop clearly enhances breakup. Therefore, checking the parameterisation of these new studies against Phillips et al. (2018) mode 1 parameterisation needs to be done carefully, and we feel this is outside the scope of our paper.

Kleinheins et al. (2021) present pressure release events rather than number of SIP particles because it is difficult to see small splinters visually. However, they find 3x more pressure release events than ejected particles (counted visually in tKeinert et al. [2020]). Keinert et al. (2020) found that typically a thermal peak at -15 °C occurred where there were around 1-2 events per droplet. The number of small splinters from the Phillips et al. (2018) parameterisation for the same-sized drops is shown below. This has a peak at -15 °C of around 6-7, which is the same order of magnitude as Kleinheins et al. (2021); however, Kleinheins et al. (2021) showed that the peak was not centred on -15 °C. Although there are differences, it is not clear which is correct – as acknowledged by Kleinheins et al. (2021), the number of pressure release events do not necessarily equate to the number of SIP particles for instance. Given the uncertainty we feel it is justified to use the parameterisation and comment on the state of the science in the paper.

[Figure]

**Summarizing**, I find the presented study a useful exercise in modelling the SIP contributions to ice enhancement in a shallow convective cloud and as such well worth publishing. Not being a part of modelling world myself, I can only presume that the model framework itself is flawless and can handle the standard microphysics properly, given the validity of underlying parameterizations. In case of M2, I have strong doubts that the splashing condition is fulfilled in a shallow convective cloud and that the M2 SIP mechanism will be important under the circumstances. I do not doubt, however, that if the splashing condition were fulfilled, the M2 would become a valid SIP mechanism, but the validity of parameterization should be convincingly shown, through experiments or via small-scale physical modelling of a collision-freezing event. One possible suggestion would be to choose the modelling case differently (deep convective cloud?) to reproduce these conditions, but this would be a different study. I feel this is a major critical point since the study is dedicated to the evaluation of the M2 mechanism and therefore request "major revisions".

**AC:** See above responses.

**Minor comments:**

**RC:** Line 73: Reference to (Hobbs and Rangno, 1990) is given twice.
**AC:** Removed.

**RC:** Table 1: why is the effective density of *larger* colliding particle and the aspect ratio of the *smaller* colliding ice particle is given? Aren't both parameters for both colliding particles are important to calculate the relative collision velocity?

**AC:** It is not the relative collision velocity that is being calculated here. Relative collision velocity is calculated from the difference in sedimentation speeds in the model, which depends on the density and aspect ratio as mentioned. Table 1 is used to identify the parameters of the surface of the ice category, which go into equation 13 of Phillips et al (2017). It is to be used in conjunction with Table 1 of Phillips et al (2017)

**RC:** Equation (9) should include max $(DE - DE\ , 0)$ as in (Phillips et al., 2018), to avoid getting negative values for subcritical dimensionless energy (if $DE < DE\ $).
**AC:** Changed.

**RC:** Line 156: $f(T)$ is the mass FRACTION of water converted to ice after the end of the first freezing stage, not the "mass of frozen drop...".
**AC:** Changed.

**RC:** Figure 4: the line colours for $\Phi$ values of 0.3 and 0.1 should be swapped. The way it is shown, the lower value of $\Phi$ results in a stronger ice enhancement.
**AC:** Changed.

**RC:** Supplement, figures S17 to S20: The colour bar on the right supposedly gives the number of particles per unit volume; the units of the number concentration are not given.
**AC:** Added units.

**Cited literature**

Phillips, V. T. J., Patade, S., Gutierrez, J., and Bansemer, A.: Secondary Ice Production by Fragmentation of Freezing Drops: Formulation and Theory, Journal of the Atmospheric Sciences, 75, 3031-3070, 10.1175/jas-d-17-0190.1, 2018.

James, R. L., Phillips, V. T. J., and Connolly, P. J.: Secondary ice production during the break-up of freezing water drops on impact with ice particles, Atmos. Chem. Phys., 21, 18519-18530, 10.5194/acp-21-185192021, 2021.

Low, T. B. and List, R.: Collision, Coalescence and Breakup of Raindrops. Part I: Experimentally Established Coalescence Efficiencies and Fragment Size Distributions in Breakup, Journal of Atmospheric Sciences, 39, 1591-1606, 10.1175/1520-0469(1982)039<1591:Ccabor>2.0.Co;2, 1982.

Testik, F. Y., Barros, A. P., and Bliven, L. F.: Toward a Physical Characterization of Raindrop Collision Outcome Regimes, Journal of the Atmospheric Sciences, 68, 1097-1113, 10.1175/2010jas3706.1, 2011.

Charalampous, G. and Hardalupas, Y.: Collisions of droplets on spherical particles, Physics of Fluids, 29, 103305, 10.1063/1.5005124, 2017.

Sykes, T. C., Fudge, B. D., Quetzeri-Santiago, M. A., Castrejón-Pita, J. R., and Castrejón-Pita, A. A.: Droplet splashing on curved substrates, J. Colloid Interface Sci., 615, 227-235, https://doi.org/10.1016/j.jcis.2022.01.136, 2022.

Zhao, X., Liu, X., Phillips, V. T. J., and Patade, S.: Impacts of secondary ice production on Arctic mixedphase clouds based on ARM observations and CAM6 single-column model simulations, Atmos. Chem. Phys., 21, 5685-5703, 10.5194/acp-21-5685-2021, 2021.

Kleinheins, J., Kiselev, A., Keinert, A., Kind, M., and Leisner, T.: Thermal imaging of freezing drizzle droplets: pressure release events as a source of secondary ice particles, Journal of the Atmospheric Sciences, 10.1175/jas-d-20-0323.1, 2021.

Keinert, A., Spannagel, D., Leisner, T., and Kiselev, A.: Secondary Ice Production upon Freezing of Freely Falling Drizzle Droplets, Journal of the Atmospheric Sciences, 77, 2959-2967, 10.1175/jas-d-20-0081.1, 2020.

Lauber, A., Kiselev, A., Pander, T., Handmann, P. V. K., and Leisner, T.: Secondary Ice Formation during Freezing of Levitated Droplets, Journal of the Atmospheric Sciences, 75, 2815-2826, 10.1175/jas-d-180052.1

Hobbs, P. V. and Rangno, A. L.: Rapid Development of High Ice Particle Concentrations in Small Polar Maritime Cumuliform Clouds, Journal of Atmospheric Sciences, 47, 2710-2722, 10.1175/15200469(1990)047<2710:Rdohip>2.0.Co;2, 1990.

**Additional Cited References**

Pruppacher, H.R. and Klett, J.D. 2010. *Microphysics of clouds and precipitation*. Dordrecht ;: Springer.

---

## Author Comment (AC2)

**Review of "A bin-microphysics parcel model investigation of secondary ice formation in an idealised shallow convective cloud" by James and colleagues**

We thank the referee for their time in reviewing our manuscript and appreciate the constructive feedback provided.

**RC: Verdict**

This paper is lucid and intriguing.  Minor modifications to the text are needed before it is published.

**Major comments**

**RC:** It would be a good idea to provide a precise definition of the meaning, values and units of all symbols in a table.  This would clarify things when the text does not specify all the needed details.
**AC:** We have put the list of symbols in Appendix B.

| Symbol | Description | Value and units |
|---|---|---|
| $A$ | Number density of the breakable asperities in the region of contact | - |

| Symbol | Description | Value and units |
|---|---|---|
| $c_w$ | Specific heat capacity of liquid water | 4200 J kg$^{-1}$ K$^{-1}$ |
| $C$ | Asperity fragility coefficient | - |
| $D_a$ | Diameter of aerosol particle | m |
| $D_d$ | Diameter of drop in mode 2 | m |
| $D_{a,m}$ | Median aerosol particle diameter of mode | (m) |
| $D_{i,m}$ | Median aerosol particle diameter of mode A, B or C | nm |
| $D_s$ | Diameter of the smaller colliding particle in ice-ice collisional breakup | m |
| $DE$ | Dimensionless energy | - |
| $DE_{crit}$ | Critical value of dimensionless energy for onset of splashing | 0.2, unless otherwise stated |
| $f$ | Mass fraction of a drop frozen by the end of stage 1 freezing | - |
| $f_{RS}$ | Function of rime splintering | - |
| $F$ | Interpolation function for the onset of fragmentation | - |
| $K_{0(CB)}$ | Collisional kinetic energy at impact | J |
| $L_f$ | Specific latent heat of freezing | $3.3 \times 10^5$ J kg$^{-1}$ |
| $m_r$ | Mass of rime | kg |
| $m_1, m_2$ | Mass of colliding ice particles | kg |
| $m_d$ | Mass of drop in mode 2 | kg |
| $m_i$ | Mass of ice particle in mode 2 | kg |
| $N$ | Number density of aerosol particles | kg$^{-1}$ |
| $N_{CB}$ | Number of ice particles due to ice-ice collisional breakup | - |
| $N_i$ | Total number density of aerosol particles of mode A, B or C | cm$^{-3}$ |
| $N_{M1T}$ | Total number of ice particles due to mode 1 | - |
| $N_{M1L}$ | Total number of large ice particles due to mode 1 | - |
| $N_{M2}$ | Number of ice particles per drop accreted due to mode 2 | - |
| $N_{RS}$ | Number of ice particles due to rime splintering | - |
| $N_T$ | Total number of aerosol particles of mode | kg$^{-1}$ |
| $R_{FL}$ | Rime fraction of the larger colliding particle | - |
| $R_{FS}$ | Rime fraction of the smaller colliding particle | - |
| $t$ | Time | s |
| $T$ | Freezing temperature of water drop | °C |
| $T_0$ | Value of T at maximum of Lorentzian for Eq.5 | °C |
| $T_{B0}$ | Value of T at maximum of Lorentzian for Eq.6 | °C |
| $v_1, v_2$ | Fall speed of colliding ice particles | m s$^{-1}$ |
| $v_d$ | Fall speed of drop in mode 2 | m s$^{-1}$ |
| $v_i$ | Fall speed of ice particle in mode 2 | m s$^{-1}$ |

| Symbol | Description | Value and units |
|---|---|---|
| $w$ | Updraft speed | m s$^{-1}$ |
| $\alpha$ | Equivalent spherical surface area of the smaller colliding particle | m$^2$ |
| $\beta$ | Parameter in Eq. 5 | K$^{-1}$ |
| $\beta_B$ | Parameter in Eq. 6 | K$^{-1}$ |
| $\gamma$ | Parameter related to riming intensity | - |
| $\gamma_u$ | Surface tension of liquid water | 0.073 J m$^{-2}$ |
| $\zeta$ | Intensity of Lorentzian in Eq. | - |
| $\zeta_B$ | Intensity of Lorentzian in Eq. | - |
| $\eta$ | Half width of Lorentzian in Eq. 5 | °C |
| $\eta_B$ | Half width of Lorentzian in Eq. 6 | °C |
| $\kappa$ | Hygroscopicity parameter | 0.61 |
| $\rho_L$ | Density of the larger colliding particle | kg m$^{-3}$ |
| $\sigma_g$ | Standard geometric deviation of the logarithmic distribution | - |
| $\sigma_i$ | Standard geometric deviation of the logarithmic distribution of mode A, B or C | - |
| $\Phi$ | Probability of any drop in the mode 2 splash containing ice | 0.3, unless otherwise stated |
| $\Phi_s$ | Aspect ratio of the smaller colliding particle | - |
| $\Omega$ | Interpolating function for the onset of fragmentation | - |

**RC:** Regarding the treatment of Mode 1 of raindrop-freezing fragmentation by Phillips et al. (2018), it should be emphasized in the introduction of the present paper by James et al. that numbers of secondary ice fragments were based only on lab observations of freezing drops in free-fall. No observations of electrodynamically levitated drops were used. Thus, Phillips et al. (2018) were vindicated, since it has since been shown that free-fall is crucial for fragmentation. Regarding the treatment of Mode 2, I think there is still wide experimental uncertainty introduced by the treatment of numbers of secondary drops emitted per impact with a larger ice surface. Sensitivity tests should be done to explore the realm of uncertainty. Anyway, raindrop-freezing fragmentation does not appear to be very important in most recent studies using the Phillips et al. (2018) scheme.

**AC:** Regarding the Mode 1 parameterisation being based on drops in free fall only, we have added *'Based on the available laboratory literature of drops in free fall only…'*

Regarding Mode 2, we have added the following sentence in the introduction *'However, large experimental uncertainties exist around the treatment of the number of secondary drops emitted per impact with a more massive ice surface.'*

**RC:** I think a more serious issue of the present paper is the lack of dependency on mean droplet size of the H-M process of rime-splintering. This was seen in lab studies by Hallett and Mossop, and was also consistent with aircraft observations of convective clouds by Harris-Hobbs and Cooper (1987) and Blyth et al. (1993). Yes, those studies involved artificial shattering biases on optical probes, but the correction merely introduces a systematic bias and the qualitative correlation with H-M conditions still likely applies. Can the simulations be re-done with a factor depending on mean droplet size somehow, including this in Eq (2), to account for the numbers of cloud-droplets > 24 microns ?

**AC:** We acknowledge that there is a dependence of the rime-splintering process on the mean drop size. This has been communicated by Harris-Hobbs and Cooper (1987), in addition to Mossop (1976). The mean drop size dependence is not included here. The justification is twofold:

(1) often there are processes that lead to broadening on the drop size distribution that parcel models do not typically capture (e.g. entrainment / mixing)

(2) The case where the rime-splintering process had an appreciable effect was the warmer / deeper cloud simulation with natural aerosol (Fig. 10a). As shown in the supplementary material (Fig. S19a) the droplet sizes are > 24 microns in this case.

**RC:** Thus, in view of the empirical uncertainty about splashing in Mode 2 of raindrop-freezing fragmentation, it would be prudent for James et al. to perform sensitivity tests of their cloud simulations by raising the onset threshold of DE (0.2 currently) by a factor of 10 or 30. Also, sensitivity tests on reducing the constant of proportionality by an order of magnitude would also explore the range of experiment uncertainty. Error-bars reflecting this uncertainty from Mode 2 are needed on the control simulation plots of IE ratio.

**AC:** We have completed sensitivity studies on DEcrit for a factor of 10 and 30. See response to Line 170 in the Specific Comments below. The constant of proportionality was reduced by an order of magnitude to 0.01, and lower, (see Figs. 4 & 8) and showed that very little ice enhancement occurred due to mode 2. Hence, showing that further experimental studies are required to help constrain this uncertainty.

**Comments about the Review by Kiselev about treatment of Mode 2**

I am grateful for the lucid comments about raindrop-freezing fragmentation during drop collisions with a larger ice particle (Mode 2) from the reviewer, Kiselev.   It is illuminating to see papers published in colloidal science and fluid dynamics journals of relevance to splashing.

Curiously, in a sense, the two papers cited by the review by Kiselev qualitatively confirm the validity of the general theoretical approach for the treatment of Mode 2 by Phillips et al. (2018) because they show that the splashing onset condition is related to the Weber number.   This Weber number is identical to the dimensionless energy ("DE") of the 2018 scheme, except for a factor of 12.   So, the main import of Kiselev's criticism of the treatment of Mode 2 concerns uncertainty in the empirical estimation of parameters by Phillips et al. as applied in the present paper by James et al.

I would like to clarify one or two details about Kiselev's critique of the treatment of Mode 2 of raindrop freezing fragmentation by Phillips et al. (2018).

First, observations of drop-drop collisions were not the only lab observations used to constrain the splashing scheme in the treatment of Mode 2.  There was also comparison with the observations by Levin and Hobbs (1971) of drops (2.5 mm, 4.2 m/s) falling on a rough copper hemisphere.  Levin and Hobbs observed about 150 splash-drops produced per collision.  They reported that there was onset of splashing at about 0.5 m/s impact speed.   Thus, the dimensionless energy (DE = initial collision kinetic energy divided by drop surface energy) per drop impact was $500 \, D \, v^2 \, /(6 \times 0.073 ) = 50$ (for about 150 splash-droplets per collision) and 0.7 (for onset of secondary droplet emission or splashing).  This critical onset value of DE (0.7) is between one and two orders of magnitude lower than the threshold claimed by Kiselev in the review, who analyzed the Sykes et al. and Charalampous data.

Moreover, Levin and Hobbs observed that target roughness is crucial for the production of secondary drops, and presumably must have selected a rough copper sphere with the aim of better representing the actual roughness of ice precipitation particles, in the quest to study charge separation in clouds. Inspection of the results from Levin and Hobbs seem to suggest an approximate proportionality between number of secondary droplets and DE.  Intriguingly, Levin and Hobbs report observing that a "crown" of fluid would be produced on impact, with the secondary droplets being emitted from the crown (e.g. by jets). On very smooth surfaces, they observed that there would be no crown, and presumably little or no secondary droplet emission (no splashing).

Thus, although Phillips et al. (2018) in the text cited the drop-drop collisions as the source of the critical DE value (0.2), their simultaneous use of the Levin and Hobbs observations of drops on a rough copper hemisphere also approximately support this value (0.7, the same order of magnitude as 0.2).  In fact, Phillips et al. used chiefly the Levin and Hobbs results for the most important parameter of the scheme, namely the coefficient of proportionality between drop number and excess DE, rather than the dropdrop collision data.

Secondly, the review by Kiselev claims that factors of surface roughness, water adsorbed onto the surface and curvature of the surface were not accounted for in the treatment by Phillips et al. (2018).   Is this claim true ?   Yes and no.   Naturally, these factors are greatly sensitive, as observed by Levin and Hobbs for roughness, as noted above.  Yet, Levin and Hobbs knew that these factors were influential and designed their copper sphere experiment so as to be representative for collisions with ice precipitation particles in natural clouds (their copper sphere was dry, as with ice during dry growth by riming, and rough, as is true of most riming;  their ratio of the radii of curvature between the incident drop and

target was about 10%, a plausible order of magnitude for Mode 2 collisions in natural clouds). Their focus was on charge separation in electrified clouds. So the average conditions of these three factors may be implicitly factored into the Levin and Hobbs results and hence into the Phillips et al. (2018) scheme.

By contrast, the two laboratory studies cited by Kiselev involve observations of extremely smooth spherical surfaces. Sykes et al. (2022) used "smooth untreated borosilicate-glass substrates" while Charalampous and Hardalupas (2017) used similarly smooth glass (smoothness of 35 nm). While such studies are theoretically illuminating (they nicely demonstrate that the ratio of drop to target radius of curvature is a sensitive parameter rather than the target radius per se), the observations of Levin and Hobbs about the crucial role of target roughness for the splashing process indicates that they cannot be used quantitatively for constraining Mode 2.

Finally, Schremb et al. (2017) observed splash-droplets from drops of 3 mm in diameter falling at about 2 m/sec on a larger ice target, with a reported Weber number of about 200 (DE value of about 15). Although they did not report splash-drop numbers, analysis of the published photos in their paper and my personal communication with Schremb shows that dozens of splash-drops were emitted per collision. This suggests that for ice, the critical DE value for onset of splashing must be much less than about 15, contrary to the lab results cited by the review. Perhaps onset of splashing in Schremb's experiment would be expected to have occurred at a critical DE value of about unity, given the proportionality noted above inferred from the Levin and Hobbs results.

On the one hand, the Mode 2 treatment by Phillips et al. might have been an over-estimate, since Levin and Hobbs (1971) observed 10 times more splash-drops for a rough curved copper surface than for similar values of drop size, DE and Weber number compared to Schremb et al. with a larger ice surface (a vertical columnar shape). Alternatively, one could argue that the roughness of the surface is more realistic in the case of Levin and Hobbs than for Schremb et al., such that Schremb et al. may have observed too few (unreported) splash-droplets per collision if their ice surface was smoother than for ice precipitation in natural clouds. There is much uncertainty still, but roughness appears to be crucial.

**In summary, there seems to be variability of the splashing among drops among different types of larger solid surface for different lab experiments, even for similar macroscopic dynamics of collision. The observations by Levin and Hobbs of drops falling on a rough copper sphere were designed for representativeness when studying charging in collision of ice particles in natural clouds. They found target roughness is crucial for secondary droplet emission (and for splashing in Mode 2). Ice precipitation in natural clouds is almost instantly roughened by riming. Hence, the smooth-glasstarget observations cited in the review by Kiselev, though useful theoretically for the physics of splashing, must greatly under-estimate the real splashing in natural clouds during Mode 2 by ice precipitation. The smooth-glass-target observations cannot be supposed to be more reliable than observations by Levin and Hobbs.**

**Of course, future lab experiments on this topic will be invaluable to elucidate Mode 2 of raindropfreezing fragmentation in a representative manner.**

**Specific comments**

**RC:** Line 90:  More details of the model description are needed in the text.  What are the microphysical species represented ?  What processes of growth are treated ?  Is raindrop-freezing treated ?
**AC:** We have added further details highlighted in yellow below.

**2.1  Model description**

All simulations in this paper used the bin microphysics model (BMM, https://github.com/UoM-maul1609/bin-microphysics-model an adapted version of the control model described in Fowler et al. (2020), developed at the University of Manchester. The bin microphysics model includes activation of cloud droplets and condensation / deposition from water vapour, collision and coalescence of water drops, inertial impaction of aerosol particles, ice-ice aggregation, riming and secondary ice processes.

Aerosol particles are represented as multiple log-normal modes of different chemical compositions (externally-mixed modes). Each externally-mixed mode is described by an internal mixture that has the same chemical composition across all sizes. The BMM is initialised by summing multiple log-normal size distributions:

$$\frac{dN}{d\ln(D_a)} = \frac{N_T}{\sqrt{2\pi}\ln\sigma_g}\exp\left[-\frac{\ln^2\left(\frac{D_a}{D_{a,m}}\right)}{2\ln^2\sigma_g}\right] \tag{1}$$

where $N$ is the number density of aerosol particles, $D_a$ is the aerosol particle diameter for the mode, $N_T$ is the total number of aerosol particles in the mode, $D_a,m$ is the median aerosol particle diameter for the mode and $\sigma_g$ is the geometric standard deviation of the logarithmic distribution.

The activation of cloud condensation nuclei is calculated from condensation of liquid water onto the aerosol particles with the equilibrium vapour pressure described by $\kappa$-Kohler theory, where the size and hygroscopicity of an aerosol particle is related by a single parameter, $\kappa$ (Petters and Kreidenweis, 2007). The rate of drop growth via diffusion takes into account mass accommodation through modified diffusivity and conductivity terms (Jacobson, 2005; H.R. Pruppacher, 2010). While initial growth of cloud drops occurs via the diffusional growth equation, later growth to raindrops occurs via the collision-coalescence process. Collision-coalescence growth is described by the stochastic collection equation which is solved using the method of Bott (1998), and collisional efficiencies are calculated according to the Long (1974) kernel. The model treats binned distribution for liquid particles and a separate binned distribution for ice particles.

Homogeneous freezing from a supercooled water drop follows the method described in Koop et al. (2000). Heterogeneous freezing occurring via ice nucleating particles is calculated using the DeMott et al. (2010) ice nucleation parameterisation, which requires knowledge of the aerosol particle number density with diameter $\geq 0.5$ µm. The same parameterisation is used to describe the freezing of rain drops. We first determine the number of aerosol particles with diameter $\geq 0.5$ µm contained within a rain drop and multiply this by the number concentration of particles within the same category. The number of ice nucleating particles that are active is then calculated using the DeMott et al. (2010) parameterisation. This is scaled by how many aerosol particles are contained within the drop giving the number of frozen drops in that category.

Ice particle growth from the vapour is described using a growth rate which takes into account mass accommodation through modified diffusivity and conductivity terms (Jacobson, 2005). Once formed, ice particles grow according to the model described in Chen and Lamb (1994). The ice particle bins carry properties that are averaged within a mass bin. These properties are the aspect ratio of the ice crystals; the volume of the crystals; the rime mass; and the number of ice crystal 'monomers' per ice

particle. Ice-ice aggregation and riming are also calculated using the method of Bott (1998), which is modified to transport the extra properties discussed above. The terminal velocity of ice particles is determined from Heymsfield and Westbrook (2010) based on the mass and shape of the ice particles.

**RC:** Line 120:   Why is the observed dependency on cloud-droplet size not represented in Eq (2) ?  Drops larger than 24 microns are needed for rime-splintering, as shown by Hallett and Mossop's later work.
**AC:** See above.

**RC:** Line 148: It might be good to clarify that Mode 2 is for raindrops.

**AC:** We have added the following sentence 'For fragmentation to occur, the drop diameter must be greater than 0.15 mm and the mass of the drop must be less massive than the ice particle.'

**RC:** Line 155: Need to say that NM2 is the number of fragments per drop accreted.

**AC:** Added. Sentence now says: 'Then the number of fragments per drop accreted due to M2 ($N_{M2}$)...'

**RC:** Line 167: It is unclear what is meant here: "We ran all possible combinations using the methodology described by Montgomery (2013); Teller and Levin (2008). For k factors there are $2^k$ different combinations". What is k ? What factors are being mentioned ? What combinations are being mentioned ?

**AC:** We have rephrased the sentence 'We ran all possible combinations of SIP mechanisms which gave a total of 15 simulations for each sensitivity investigated in addition to a control simulation with no SIP mechanisms. The combinations are given in Table A1 of the Appendix.'

**RC:** Line 170: as noted above, there is a need to perform sensitivity tests with respect to the number of splash droplets per collision predicted for Mode 2 of raindrop-freezing fragmentation. This is due to the experimental uncertainty.

**AC:** We have added M2 DE$_{crit}$ as a sensitivity study to Table 2 and added the following section:

**3.2.4  M2 $\Phi$ and DE$_{crit}$**

The M2 parameterisation depends on both the probability of the splash containing ice ($\Phi$) and the onset of splashing ($DE_{crit}$) as shown in Eq. 9. Both of these parameters are determined on experimental studies and present a source of uncertainty. Therefore, we individually investigated the M2 $\Phi$ and $DE_{crit}$ for shallower clouds with an updraft speed of 2 m s$^{-1}$ for both natural and near-city aerosol size distributions and warmer and colder cloud bases.

The results are given in Sections 4.1.4 and 4.2.4 for a natural and near-city aerosol size distribution respectively.

**4.1.4  Sensitivity test: M2 DE$_{Crit}$**

The M2 parameterisation given in Eq. 9 requires the onset of splashing, denoted as DE$_{crit}$. We used the value of 0.2 given in Phillips et al. (2018) based on laboratory data of drops colliding on roughened copper hemispheres from Levin et al. (1971). To test the sensitivity of ice enhancement to DE$_{crit}$ during M2 simulations, we ran simulations for warmer and colder cloud bases with natural aerosol size distribution and updraft speeds of 2 m s$^{-1}$ using the following DE$_{crit}$ values: 0.2, 3 & 6. The results are plotted in Figure 5.

Shallower cloud with natural aerosol
size distributions: DE$_{crit}$

**Figure 5.** M2 ice enhancement against simulation time for three DE$_{crit}$ values (0.2, 3 and 6) for a shallower (1.3 km) cloud with natural aerosol size distributions and updraft speed of 2 m s$^{-1}$. Warmer refers to cloud base temperatures of 7 °C, and colder refers to cloud base temperatures of 0 °C.

**4.2.4   Sensitivity test: M2 DE$_{Crit}$**

Figure 9 shows the M2 ice enhancement against simulation time for three DE$_{crit}$ values, 0.2, 3 and 6, for shallower clouds with a near-city aerosol size distribution. In general, lower values of DE$_{crit}$ had earlier maximum ice enhancement peaks compared to higher values of DE$_{crit}$. The maximum ice enhancement peaks were greater for lower DE$_{crit}$ values.

[Figure]

**Figure 9.** M2 ice enhancement against simulation time for three DE$_{crit}$ values (0.2, 3 and 6) for a shallower (1.3 km) cloud with near-city aerosol size distributions and updraft speed of 2 m s$^{-1}$. Warmer refers to cloud base temperatures of 7 °C, and colder refers to cloud base temperatures of 0 °C.

We have also expanded the discussion of DE$_{crit}$.

Other uncertainties in the M2 parameterisation include the critical value for onset of splashing ($DE_{cit}$). For splashing to occur, the dimensionless energy must be greater than $DE_{crit}$. Phillips et al. (2018) estimated a value for $DE_{crit}$ of ~0.2 based on room temperature experiments of colliding drops by Low and List (1982) and acknowledged that this was a source of uncertainty. The equation used to describe number of secondary liquid drops was further constrained using laboratory experiments by Levin et al. (1971). These experiments consisted of 2.5 mm drops impacting on a rough copper sphere at room temperature. The number of secondary drops formed in collision experiments is sensitive both to the geometry and impact material; hence the expression used to describe the number of liquid fragments presents another source of uncertainty. We increased the $DE_{crit}$ by a factor of 10 and 30 to demonstrate the sensitivity of M2 ice enhancement on the DE$_{crit}$, as

shown in Figs. 5 and 9. When the $DE_{crit}$ was increased by a factor of 10 or 30, the ice enhancement decreased compared to the stated $DE_{crit}$ in Phillips et al. (2018), but ice enhancement still occurred. Despite the uncertainties within the M2 parameterisation, and in agreement with Phillips et al. (2018), it is more erroneous not to include M2 within simulations. Our laboratory experiments strongly suggest that the M2 SIP mechanism could be active (James et al., 2021), and further laboratory studies into this mechanism will help reduce the uncertainties listed.

**RC:** Line 180:  There is over-loading of the terms "Mode 1" and "Mode 2" in Table 3 to refer to aerosol modes.  Can a different label be used for aerosol modes (e.g., Modes A, B, C) ?
**AC:** Changed.

**RC:** Line 248:  A problem with the DeMott (2010) parameterization is that there is no dependency on aerosol chemistry and that on size is only represented in a basic way, using a threshold.  There is no direct dependency on dust surface area.  If the aerosol chemical composition was observed for the simulated cases, then other schemes are possible (e.g. Phillips et al. 2008, 2013).
AC: This is correct, but there are also often no measurements of the chemistry or dust surface area. Hence, we choose a simpler parameterisation to represent primary INP.

**RC:** Line 298:  the plural of "maximum" is "maxima", as it is a Latin word.
**AC:** Changed.

**RC:** Figure 5:  I think it would be wise to include an error-bar in the simulations arising from errors in the parameters of Mode 2, in view of the discussions above and by Kiselev's review.   Also, a logarithmic yaxis would be more appropriate in view of the uncertainty.
AC: It is not straightforward to do this due to the non-linearities involved and the structural uncertainty in the parameterisation (as discussed by the Reviewer 1 there are not only parametric uncertainties, but also structural). However, an idea of the uncertainties can be found looking at Figs. 4 & 8 (Mode 2 $\Phi$) and Figs. 5 & 9 (Mode 2 $DE_{crit}$) as unknowns.

The purpose of the paper is to show the potential importance of M2 compared to the other mechanism, given our best guess.

**RC:** Line 568:  the sentence does not make grammatical sense.  Did Sotiropolou et al. explicitly treat Mode 2 ?

**AC:** We have rephrased the sentence to the following:

'Other studies which modelled the M2 SIP mechanism with the M1 SIP mechanism did not provide a breakdown of the contribution from M1 and M2 (Zhao et al., 2021, Zhao and Liu, 2022, Huang et al., 2022, Karalis et al., 2022). Two studies found that M2 combined with M1 was not an effective SIP mechanism in the simulated conditions (Sotiropoulou et al., 2020, Georgakaki et al., 2022). Sotiropoulou et al. (2020) stated that this was due to their thermodynamic conditions with relatively cold cloud base temperatures and Georgakaki et al. (2022) stated that this was due to the drops being too small to initiate M1 + M2 in their simulated conditions.'

**RC:** Line 578:  Need to explain the mechanism for how high IN concentrations in the model suppress SIP and coalescence.  Do they cause subsaturation with respect to liquid, from vapour growth of ice, and is this what curbs the droplet growth, inhibiting raindrop-freezing fragmentation ?  Or is the liquid depleted by riming of the primary ice ?

As argued by Yano and Phillips (2011), there is an upper limit to the ice concentration that depends on vertical velocity and temperature, corresponding to the onset of subsaturation with respect to liquid. When the cloud-liquid evaporates and the cloud becomes ice-only, all SIP ceases.   Waman et al. (2021) nicely showed this happening, so that paper needs to be cited.   Could the lack of SIP when INPs are numerous be due to this upper ceiling being reached by the primary ice, or even approached ?

**AC:** The high IN concentration in the model suppresses SIP and coalescence due to the Wegener-Bergeron Findeison process. Below are plots of liquid water content, ice water content and ice enhancement over the control simulation for a mode 2 simulation with a natural aerosol size distribution. These plots show an extended simulation run compared to the plots in the paper.

[Figure]

[Figure]

[Figure]

- When the simulations are run for longer, all INP concentrations in the colder simulations eventually deplete the LWC, but the x10 INP occurs earlier in the simulations and has a different profile.
- The IWC saturates around 1500 g/kg for all simulations with the x10 INP occurring earlier. The IWC is higher earlier in x10 INP simulations, corresponding to a lower LWC earlier in the simulations.
- Only one ice enhancement peak was observed for x10 INP, whereas for the lower INP concentrations there are two peaks, the first smaller and the second larger.

The Wegener-Bergeron Findeisen process occurs earlier in the x10 INP simulations as initially there is a higher IWC earlier in the simulations, which causes more vapour to condense onto the ice suppressing the coalescence process, hence reducing SIP. Whereas, for the x0.1 and x1 INP simulations, coalescence occurs earlier due to lower IWC allowing SIP formation, which increases ICNC.

We have added the liquid and ice water content figures to the supplement and added the following paragraph:

the RS mechanism was 'turned off' due to the Wegener-Bergeron Findeisen (WBF) process (Crawford et al., 2012). Figures S.25 & 26 of the Supplement show the liquid and ice water contents for M2 simulations with a natural aerosol size distribution, respectively, with extended simulation runtimes, demonstrating that the WBF process also occurred in our colder cloud base simulations. However, the WBF process occurred earlier, with a different profile, in the ×10 INP concentration simulation compared to the ×1 or ×0.1 INP concentration simulations as initially there was a higher ice water content in the simulations, which allowed more water vapour to condense onto the ice suppressing the coalescence process, hence reducing SIP. Whereas, for the ×0.1 and ×1 INP simulations, coalescence occurred earlier due to the lower ice water content allowing SIP formation, which increased the ice enhancement.

References
Harris-Hobbs, R. L., and W. A. Cooper, 1987: Field Evidence Supporting Quantitative Predictions of Secondary Ice Production Rates. J. Atmos. Sci., 44, 1071–1082, https://doi.org/10.1175/1520-0469(1987)044<1071:FESQPO>2.0.CO;2.
Mossop, S.C. (1976), Production of secondary ice particles during the growth of graupel by riming. Q.J.R. Meteorol. Soc., 102: 45-57. https://doi.org/10.1002/qj.49710243104